# Systems-level analyses of protein-protein interaction network dysfunctions via epichaperomics identify cancer-specific mechanisms of stress adaptation

Anna Rodina [1,14], Chao Xu[1,14], Chander S. Digwal [1,14], Suhasini Joshi[1,14], Yogita Patel[2,14], Anand R. Santhaseela [1], Sadik Bay[1], Swathi Merugu[1], Aftab Alam[1], Pengrong Yan[1], Chenghua Yang[1,11], Tanaya Roychowdhury[1], Palak Panchal[1], Liza Shrestha[1], Yanlong Kang[1], Sahil Sharma [1], Justina Almodovar[1], Adriana Corben[3,12], Mary L. Alpaugh [1,13], Shanu Modi[4], Monica L. Guzman [5], Teng Fei [6], Tony Taldone[1,15], Stephen D. Ginsberg[7,8], Hediye Erdjument-Bromage[9], Thomas A. Neubert[9,15], Katia Manova-Todorova[10,15], Meng-Fu Bryan Tsou[10,15], Jason C. Young [2,15,16], Tai Wang [1,15] ✉ & Gabriela Chiosis [1,4,15] ✉

Systems-level assessments of protein-protein interaction (PPI) network dysfunctions are currently out-of-reach because approaches enabling proteome-wide identification, analysis, and modulation of context-specific PPI changes in native (unengineered) cells and tissues are lacking. Herein, we take advantage of chemical binders of maladaptive scaffolding structures termed epichaperomes and develop an epichaperome-based 'omics platform, epichaperomics, to identify PPI alterations in disease. We provide multiple lines of evidence, at both biochemical and functional levels, demonstrating the importance of these probes to identify and study PPI network dysfunctions and provide mechanistically and therapeutically relevant proteome-wide insights. As proof-of-principle, we derive systems-level insight into PPI dysfunctions of cancer cells which enabled the discovery of a context-dependent mechanism by which cancer cells enhance the fitness of mitotic protein networks. Importantly, our systems levels analyses support the use of epichaperome chemical binders as therapeutic strategies aimed at normalizing PPI networks.

Proteins do not function in isolation[1]. Biochemical interactions of thousands of proteins give rise to cell-type and cell-state specific networks that result in phenotypic manifestations of cells, tissues and entire organisms[2–7]. In this context, the intricate cellular networks of proteins linked through interactions, i.e. protein-protein interaction (PPI) networks, are high-fidelity maps of how disease-specific stressors, including genetic lesions, proteotoxic and environmental insults, individually or combined, alter proteome connectivity and perturb the system as a whole[8–11].

PPI networks are context dependent[3,7,12–14], and addressing disease mechanisms of stressor adaptation (i.e. how stressors result in and impact a phenotype) requires a systems-level approach to the study of protein connectivity in a cell- and tissue-type specific manner[3,7,13]. However, direct measurement of how thousands of

proteins interact in diseased cells and tissues, and moreover, understanding the functional impact of these intricate context-specific PPIs, are technically challenging using current interrogative approaches[7,15]. A solution is provided by discoveries in disease biology that link stressor-induced protein connectivity perturbations to the formation of epichaperomes[2,16]. These pathologic scaffolds, composed of tightly bound chaperones, co-chaperones, and other factors, mediate how proteins anomalously interact and organize inside cells, which aberrantly affects the function of PPI networks, and in turn, cellular phenotypes[2,17–20]. Capturing epichaperomes and the proteome at large negatively impacted by these critical scaffolds therefore provides informative clues for direct access to context-dependent PPI perturbations in diseases, and to the functional outcome of such changes in native biological systems[17–19,21]. We recently introduced the term epichaperomics to describe the affinity-purification method that uses epichaperomes as baits to analyse context-specific alterations in protein connectivity and study disease specific interactomes[22].

Epichaperomics is an affinity-purification method where epichaperomes are the natively expressed 'biological' baits[22] (as opposed to classical affinity purification where a tagged protein is exogenously introduced into a cell). A multitude of endogenous baits (i.e. the distinct epichaperome structures characteristic of a specific cellular context), each having individual interactors, is natively present and interacts tightly with its interactors. Chemical probes serve as a tool to trap, enrich, and isolate the epichaperome/interactor complexes from the protein mixture, and thus have been instrumental in the identification and study of PPI network dysfunctions through epichaperomics. By eliminating the need for exogenous introduction of a tagged protein as bait, epichaperomics is applicable for dissecting native cellular states, and thus appropriate for the analysis of both cultured cells and primary specimens. Probes however are currently limited to capturing heat-shock protein 90 (HSP90)-incorporating epichaperomes[19,23]. Chaperone composition in epichaperomes is context- and stressor-dependent[17–20], and thus a chemical probe toolbox directed to key chaperones, other than HSP90, in the context of epichaperomes is needed to fully capitalize on knowledge that can be gained through epichaperomics.

Discovery of epichaperomics probes is challenging. To be effective, probes should not only favor epichaperomes over chaperones but should also trap and isolate thousands of context-dependent epichaperome-proteome complexes from a given protein mixture[22]. This is not trivial considering both the abundance and ubiquity of chaperones in comparison to epichaperomes as well as the structural similarities between these two protein pools[17,19–21].

We here overcome these roadblocks and introduce a toolset of HSP70 ligands and their derived-probes along with demonstrating their preference for HSP70-incorporating epichaperomes, referred to here as epiHSP70s, over the abundant and ubiquitous HSP70s. HSP70 paralogs, especially heat shock cognate 70 (HSC70), are recruited into epichaperomes in cancer, Alzheimer's disease (AD) and Parkinson's disease[16–19] and an epiHSP70s epichaperomics probe would provide direct access to proteome connectivity perturbations in these diseases and to the functional outcome of such changes in native biological systems[22,24]. To enable these high value studies, we here also provide a methodological roadmap in using the epiHSP70s probes to perform epichaperomics studies. It encompasses 1. how to design and perform an epiHSP70 epichaperomics study, 2. how to analyse the epichaperomics datasets, and finally, 3. how to validate the output of the epiHSP70s epichaperomics studies. As proof-of-principle, for validation, we apply the epiHSP70s epichaperomics to cancer cells to identify mitotic regulator proteins whose connectivity is rewired to alter mitotic spindle formation and discover a context-specific mechanism how the fitness of mitotic protein pathways is enhanced in cancer.

## Results
### Probe preference for epiHSP70s over HSP70s
The human HSP70 chaperones, HSC70 and HSP70 (referred here as HSP70s), are composed of two major domains: an ~45 kDa, N-terminal nucleotide binding domain that contains the regulatory ATP/ADP binding pocket and a C-terminal ~25 kDa substrate binding domain joined together by a flexible linker[25,26]. Through a combination of computational, biochemical and functional approaches we discovered and reported on a druggable allosteric pocket located in the N-terminal domain of HSP70s[27,28]. This cavity contains a reactive cysteine (Cys267) and we designed the ligand YK5, an irreversible inhibitor to covalently interact with this residue upon binding to HSP70s (Fig. 1a, Supplementary Figs. 1–6 and Supplementary Note 1)[27,29]. Binding of YK5 to HSP70s was computationally predicted, experimentally demonstrated in cellulo, biochemically and functionally, and further through medicinal chemistry and structure-activity relationship studies[27,29,30]. We used YK5-B, a biotinylated analog of YK5 (Fig. 1a, Supplementary Figs. 1–6 and Supplementary Note 1), to confirm proteome-wide selectivity of these ligands for HSP70s in cancer cells[27,31]. Specifically, we demonstrated that when YK5-B was incubated with the thousands of proteins found in a cancer cell, it isolated HSP70s as the sole direct interactors of the probe[27,31]. Thus, despite containing an aminopyridine-derived acrylamide moiety, YK5 is not excessively reactive and, in addition, has an appropriate fit in the active site of the target[27,29,30].

Whether YK5 and YK5-derivatives however prefer binding to epiHSP70s over HSP70s remains unknown. To address this, we include in our study several YK-derivatives, both covalent and non-covalent interactors, with a spectrum of on-target potencies (IC$_{50}$ for epiHSP70s disruption in cancer cells, YK5 = 5 μM, YK198 = 500 nM and LSI137 = 60 nM, see Fig. 1a, Supplementary Figs. 1–6 and Supplementary Note 1). The scientific premise for including several YK-derivatives with a spectrum of on-target potencies is that if these probes show similar biological effects when tested at concentrations that reflect their target engagement, this is supportive of specific, on-epiHSP70s target activity. We also include YK5-B and the negative control YK56, structurally similar to YK5 (i.e. contains the reactive acrylamide and the YK5 scaffold), but chiefly inactive on epiHSP70 (Fig. 1a, Supplementary Figs. 7–13 and Supplementary Note 1).

In contrast to folding chaperone complexes, which have evolved to be dynamic and short-lived[32], epichaperomes are long-lived heterooligomeric assemblies of tightly bound chaperones, co-chaperones and other factors[17,19]. HSP70s, principally HSC70, are found in epichaperomes alongside other chaperones, co-chaperones and scaffolding proteins, such as HSP90, HSP110, CDC37, AHA1, HOP (ref. 17–19,21 and Fig. 1a). Therefore, when epichaperome-positive cell homogenates are assayed by Native-PAGE gels, a number of distinct and indistinct high-molecular-weight species are observed for epichaperomes in addition to the main band(s) characteristic of chaperones (ref. 17–19,21 and Fig. 1a). This biochemical property enabled us to test the binding preference of YKs for epiHSP70s assemblies over HSP70s assemblies by analysing, through affinity capture techniques, the impact of beads-immobilized YKs on the two pools, in cells with equivalent total HSP70s levels, but distinct in their epichaperome content (i.e. cancer cells with high-epiHSP70 or low-epiHSP70 expression but equivalent HSP70s, ref. 17 and below).

We first tested the HSP70s pool captured, and the pool depleted, by a solid support immobilized YK5 (i.e. YK5-B beads). Epichaperomes represent a fraction of the total chaperone pools[17,33], and thus epiHSP70s are a small amount when compared to the abundant HSP70s (i.e., 5-35%, depending on the cancer cell line, ref. 17 and Fig. 1a). Single capture experiments using increasing concentrations of YK5-B immobilized on beads captured only a fraction of the cellular HSP70s pool (Fig. 1b and Supplementary Fig. 13a, b). A similar result was found when we conducted sequential capture experiments with equal amounts of

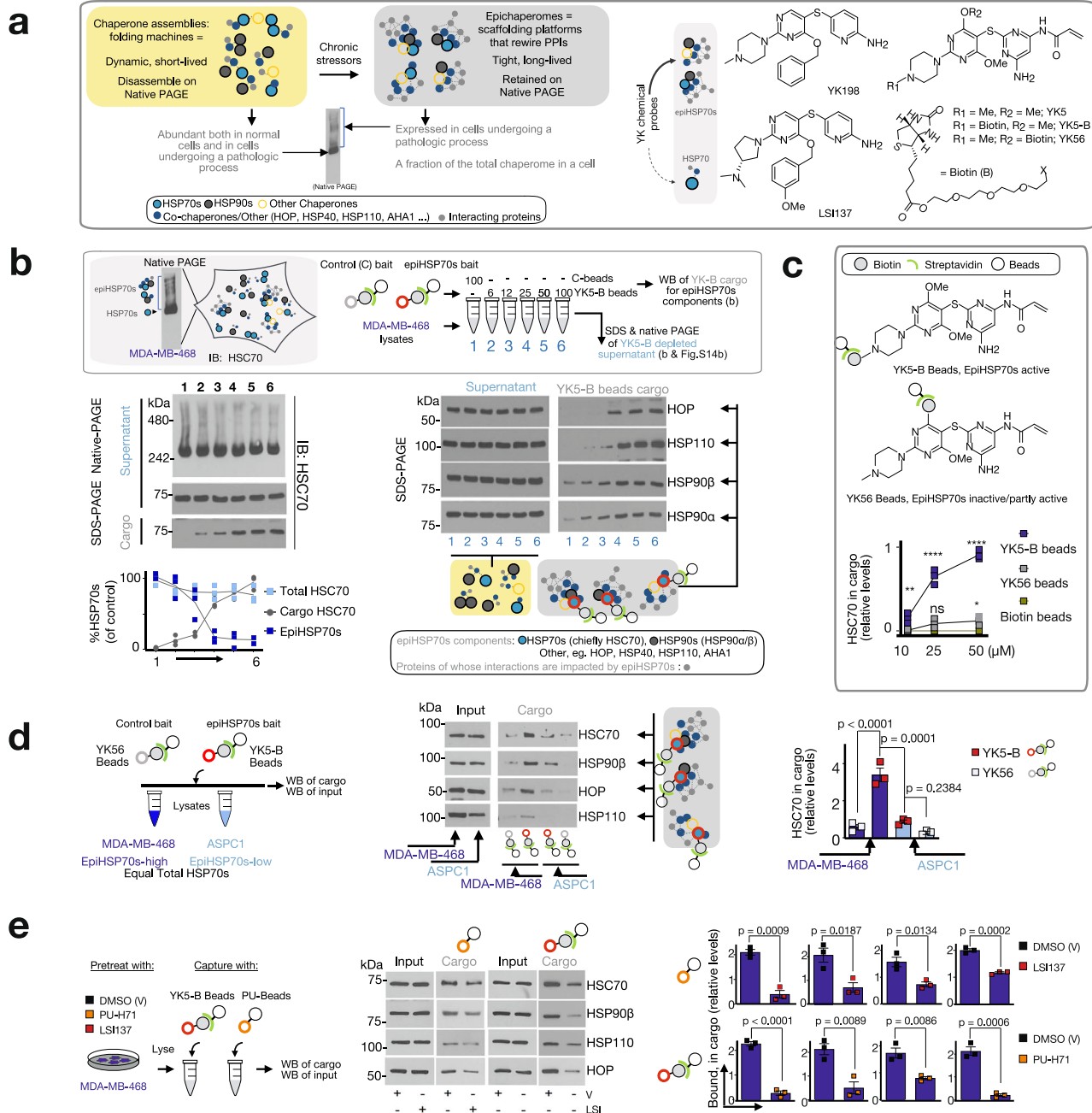

**Fig. 1 | Characterization of the epiHSP70s probes through affinity purification techniques. a** Schematic showing the biochemical and functional distinctions between epiHSP70s and HSP70s (left). Chemical structure of several YK-type ligands (right). See also Supplementary Figs. 1–13. **b, c** Affinity purifications with indicated probe concentrations performed in MDA-MB-468 cell homogenates. Data are presented as mean, two-way ANOVA, $n = 3$, $p < 0.0001$, F (2, 36) = 80.18 (**b**) and two-way ANOVA, $n = 3$, $p < 0.0001$, F (2, 18) = 390.4 (**c**), with Dunnett's post-hoc. YK5-B versus Control, $p = 0.0013$, $p < 0.0001$ and $p < 0.0001$ and YK56 versus Control, $p = 0.9796$, $p = 0.0655$ and $p < 0.0116$ at 10, 25 and 50 μM, respectively. See also Supplementary Fig. 14a–c. **d** Epichaperome components captured by the YK5-B probe and the control probe YK56 in epiHSP70s-high (MDA-MB-468) and -low

(ASPC1) cancer cells, which have equal total chaperone levels, as indicated. Data are presented as mean ± s.e.m., one-way ANOVA, $n = 3$, $p < 0.0001$, F (3, 8) = 49.61, with Tukey's post-hoc. **e** EpiHSP70s components captured by the probes in cells pretreated with vehicle, LSI137 (1 μM) or PU-H71 (1 μM), as indicated. Data are presented as mean ± s.e.m., $n = 3$, unpaired two-tailed $t$-test; df = 4; PU-beads: $t = 8.878$, 3.823, 4.226, and 12.50 and YK5-B beads: $t = 16.04$, 4.759, 4.802 and 9.947 for HSC70, HSP90β, HOP and HSP110, respectively. **d, e** Protein amount loaded for Input represents 10% of the protein amount incubated with the beads. Abbreviations: HSP90, heat shock protein 90; HSC70, heat shock cognate 70; HOP, HSP-organizing protein; AHA1, activator of HSP90 ATPase activity 1. Source data are provided as a Source Data file.

YK5-B beads (Supplementary Fig. 14a, c). Conversely, an antibody with no specificity for the individual HSP70s pools, captured the entire cellular content of this protein (Supplementary Fig. 14a, c).

The HSP70s pool captured by YK5-B beads is the fraction corresponding to epiHSP70s. This was demonstrated by Native-PAGE and SDS-PAGE analyses of the supernatant of YK5-B beads-depleted

homogenates as evidenced by the disappearance, in Native-PAGE, of high-molecular-weight species characteristic of epichaperomes, but not the main band(s) characteristic of chaperones (Fig. 1b and Supplementary Fig. 14a, b), which was paralleled, as seen on Western blot, by enrichment of epiHSP70s components in the YK5-B beads cargo (Fig. 1b). EpiHSP70s depletion by YK5-B beads resulted in a minor

decrease in total chaperone levels (Fig. 1b and Supplementary Fig. 14b, c, see SDS-PAGE analyses of the supernatant), consistent with only a fraction of cellular chaperones being epichaperome constituents[17]. Also supportive of preferential epiHSP70 capture by YK5-B is that YK56 beads, consisting of a solid-support immobilized YK5 derivative used as a negative, less active, control, captured significantly less amounts of epiHSP70s constituent chaperones (Fig. 1c) and epiHSP70s inter-actors (see further), than the YK5-B beads.

MDA-MB-468 and ASPC1 cells are cancer cells with comparable levels of HSP70s and of other epiHSP70s component chaperones but differentiated by their epichaperomes content, with MDA-MB-468 being epiHSP70s-high and ASPC1 epiHSP70s-low (refs. [17,34] and Supplementary Fig. 14a). We found YK5-B beads captured significantly more epiHSP70s-component chaperones, such as HSC70, HSP90, HSP110 and HOP, in MDA-MB-468 than in ASPC1 cancer cells (Fig. 1d).

HSC70 co-exists with HSP90 in a subset of epichaperomes[17,19,21]. PU-H71 is an epichaperome probe that preferentially interacts with HSP90 when this chaperone is incorporated into epichaperomes[17,19]. Supportive of the preferential interaction of YKs for epiHSP70s over HSP70s, we found significantly less epiHSP70s component chaperones (i.e. HSC70, HSP90, HSP110, HOP) captured by YK5-B beads from cells pretreated with PU-H71 (Fig. 1e). Analogously, PU-H71 immobilized beads (PU-beads) captured significantly less epiHSP70s-component chaperones in cells pre-treated with LSI137 (Fig. 1e).

The distinct nature of epiHSP70s and HSP70s assemblies, and the impact of YKs on each species within the physiological context of a cell, can also be studied and differentiated through techniques complementary to affinity purifications, such the cellular thermal shift assay (CETSA)[35]. CETSA allows quantitative measurement of the stability of proteins and is based on the principle that heat-induced protein unfolding leads to rapid precipitation in the cellular environment. Small molecule binding influences the thermal stability of a given protein and the method is often used to evaluate interaction of a given small molecule with a protein. CETSA can similarly evaluate and confirm binding of a small molecule probe to a stable protein complex (in this context, epiHSP70s) as opposed to individual proteins or dynamic protein complexes (in this context, folding or free HSP70s). Interacting proteins residing in stable complexes (in this context, epiHSP70 components such as HSP90, HSP110 and others) would be prone to being influenced upon YK binding, in contrast to dispersed proteins in a cell. Importantly, YK impacted proteins within stable complexes would more likely co-aggregate upon heat denaturation and therefore epiHSP70 components would exhibit similar, non-random progressive insolubilities when subjected to an increasing temperature gradient[36]. In other words, in epiHSP70s expressing cells, YKs should impact the thermal stability of HSC70 and of other epiHSP70s component chaperones (e.g., HSP90, HSP110 and others), as opposed to only HSC70 (Fig. 2a, schematic).

In epiHSP70 expressing cells treated with each of the three evaluated YKs—YK5, YK198 and LSI137—we observed in CETSA, performed both in cells and cell homogenates, a significant thermal stabilization not only of HSC70 but also of other epiHSP70s component chaperones and co-chaperone (i.e. HSP90s, HSP110, HSP40, HOP, AHA1) (Fig. 2b–f and Supplementary Fig. 15a, b). For each tested YKs, epiHSP70s-component chaperones followed a similar stability profile. Because HSC70 and HSP70, but not other proteins, are direct YK binders (ref. [27,31] and see below), these findings are in strong support of these proteins being acted upon by YKs as part of epichaperomes rather than as individual chaperones or as dynamic folding chaperone assemblies. YKs had significantly less impact on the thermal stability of epiHSP70s component chaperones in ASPC1 cells (epiHSP70s-low, HSP70s-high) when compared to MDA-MB-468 cells (epiHSP70s-high, HSP70s-high), also in strong support of their preference for epiHSP70s over HSP70s (Fig. 2f).

Not all small molecule chaperone ligands act productively on epichaperomes (i.e. completely suppress epichaperomes and thus are

able to shut-down their pathologic function)[19,21]. When we compared YKs to HSP70 ligands with distinct biochemical mechanisms, we observed VER155008, a ligand that inserts into the ATP/ADP binding pocket of HSP70s, and MKT077, a ligand that binds to an allosteric pocket in HSP70s distinct from that occupied by YKs[37,38], only minimally recapitulated the biochemical profile induced on epiHSP70s by YKs (Fig. 2d, e).

Collectively, these numerous complementary biochemical experiments (e.g. affinity purifications and CETSA), combined with functional validation (see below), and the use of both cellular and chemical positive- and negative-controls, strongly support the preference of YKs for epiHSP70s over HSP70s.

### Trapping of epiHSP70s bound to interactors

Affinity purification and CETSA experiments together with Native-PAGE analyses of epichaperomes indicate that YKs stabilize epiHSP70s along with interacting proteins prior to epichaperome disassembly. To further support this observation, we performed a time-dependent analysis of epichaperomes in cells treated with YKs by analysing epiHSP70s component chaperones and interactors through Native PAGE, YK5-B affinity purification and through immunoprecipitation (IP) with an HSP70-specific antibody (Fig. 3a and Supplementary Fig. 15c, d). We also monitored steady-state levels of these proteins through western blot (Fig. 3b).

Immediately after the addition of YKs to the cells, we observed what appeared as a transient increase in epichaperome signal on Native-PAGE (Fig. 3c) but no change in the total chaperone levels on SDS-PAGE (Fig. 3b). Immunoprecipitation (IP) with an HSP70-specific antibody, along with affinity purification and CETSA (Figs. 1, 2), support this signal increase is due to trapping or stabilization of epiHSP70s components by YKs upon binding to epiHSP70s, rather than an increase in epiHSP70 levels (Fig. 3d, see increase in the levels of epiHSP70s-component chaperones bound to HSC70 but no increase in their total levels, Fig. 3b). The YK-stabilized epiHSP70s assemblies contained protein interactors, such as evidenced for signal transducer and activator of transcription 3 (STAT3) by CETSA (Fig. 2b), IP (Fig. 3e) and analysis of the YK5-B cargo (see below). We recapitulated these findings using reconstituted systems, to show the biochemical effect of YKs on HSP70s manifests majorly at the allosteric release of the interactor protein, impairing its release and delaying ATP insertion. YKs only minorly affected the chaperone ATPase activity of HSP70s (Supplementary Fig. 16a–h, Supplementary Note 2).

In aggregate, our data indicate for YKs both a preference for epiHSP70s over HSP70s and a binding mechanism whereby epiHSP70s assemblies are initially trapped upon ligand binding along with their interacting proteins. These features propose the YK5-B probe may be applicable for unbiased systems level investigation and modulation of disease-specific PPI network dysfunctions (Fig. 4a, b)[24], a hypothesis we investigate below.

### YK5-B epichaperomics

We unbiasedly assayed through mass spectrometry (MS) the inter-actors isolated by YK5-B in the epiHSP70s-high MDA-MB-468 cancer cell line. Experiments were performed by first adding the cell-permeable YK5-B probe (or an epiHSP70s-inert probe, as control) to live cells and then isolating protein complexes on streptavidin-coated beads following cell lysis. Alternatively, we used the streptavidin bead immobilized YK5-B (or the control probe) to isolate protein complexes from already lysed cells. Upon release of proteins from the solid support by SDS, we applied the protein mixture to SDS-PAGE followed by in-gel digestion and MS for protein identification and quantification (Fig. 5a and Supplementary Fig. 15d). YK5-B-derived epiHSP70s epichaperomics datasets and associated analyses can be found in Supplementary Data 1,2.

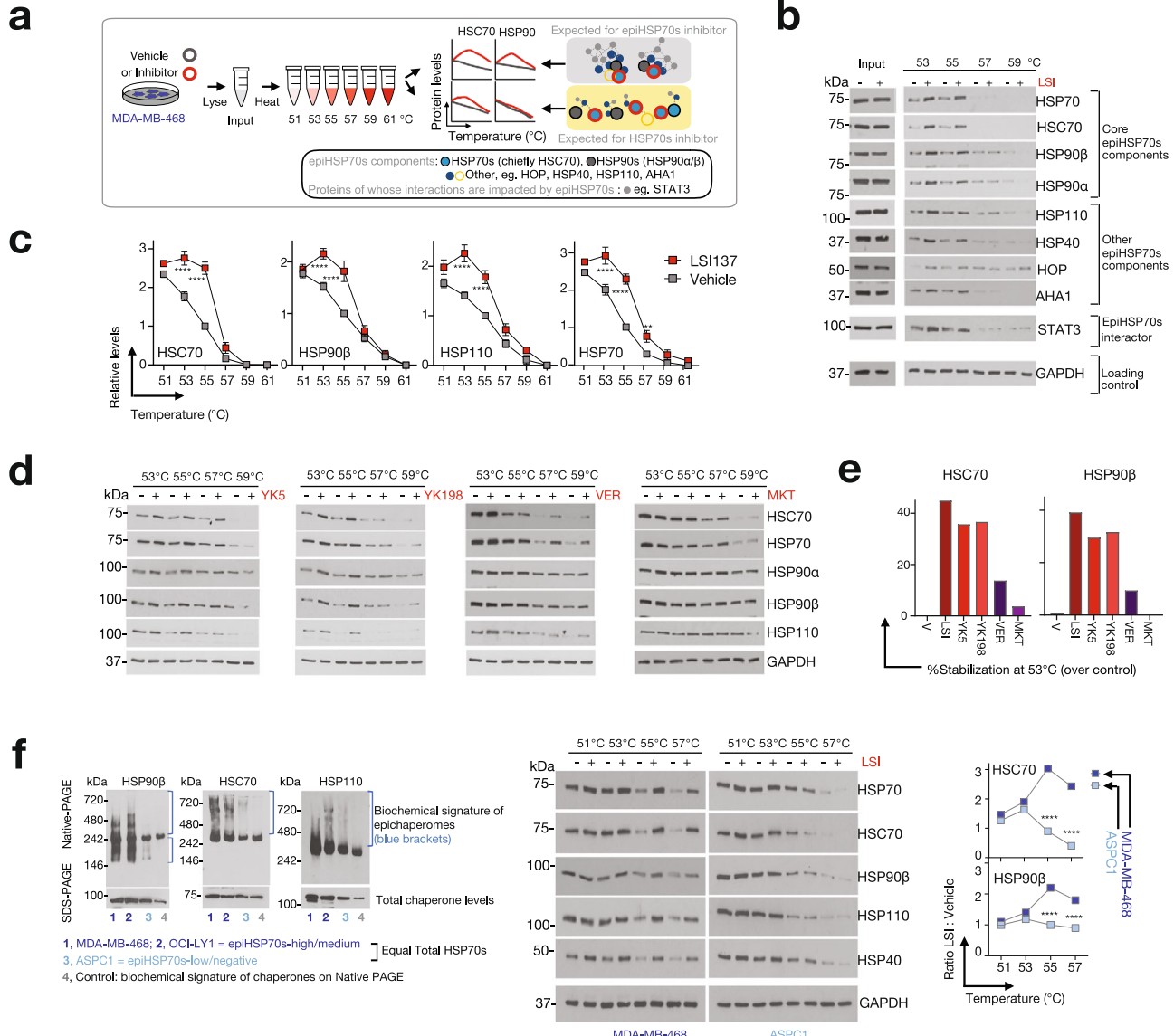

**Fig. 2 | Characterization of the epiHSP70s probes through the cellular thermal shift assay (CETSA). a** CETSA outline for experiments in (**b**–**e**). Homogenates from cells treated with vehicle (DMSO) or inhibitors (LSI137, 10 μM; YK5, 20 μM; YK198, 25 μM; VER155008, 50 μM; MKT077, 20 μM) were aliquoted and heated to a range of temperatures. Detection of changes in protein stability for epiHSP70s component chaperones and co-chaperones was performed by Western blot analysis. **b**, **c** Representative gels (**b**) and melting curves (**c**) for vehicle- and LSI137-treated cells. Data are presented as mean ± s.e.m., two-way ANOVA, vehicle (V), $n = 11$, LSI137 (LSI), $n = 5$, $p < 0.0001$, F (1, 84) = 103.3, 38.04, 61.46 and 88.14 for HSC70, HSP90β, HSP110 and HSP70, Sidak's post-hoc; $p < 0.0001$ for HSC70, HSP90β, HSP110 and HSP70 at 53 °C and 55 °C, LSI compared to V. Values normalized to those obtained for vehicle at 55 °C. See also Supplementary Fig. 15a, b. **d** Representative gels ($n = 3$ repeats) for cells treated as in (**a**) with vehicle-, YK5-, YK198-,

VER155008 (VER)- and MKT077 (MKT). **e** Treatment-specific stabilization of epiHSP70s components in cell extracts heated at 53 °C, for experiments as in (**a**). Graph, mean values normalized to those obtained for Vehicle at 55 °C.
**f** Comparative analysis of the stability of epiHSP70s component chaperones performed, as in (**a**), in epiHSP70s-high (MDA-MB-468) and epiHSP70s-low (ASPC1) cancer cells (see Native PAGE). Both cell lines have similar total levels of HSP70s and HSP90s (see SDS PAGE here, and see immunofluorescence further). Graph, mean ($n = 3$). Data are presented as mean ± s.e.m., two-way ANOVA, $n = 3$, $p < 0.0001$, F (1, 16) = 386.4 and 246.5 for HSC70 and HSP90, respectively. Sidak's post-hoc, $p < 0.0001$ for HSC70 and HSP90β at 55 °C and 57 °C, MDA-MB-468 compared to ASPC1. Source data, along with relevant statistical analyses and analysis data, are provided as a Source Data file.

We evaluated the quality of the interactors first by qualitative inspection of Coomassie blue stained gels of the YK5-B cargo in a side-by-side comparison with a traditional antibody purification (Fig. 5b). YK5-B outperformed the HSP70s antibody in the amount of captured cargo, and the capture capacity of YK5-B was at par with the validated epiHSP90s epichaperomics probe PU-beads[17,19,21] (Fig. 5b). Wash-off of the cargo with high salt containing buffer confirmed HSP70s as the sole direct interactors of the probe (Fig. 5c).

The YK5-B cargo was subsequently subjected to interactor identification and quantification by MS using label-free strategies. We

performed both spectral counting and ion intensity-based quantification analyses[39]. Detected proteins from the YK5-B cargo were ranked as Grade A ($p$-value < 0.1; 2481 proteins), Grade B ($0.1 \le p$-value $\le 0.25$; 1018 proteins) and Grade C ($0.25 < p$-value < 0.5; 204 proteins) (Fig. 5d) and subjected to further analyses.

We compared and contrasted cargo captured through YK5-B epichaperomics to protein expression levels, as detected previously by a proteomic-based approach[12]. Consistent with PPI changes driven consistently by protein-modifying mechanisms, including alterations in post-translational modifications and stabilization of disease-enriched

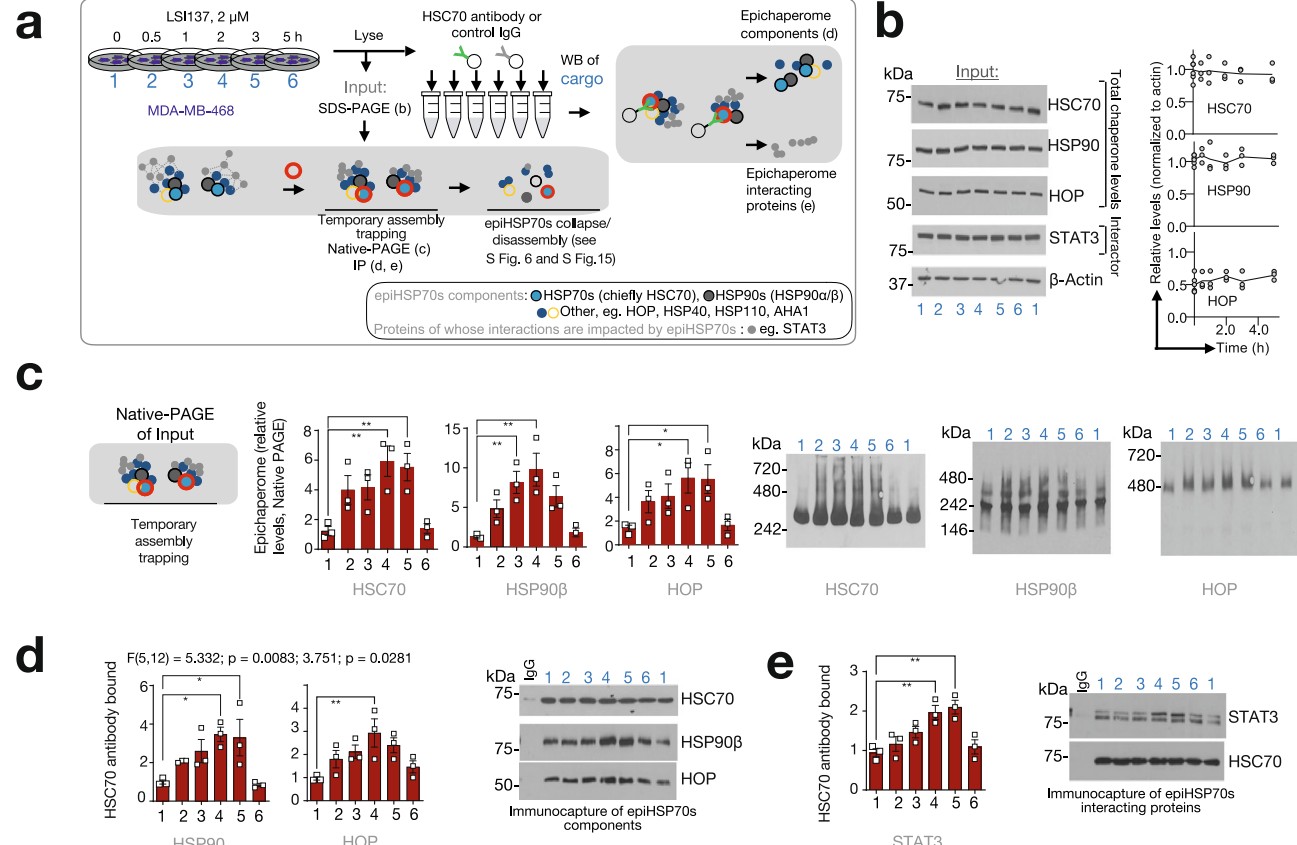

**Fig. 3 | Characterization of the epiHSP70s probes through immunoprecipitation techniques. a** Experiment outline. See also Supplementary Fig. 15c. **b**–**e** Monitoring by SDS-PAGE (**b**), Native-PAGE (**c**) or immunoprecipitation with an anti-HSC70 antibody (**d**, **e**) followed by immunoblotting, of epiHSP70s components (i.e. constituent chaperones and co-chaperones, **d**) and of epiHSP70s interactor proteins (i.e. STAT3, **e**) in cells treated as in (**a**). Data in (**b**), graph, mean of indicated individual biological replicates ($n = 6$, time 0 h and $n = 3$ for each other time point). Data in (**c**–**e**) are presented as mean ± s.e.m., $n = 3$ biological replicates, one-way ANOVA (**c**) HSC70: $p = 0.0039$, F (5, 12) = 6.465; HSP90β: $p = 0.0024$, F (5, 12) = 7.238; HOP: $p = 0.0273$, F (5, 12) = 3.787; (**d**) HSP90: $p = 0.0083$, F (5, 12) = 5.332; HOP: $p = 0.0281$, F (5, 12) = 3.751; and (**e**) STAT3: $p = 0.0011$, F (5, 12) = 8.693, with Dunnett's post-hoc. β-actin, protein loading control; IgG, isogenic control. Source data are provided as a Source Data file.

protein conformations, and only periodically influenced by changes in protein expression[14], we found YK5-B interactors encompassed both low and high abundance proteins (Supplementary Fig. 17a). Importantly, the extent by which proteins were incorporated into PPIs was independent of the overall expression of each protein within the specific cellular context of MDA-MB-468 cells (Supplementary Fig. 17a).

HSC70 co-exists with HSP90 in a subset of epichaperomes[17,19,21], and therefore it is expected that epiHSP70 interactors would be also identified by the epiHSP90 bait. We compared YK5-B interactors with those identified by PU-beads (as in Fig. 5b)[17,19,21] to find a significant number of overlapping epiHSP70 and epiHSP90 interactors in our dataset ($p = 0.031$; Supplementary Fig. 17b, c). The co-presence of HSP90s and HSP70s in a significant subset of epichaperomes is supported by experiments analysing the apoptotic sensitivity of 73 cancer cell lines encompassing 9 tumor types, where we find 66% exhibit similar vulnerability to both epiHSP70 (i.e., YK198) and epiHSP90 (i.e., PU-H71) agents (Fig. 6a, intracellular ATP level measurements). However, cell lines more sensitive to YK198 than to PU-H71 (and vice versa) were observed, positing HSP90-independent epiHSP70s (and HSP70-independent epiHSP90s) may exist in cancer cells, which remains to be investigated. Supportive of epiHSP70s formation being a prevalent mechanism in tumors, 68% of the examined 73 cell lines, irrespective of the tumor type, were sensitive to YK198, a percentage confirmed in patient-derived breast cancer explants (Fig. 6b–d). We found 5 out of 8 of these primary breast cancer specimens express medium to high epiHSP70s (Figs. 6c), and 8 out of 12 undergo substantial apoptosis

(>25% apoptosis over control, Fig. 6d) when treated for 24 h with epiHSP70s agents. Importantly, normal cells in the breast parenchyma (i.e., epichaperome negative) remained unaltered by epiHSP70s agents (Fig. 6e, f).

Next, we constructed PPI maps of the identified interactors to analyse their topology (Fig. 7a and Supplementary Fig. 18a). Analysis of topological parameters in a PPI network may inform on interactor quality and nature, as it provides insight into both the connection and the relationship between nodes (i.e. proteins) in a network[40]. We first investigated chaperone and co-chaperone components and found them to be strategically positioned around the core HSP70s chaperones in a layered topology. Chaperones and co-chaperones surrounded HSP70s and were in turn encircled by a constellation of scaffolding proteins and isomerases, a topology consistent with the reported function of epichaperomes as scaffolding platforms that rewire PPIs[2,18–20]. The layer of interacting proteins also presented a weighted, layered topology (Fig. 7a and Supplementary Fig. 18a). The layered character of the network was also evident from calculations of several topological parameters (Supplementary Fig. 18b). For example, reported HSP70s interactors tended to form a dense network centered on HSP70s, whereas by comparison epiHSP70s interactors expanded from around the dense inner core into outside layers, as reflected in the network centralization and the clustering coefficients. This is consistent with an ability of the YK5-B bait to select for epiHSP70s found in cells alongside abundant HSP70s, as well as enrich for and trap epiHSP70s-interactors, capturing both epiHSP70s direct and indirect interactors.

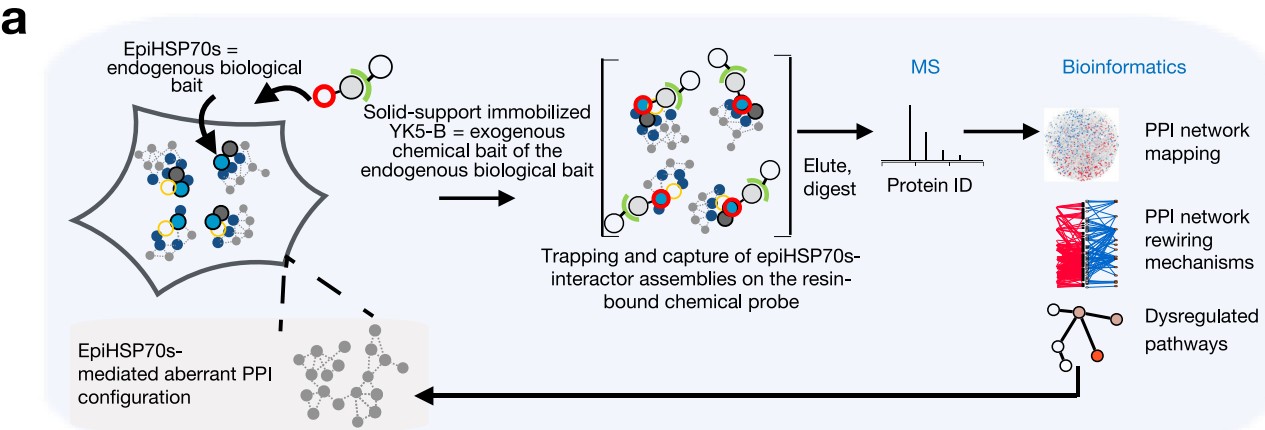

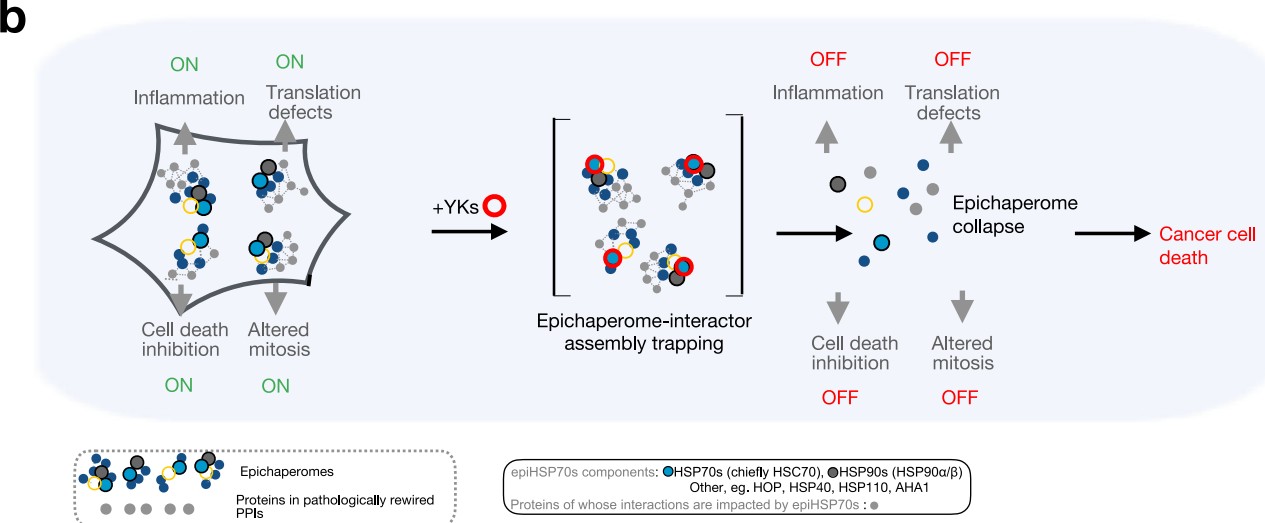

**Fig. 4 | Principles for the use of YKs in epichaperomics and as chemical probes to revert context-dependent epiHSP70s-mediated perturbations in PPI networks. a** In epichaperomics, a multitude of endogenous baits (i.e., the distinct epichaperome structures characteristic of a specific cellular context), each having individual interactors, are natively present. Chemical probes are needed to bind core, nucleating, epichaperome components (such as HSP70s) and trap these individual epiHSP70s bound to their interacting proteins, thus retaining interactions through subsequent isolation steps and enabling their unbiased identification by mass spectrometry (MS). These characteristics enable robust identification of interactors, increasing the likelihood to correctly assign the context-specific function of a given protein. By eliminating the need for exogenous introduction of a tagged protein as bait, epichaperomics is applicable for dissecting native cellular states, and thus appropriate for the analysis of both cultured cells and primary specimens. A bioinformatics pipeline was designed to: identify the proteins whose connectivity is pathologically altered via epiHSP70s in the specific biological context; construct the context-specific PPI network maps; derive biological insight on PPI network dysfunctions and the functional outcome of these PPI network dysfunctions. A decided advantage of the epichaperomics approach is that it investigates PPI network dysfunctions in disease. This directly informs on stressor-to-phenotype relationships and provides context-dependent insights into dysfunction, as opposed to standard proteomics which inform on changes in the levels of proteins. **b** Data support a biochemical mode of action with an initial trapping of the epichaperomes by YK, followed by a time dependent collapse of these assemblies, independent of the total expression of HSP70s. epiHSP70s disruption potentially reverts context-dependent perturbations in PPI networks, a process detrimental to cancer cell survival.

Trapping direct and indirect interactors (e.g., as part of protein assemblies) is an important feature as it may enable an accurate determination of the context-specific function of a specific protein despite a potentially 'noisy' dataset (i.e. with many contaminants). A single protein may carry out many different functions with different partners in different biological contexts. It is the number of context-dependent interactors, not the total number of proteins identified by a specific method, that is of critical importance for accurate mapping of dysregulated protein pathways underlying disease phenotypes (Supplementary Fig. 19a and ref. 41). To provide support for this postulate, we compared pathway enrichment analyses performed on proteins of Grade A, $p$-value < 0.1, vs Grade A + B, $p$-value < 0.25, vs Grade A + B + C, $p$-value < 0.5, to find more lenient statistical thresholds used for interactor identification did not skew biological interpretations (Fig. 7b). Thus, although epichaperomics is an affinity purification, and thus like any such methods will contain contaminants in the pull-downs (e.g., resin-sticky, abundant proteins, proteins containing moieties reactive towards acrylamide, and others, see Fig. 5d), in the case of epichaperomics 'true' interactors are endogenously selected and enriched through their interaction with epichaperomes. False interactors that are not correctly filtered, are unlikely to skew 'true' biological information. Epichaperomics studies therefore tolerate a certain degree of background signal, and we recommend selecting proteins of Grade A, $p$-value < 0.1, or A + B, $p$-value < 0.25, for pathway enrichment studies, which we proceed to do also below.

### Functional mapping of interactors

The interaction of a protein with different binding partners within different biological contexts is reflected in the functional output of interactions (i.e. a protein may be assigned to different protein pathways in distinct cellular contexts as dictate by the presence of potential interactors, Supplementary Fig. 19a–c and ref. 41). Pathway mapping of

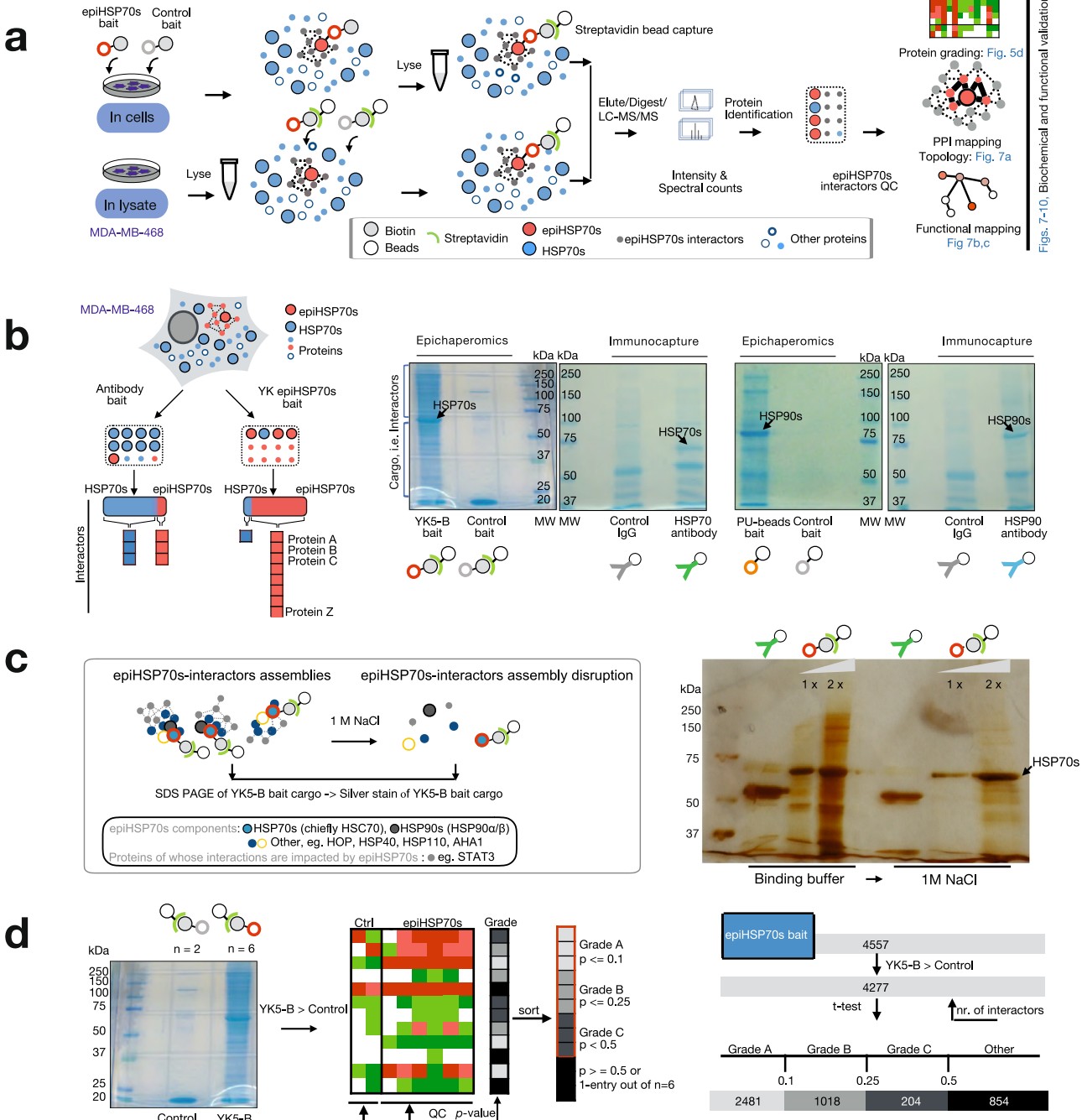

**Fig. 5 | Quality control of the interactors isolated by the epiHSP70s probe.**
**a** Experimental design to probe the utility and robustness of YK5-B in epicha-peromics analyses. **b** Theoretical basis showing improved interactor identification and coverage for epichaperomics baits when compared to conventional affinity purification baits that do not differentiate between chaperones and epichaper-omes. Coomassie blue stained SDS-PAGE of the eluted interactors captured by traditional antibody purification vs epichaperomics is shown on the right. MW, molecular weight marker; IgG, isogenic control. **c** Silver-stained gel of the protein cargo captured by YK5-B beads (at 1x or 2x YK5-B load). Cargo was washed with either binding buffer or high-salt buffer before being subjected to SDS-PAGE, as indicated in the schematic. The cargo captured by an HSP70s antibody is shown for reference. Gel images (**b**, **c**) are representative of three independent experiments. **d** Schematic showing the grading system implemented to stratify proteins identi-fied by YK5-B capture. Source data are provided as Supplementary Data files.

epiHSP70s interactors in MDA-MB-468 cells is therefore expected to inform not only on the identity of proteins that become altered in this cellular context, but also, more importantly, inform on the functional impact of such alterations.

Reactome pathway enrichment analysis of the epiHSP70 inter-actors indicated that proteins impacted by epiHSP70s map to key biological processes known to be altered in cancer: mitotic cell cycle, programmed cell death, p53-regulated transcription, DNA-repair,

translation, transport, glucose metabolism, signaling as well as immune regulatory pathways (e.g. Interferon signaling, Class I MHC antigen processing) ($p$-value adjusted FDR < 0.05, Fig. 7c, Supple-mentary Fig. 19b). Focusing on signaling pathways, our analyses found that proteins impacted by epiHSP70s in MDA-MB-468 cancer cells would result in activation of constitutive EGFR signaling, activated AKT and PTEN stabilization, oncogenic MAPK signaling, non-canonical NFκB signaling, constitutive NOTCH1 signaling, RhoGTPase activation

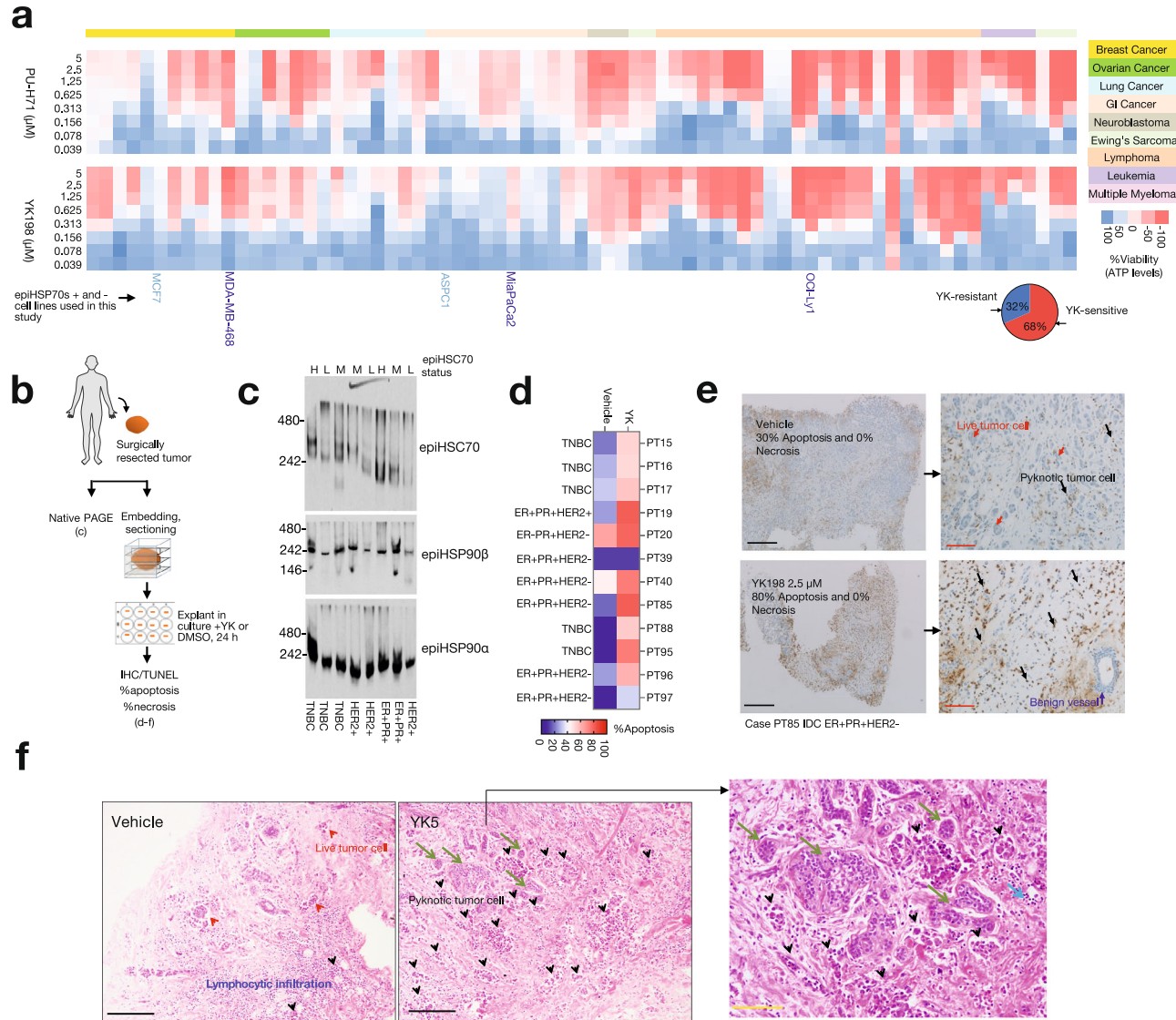

**Fig. 6 | Biological activity on YK-agents in a panel of cancer cells and in primary breast cancer explants in culture. a** Sensitivity of cancer cells (measured by ATP levels) to epiHSP70s (YK198) and epiHSP90s (PU-H71) disrupters (72 h treatment). 73 cancer cell lines encompassing the indicated 9 tumor types are presented. Data are mean of biological replicates ($n = 3$). Pie chart, % of cell lines sensitive or resistant to YK198. **b** Experimental design to probe epichaperome levels and YK-agent (YK5 at 5 μM or YK198 at 2.5 μM) sensitivity of individual breast cancer (BC) specimens. **c** As in (**b**) for epichaperome levels. H high, M medium, L low. The gel image shows representative primary tumors of individual BC patients ($n = 8$). **d** As in (**b**) for sensitivity. DMSO, vehicle control. Scale bar, %cell death. PT primary tumor. TNBC, triple-negative; HER2 + , HER2-positive; ER + PR + , estrogen receptor and progesteron receptor-positive BCs. **e**, **f** Representative cases as in (**b**) that contain BC cells along benign cells (i.e. lacking epichaperomes). TUNEL (**e**) and H&E (**f**) stained specimens. In (**e**), the control section of PT85 treated with vehicle ($n = 2$

explants) shows viable invasive ductal carcinoma (IDC) cells (red arrows) and apoptotic (~30%) IDC cells (black arrows). Following treatment with YK198 ($n = 2$ explants), there is a marked increase of apoptosis ( ~ 80%). A benign vessel (dark blue arrow) remains unaltered. In (**f**), the control section of PT19 treated with vehicle (left panel, $n = 5$ explants) displays viable IDC cells (red arrow heads) and apoptotic (~30%) IDC cells (black arrows heads) associated with a benign host lymphocytic infiltration. Following treatment with YK5 ($n = 5$ explants), there is a marked increase of apoptosis (~95%) (middle panel). Benign ducts (green arrows) and the surrounding host lymphocytes (light blue arrow) within the same section, remain unaltered and are surrounded by apoptotic IDC cells (inset, right panel). Arrows and arrowheads show representative cells. Scale bar, black, 500 μm; yellow, 200 μm; red, 100 μm. Source data along with biospecimen characteristics and sample availability are provided as Supplementary Data files.

of formins, β-catenin defective regulation, toll-like receptor cascades and aberrant death receptor signaling. These identified dysfunctions in signaling pathways map to those reported for the MDA-MB-468 cancer cell line through independent techniques[12,42–46], supporting overall that YK5-B isolated interactors are to be of high fidelity, and that the probe is compatible with the rigor of epichaperomics studies.

Additional validity of the identified YK5-B interactors comes from the comparison of global changes induced in protein pathways by epiHSP70s inhibition (by either YK treatment or epiHSP70s knock-down) with those detected directly by YK5-B epichaperomics.

We found protein pathways perturbed upon epiHSP70s inhibition are commensurate with those detected in the analysis of YK5-B interactors (compare mapping of epiHSP70s interactors, Fig. 7c with mapping of proteins changed upon either YKs treatment or siRNA knock-down, Fig. 7d). Datasets and associated analyses can be found in Supplementary Data 3. In addition to these unbiased MS analyses, we validated select pathways by western blot analysis (e.g. with antibodies directed against specific signaling, mitotic cell cycle, translation pathways, and cell death proteins) (Fig. 7e and see below).

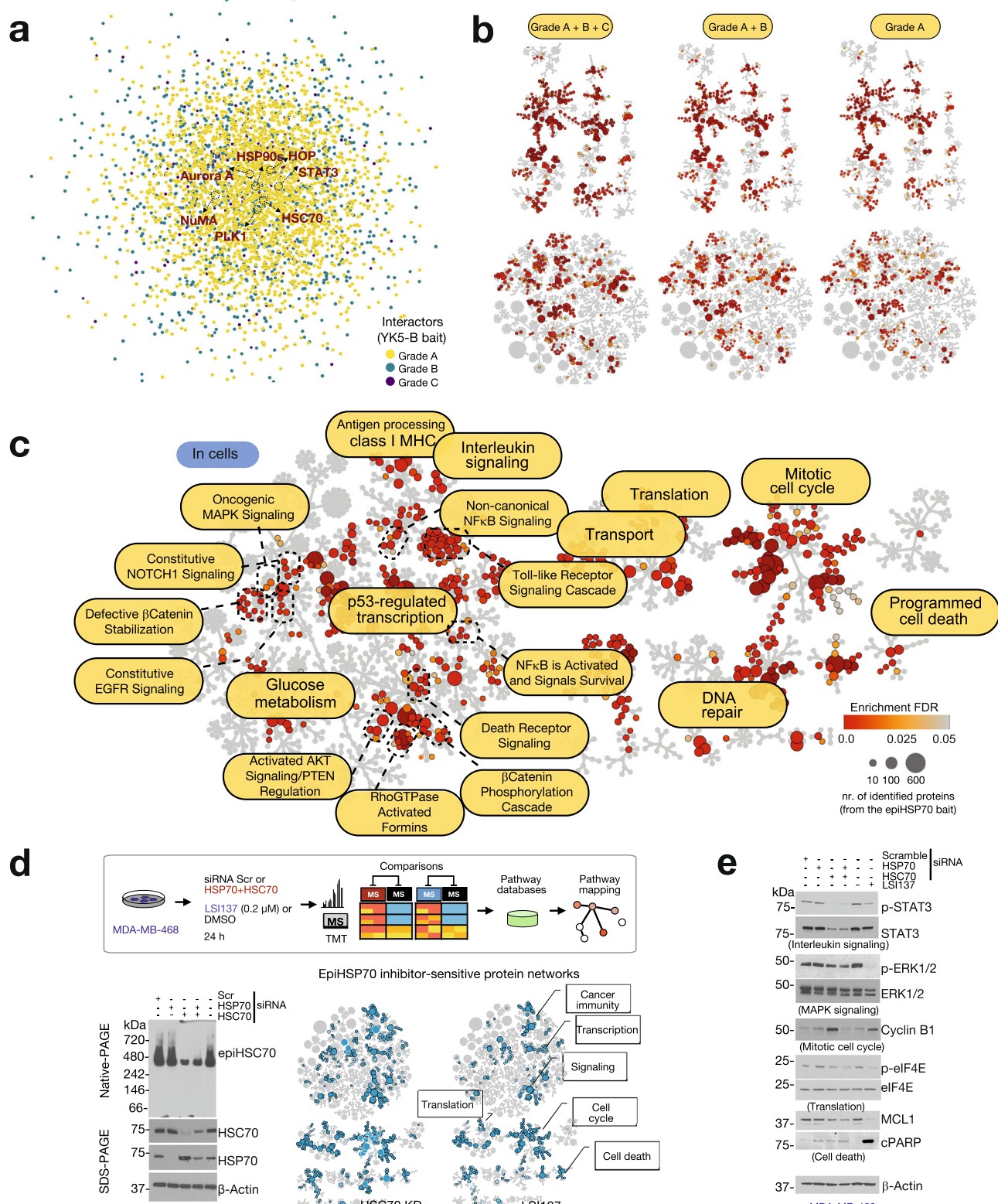

**Fig. 7 | Functional mapping of interactors identified by the epiHSP70s bait.**
**a**, **b** PPI mapping (**a**) and functional annotation (**b**, Reactome pathway mapping) of the interactors identified by the YK5-B probe in MDA-MB-468 cells. Proteins graded as in Fig. 5d are shown. The location of select epiHSP70s components (HSP70s, HSP90s and HOP) and interactors (STAT3, NuMA, PLK1, Aurora A) is also shown in (**a**). **c** Reactome pathway enrichment analysis detailing the functional annotation of Grade A proteins mapped as in (**a**). Probe was added to live cells for capture, see

Fig. 5a. **d** Proteome-wide functional changes (analysed by Reactome pathway mapping) induced by pharmacologic or genetic epiHSP70 inhibition in MDA-MB-468 cells, as indicated. **e** Validation of select proteins and protein pathways identified in panels (**c**, **d**). See Figs. 8–10 for detailed validation with focus on Cell cycle. Gel images are representative of three independent experiments. Source data are provided as Source data and Supplementary Data files.

In sum, several layers of analyses, from unbiased systems level to targeted hypothesis-driven biochemical and functional analyses, support the use of YK5-B in epichaperomics investigations. The YK5-B epichaperomics data indicate the connectivity of thousands of proteins are affected through epiHSP70s formation. These proteins map onto pathways that become altered in the specific cancer cell context, and are key to maintaining malignant phenotypes, supporting the use of epiHSP70s interactor datasets to inform on cell-specific interactomes, i.e. the intrinsic PPI networks perturbed in the specific cancer cells context. Importantly, YKs, which we find work through a mechanism of action consistent with an initial trapping of epiHSP70s along with interacting proteins followed by their disassembly and inactivation, may be useful tools to study, and potentially revert, context-dependent perturbations in PPI networks in the context of epiHSP70 positive cells. Both these postulates we address below, by focusing on select protein pathway perturbations in the context of epiHSP70s-positive cancer cells, with the goal to derive proof-of-principle mechanistic insights from YK5-B based epichaperomics.

## epiHSP70s are context-dependent mitotic regulators

Reactome analyses identified mitotic processes, especially those important in the stages of transition from $G_2/M$ throughout anaphase, among the top enriched in MDA-MB-468 cells, suggesting an important biological importance for epiHSP70s in mitosis (Fig. 7c and Supplementary Fig. 20). Conversely, although there are reports linking HSP70s to a number of mitotic processes[25,47–52], HSP70s are not considered key mitotic proteins in human cancer (https://bioplex.hms.harvard.edu/; HSPA8 and HSPA1A, the constitutive and the inducible isoforms of HSP70, also referred to as HSC70 and HSP70, respectively, have no significant GO term related to cell division/mitosis).

Immunofluorescence using epichaperome probes (Supplementary Fig. 21) or an HSC70 antibody that recognizes both HSP70s and epiHSP70s, confirmed a strong localization of epichaperomes at mitotic structures, including the spindle pole (Fig. 8a, b and Supplementary Fig. 22a). epiHSP70s levels, but not the levels of chaperones, increased significantly as cancer cells transitioned through the $G_2$ and M phases, as evidenced by immunofluorescence, as well as by Native- and SDS-PAGE analyses (Fig. 8c, d and Supplementary Fig. 22b).

To experimentally confirm dependence of mitosis on epiHSP70, and to demonstrate the context-dependence of this mechanism in epiHSP70s positive but not in epiHSP70s negative cancer cells, we analysed the functional impact of epiHSP70 inhibitors in cells with equal HSP70s levels but differentiated by their epiHSP70s content. Specifically, we verified mechanistic results from unbiased large-scale analyses obtained in epiHSP70s positive MDA-MB-468 breast cancer in a panel of epichaperome positive cancer cells (OCI-LY1, lymphoma; MiaPaCa2, pancreatic cancer; and HeLa, cervical cancer), and showed the mechanism does not apply to epichaperome-low/negative cells (e.g., ASPC1, pancreatic cancer; MCF7, breast cancer and non-transformed but proliferating, CCD-18 fibroblasts). EpiHSP70s inhibition blocked cells in mitosis, with a significantly higher impact in epiHSP70s-high cancer cells when compared to epiHSP70s-low/negative but HSP70s-high cancer cells or epiHSP70s-negative non-transformed cells, as evidenced by the $G_2/M$ population (evaluated by propidium iodide staining), cyclin B1 levels (evaluated by western blot) and the number of mitotic cells (measured by phospho-MPM2 antibody staining) (Fig. 8e, f and Supplementary Figs. 23a–c, 24a, b). epiHSP70s knock-down, but not inhibition of other HSP70s pools, recapitulated the effect of YKs in mitosis (Supplementary Fig. 25a–d). Also supportive of the context-dependence of mitotic processes on epiHSP70s, cells released from thymidine block (i.e. synchronized in $G_1/S$ and released to progress through the cell cycle) were more sensitive to epiHSP70s inhibition than cells kept in thymidine (i.e. blocked in $G_1/S$), as measured by live cell microscopy monitoring and annexin V staining. This effect was significantly more acute in epiHSP70s-high/

medium cancer cells compared to epiHSP70s-low/negative but HSP70s-high cancer cells (Supplementary Fig. 26a, b). In aggregate, these experiments both functionally validate the specificity of YKs for epiHSP70s over HSP70 and provide support for epiHSP70s emerging as a mitosis regulatory mechanism in epiHSP70s positive cancer cells.

Examining the morphology of mitotic cells, we observed epiHSP70s-positive cells that attempted to progress from $G_2$ through mitosis with pharmacologically or genetically inactivated epiHSP70s, exhibited aberrant, fragmented spindles, a diffuse mitotic plate, misaligned chromosomes, with no proper metaphase plate observed (Fig. 8g and Supplementary Fig. 27). These morphological changes directly support the outcome of YK5-B epichaperomics proposing that functionally, epiHSP70s interacting mitotic proteins map to processes important to the transition from $G_2$ throughout mitosis ((i.e. Mitotic Prometaphase (p.adjust = 7.46E-45); Separation of Sister Chromatids (p.adjust = 3.00E-40); Mitotic Metaphase and Anaphase (p.adjust = 3.10E-38); $G_2/M$ Transition (p.adjust = 6.18E-34); Mitotic Spindle Checkpoint (p.adjust = 2.27E-27); Fig. 9a and Supplementary Fig. 27)). Also confirming the key role of epiHSP70 in mitosis, none of the epiHSP70s-positive cancer cells that entered mitosis with an inhibited epiHSP70s remained alive (see cell blebbing following inability to progress through mitosis and time-dependent analysis of mitotic and apoptotic cells, Fig. 8g).

To further confirm dependence of mitosis in epiHSP70s positive cells on epiHSP70, as opposed to HSP70s, we also explored mitotic proteins and processes and analysed, both biochemically and functionally, their sensitivity to epiHSP70 inhibitors in cells with equal HSP70s levels but differentiated by their epiHSP70s content. Many proteins and protein complexes, each acting at specific stages, regulate mitotic progression[53,54], a number of which we identify to be potential epiHSP70s interactors. Among these are proteins important for mitotic spindle, spindle-assembly checkpoint signaling, centrosome regulation, establishment of correct kinetochore-microtubule structures such as Aurora kinase A (AURKA), Polo Like Kinase 1 (PLK1), Cyclin-dependent kinases (e.g. CDK2), Nuclear Mitotic Apparatus Protein 1 (NUMA1 or NuMA), Augmins (also called HAUS proteins), cytoplasmic linker protein (CLIP)–associating proteins (CLASPs), motor proteins which generate forces during mitosis, such as kinesins (e.g. KIF2C, KIF11 also called Eg5), mitotic checkpoint proteins (e.g. BUB3), and proteins important for chromosome condensation (e.g. NCAPD2)[55–58] (Fig. 9a–c and Supplementary Fig. 28). Integration of these mitotic proteins into aberrant PPIs in the context of MDA-MB-468 cells was independent of overall cellular expression (see Fig. 9c, comparison of proteomics to epichaperomics).

To exclude a direct binding, and in turn effect, of YKs on these proteins, and because several are kinases, we screened YK198 (10 µM, i.e. 20-fold higher than its $IC_{50}$ on epiHSP70s, see Supplementary Fig. 6) in the scanEDGE KINOMEscan. This screen contains 97 kinases, including many cell cycle regulators such as PLK1, AURKA and CDKs, among others; none was significantly inhibited by YK198 (Supplementary Fig. 29). To verify these proteins are indeed epiHSP70s, and not HSP70s dependent, we compared the stability and the epiHSP70s-dependence of select mitotic regulators in epiHSP70s-low and epiHSP70s-high cells with comparable HSP70s levels. When normalized to their cellular expression, we found specific mitotic regulators (e.g. NUMA1, AURKA, and PLK1) to be thermally more stable in epiHSP70s-high cells compared to epiHSP70s-low cells (Supplementary Fig. 30a), and to be thermally more sensitive to the addition of YKs in epiHSP70s-high cells than in epiHSP70s-low cells (see CETSA, Fig. 9d). We found significantly less mitotic regulators in the epiHSP70 bait pull-downs from epiHSP70s-low cells than from epiHSP70s-high cells (Supplementary Fig. 30b). The epiHSP70s bait captured significantly less mitotic regulators in cells pre-treated with the epichaperome agent PU-H71, in accord with the co-expression of HSP70s and HSP90s in a subset of epichaperomes (Supplementary Fig. 30c). Moreover, beads which have attached a YK-derivative that is less

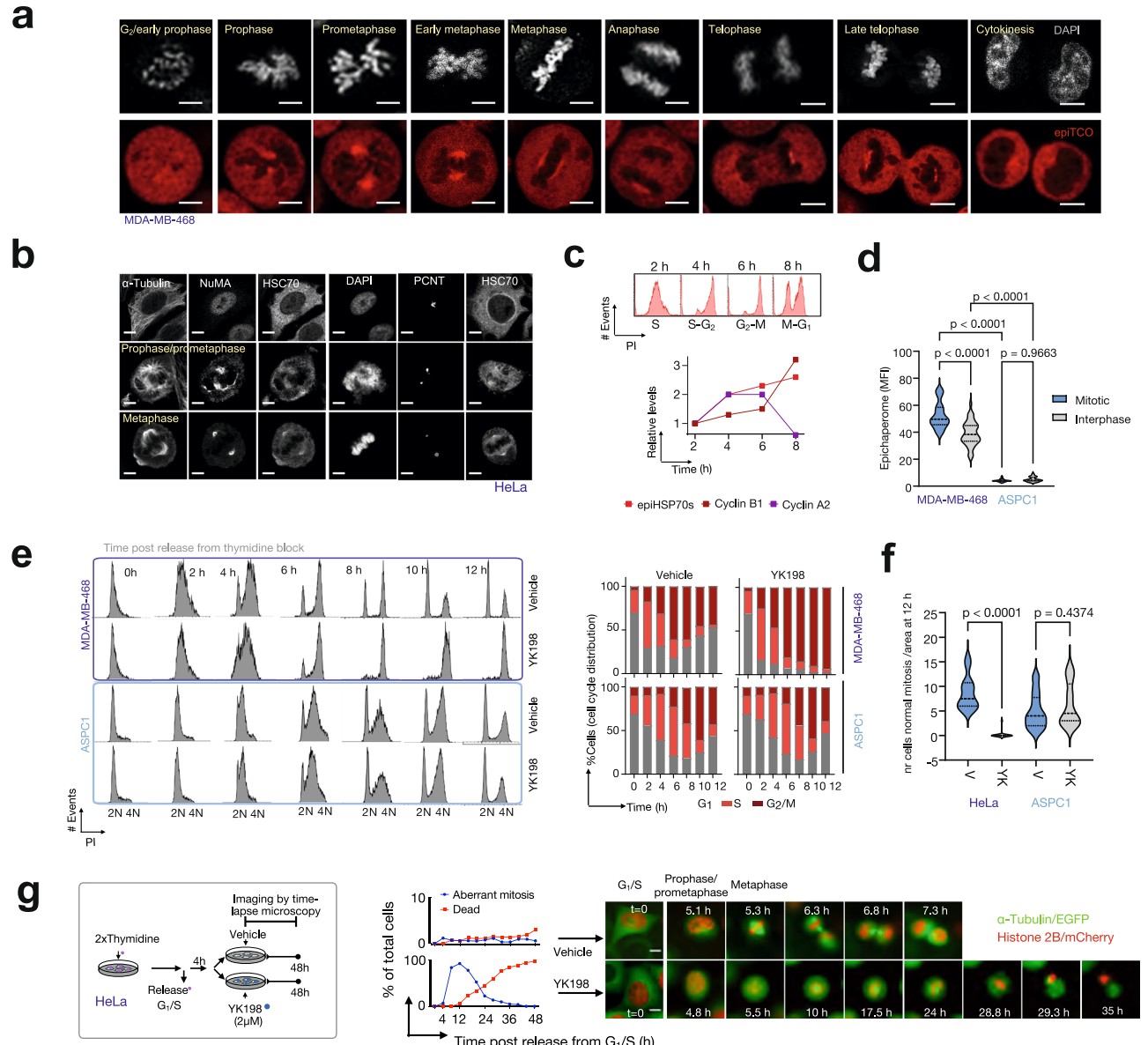

**Fig. 8 | EpiHSP70s are key context-dependent regulators of mitosis in epiHSP70s-positive cancers. a–d** Localization and expression of epichaperomes in interphase and mitotic cells monitored by IF (**a**, **b**, **d**) and Native-PAGE (**c**). Micrographs are representative of $n = 30$ mitotic cells. epiTCO, epichaperome detection reagent; DAPI, chromosomes stain. Mitotic proteins and structures are shown for reference: NuMA (at spindle pole in mitotic cells and in the nucleus in interphase cells), pericentrin (PCNT, centrosome), α-tubulin (spindle). Scale bar, 5 μm. Graph (**c**), mean of $n = 3$ experiments. Graphs (**d**), median, dotted line and quartiles, dashed lines, one-way ANOVA, $n = 40$ cells, $p < 0.0001$, $F (3, 156) = 537.7$, with Sidak's post-hoc. See also Supplementary Fig. 22. **e**, **f** Cell cycle (**e**) and confocal microscopy (**f**) analyses of epiHSP70s-high/medium (MDA-MB-468 and HeLa) and -low/negative (ASPC1) cells released from thymidine block into Vehicle or YK198 (2 μM). Graphs: (**e**) mean, $n = 3$ and (**f**) one-way ANOVA; $n = 20$ captured areas, $p < 0.0001$, $F (3, 76) = 23.82$ with Sidak's post-hoc. See also Supplementary Figs. 23–26. **g** Graphs (mean) and micrographs (representative time-lapse microscopy images) of cancer cells released from thymidine block into Vehicle ($n = 85$ cells) or YK198 ($n = 103$ cells). Micrographs shows representative cells as they enter mitosis, fail to establish a proper mitotic plate, and undergo apoptosis. Scale bar, 25 μm. See also Supplementary Fig. 27. Source data, along with relevant statistical analyses and analysis data, are provided as a Source Data file.

potent than YK5, captured significantly less mitotic regulators than the YK5-B beads (Supplementary Fig. 30b, d). Immunoprecipitation confirmed an association of representative mitotic proteins with epiHSP70s, primarily in mitotic cells (Supplementary Fig. 30e), whereby this interaction was significantly altered by epiHSP70s inhibition (Supplementary Fig. 30f, g and see further). These multiple lines of evidence confirm that the identified mitotic proteins are specific epiHSP70s interactors, and thus involved in the rewiring of mitotic PPI networks in the context of tested epiHSP70s-positive cancer cells.

We next went to functionally confirm that mitotic regulator proteins are epiHSP70s dependent, specifically by evaluating the reliance of mitotic processes these proteins participate in, on epiHSP70s. We

selected 'Recruitment of NuMA to mitotic centrosomes' (p.adjust = 3.28E-20) (Fig. 9a, Grade A proteins) for proof-of-principle investigation. NuMA is present in the nucleus during interphase[55,57–65]. However, upon nuclear envelope breakdown in mitosis, NuMA plays a role in the formation and maintenance of the spindle poles and the alignment and the segregation of chromosomes during mitotic cell division; in mitotic microtubule aster assembly and in tethering the minus ends of microtubules at the spindle poles, which is critical for the establishment and maintenance of the spindle poles[59,66–70]. In metaphase, NuMA regulates the recruitment and anchorage of the dynein-dynactin complex in the mitotic cell cortex regions situated above the two spindle poles, and hence regulates the correct orientation of the

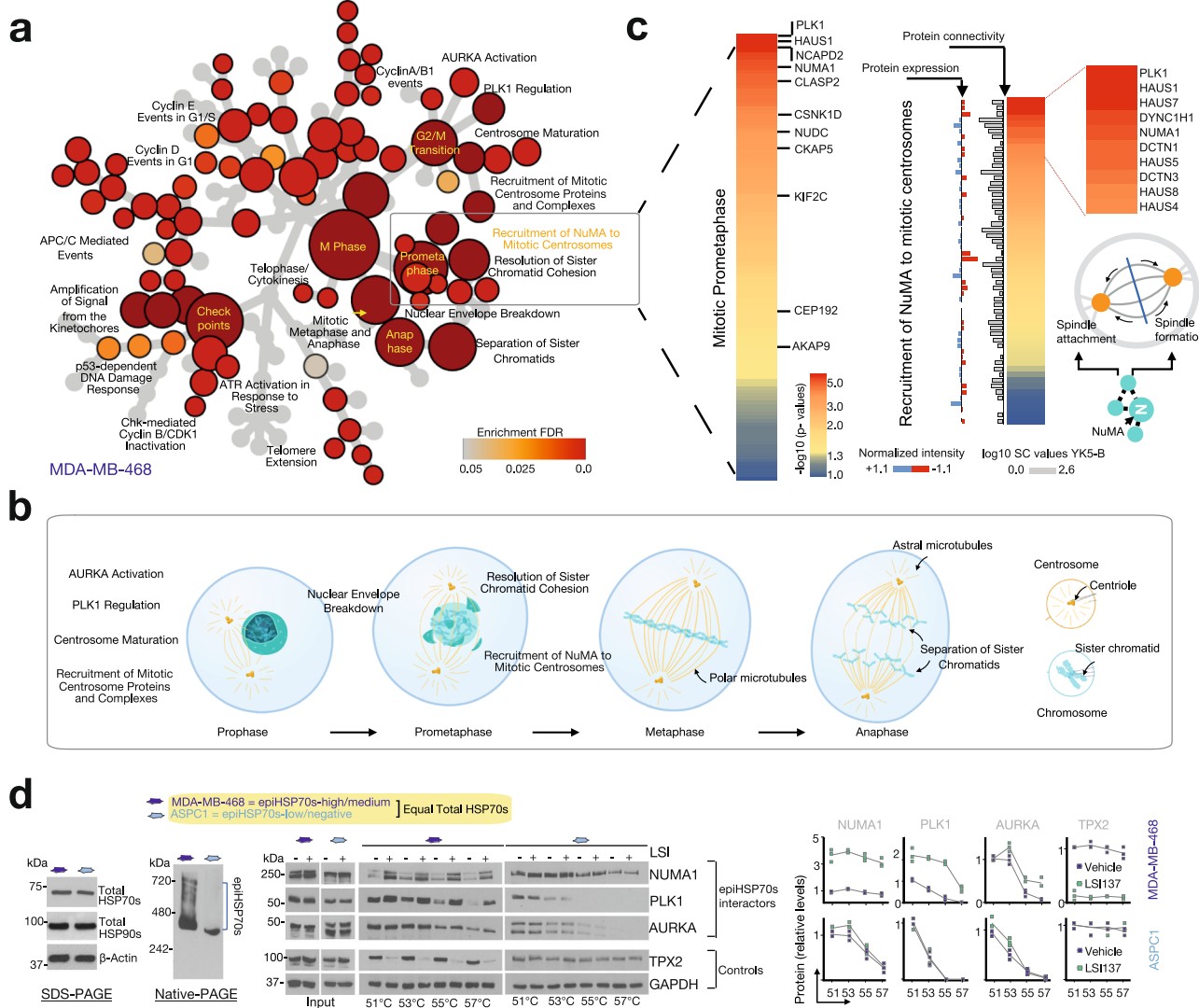

**Fig. 9 | epiHSP70s are context-dependent regulators of mitotic proteins in cancer. a** Reactome mapping of cell cycle-related protein pathways identified by the epiHSP70s epichaperomics in the epiHSP70s-high MDA-MB-468 cancer cells. **b** Stages of the mitotic cell cycle progression where most proteins, and in turn protein pathways, were identified to be dysregulated through epiHSP70s formation. Adapted from https://en.wikipedia.org/wiki/Mitosis. **c** Heatmaps show epiHSP70s interactors, as identified by the YK5-B bait, that were mapped to the indicated pathways. Representative proteins are annotated on the maps. For fully annotated maps see Supplementary Fig. 28. Proteins are graphed based on their interactor grade value (i.e. *p*-values calculated as in Fig. 5d). Scale bar, negative log10 (*p*-value). Changes in protein connectivity as detected by YK5-B epichaperomics compared to protein levels as determined by quantitative proteomics[12], are also shown. Gray bar, log10 spectral counts (SC) values, YK5-B interactors. Blue/

red bars, normalized intensity values as per ref. 12. Schematic: proteins involved in Recruitment of NuMA to mitotic centrosomes act in both assembling and pulling the mitotic spindle and in attaching it to the cell cortex. **d** Select mitotic proteins identified as in (**c**) were confirmed to be biochemically epiHSP70s-dependent by CETSA (**d**) and chemical bait-precipitation (see Supplementary Fig. 30) followed by western blot analysis. Melting curves for vehicle- and LSI137-treated (20 μM, 1.5 h) epiHSP70s-high (MDA-MB-468) and -low (ASPC1) cancer cells, with equivalent HSP70s levels, are shown. Representative gels and graphed data (*n* = 3 individual data points are shown). Values normalized to those obtained for Vehicle at 51 °C. Abbreviations: NUMA1, Nuclear Mitotic Apparatus Protein 1; PLK1, Polo Like Kinase 1; AURKA, Aurora kinase A; TPX2, TPX2 microtubule nucleation factor. Source data are provided as Source data and Supplementary Data files.

---

mitotic spindle[55,71,72]. During anaphase, NuMA mediates recruitment and accumulation of the dynein-dynactin complex at the cell membrane of the polar cortical region[55,59,72,73]. Activity of NuMA in these processes is regulated by both biochemical and functional interactions with specific proteins, including dynein, dynactin, PLK1, Eg5, among others[61–65,68,70,74].

We found that most cancer cells released from the G₂/M border can efficiently proceed to metaphase, forming a normal spindle with NuMA concentrated at the poles, where centrosomes are located (Figs. 8b and 10a, b). In contrast, in the presence of the epiHSP70s inhibitor, no cell was able to transit from the G₂/M border to metaphase (Fig. 10a). Specifically, we observed that while centrosomes

appeared intact from 40 to 80 min after exiting the G₂/M border (see pericentrin staining), a proper spindle could not be formed. Instead, random microtubule asters or small spindle-like structures were present in cells, and none of these mini spindle-like structures, while still largely associated with NuMA, were able to form an intact bipolar spindle with centrosomes tethered at the poles (Fig. 10a, see spindle association with centrosomes and NuMA). In metaphase cells, loss of NuMA at the spindle poles and centrosomes in epiHSP70s inhibited cells, accompanied a shortening in the spindle length and an increase in the distance between the cell cortex and the spindle poles (Fig. 10b), indicative of pole detachment. NuMA loss at the spindle pole could not be solely attributed to spindle disorganization−metaphase cells

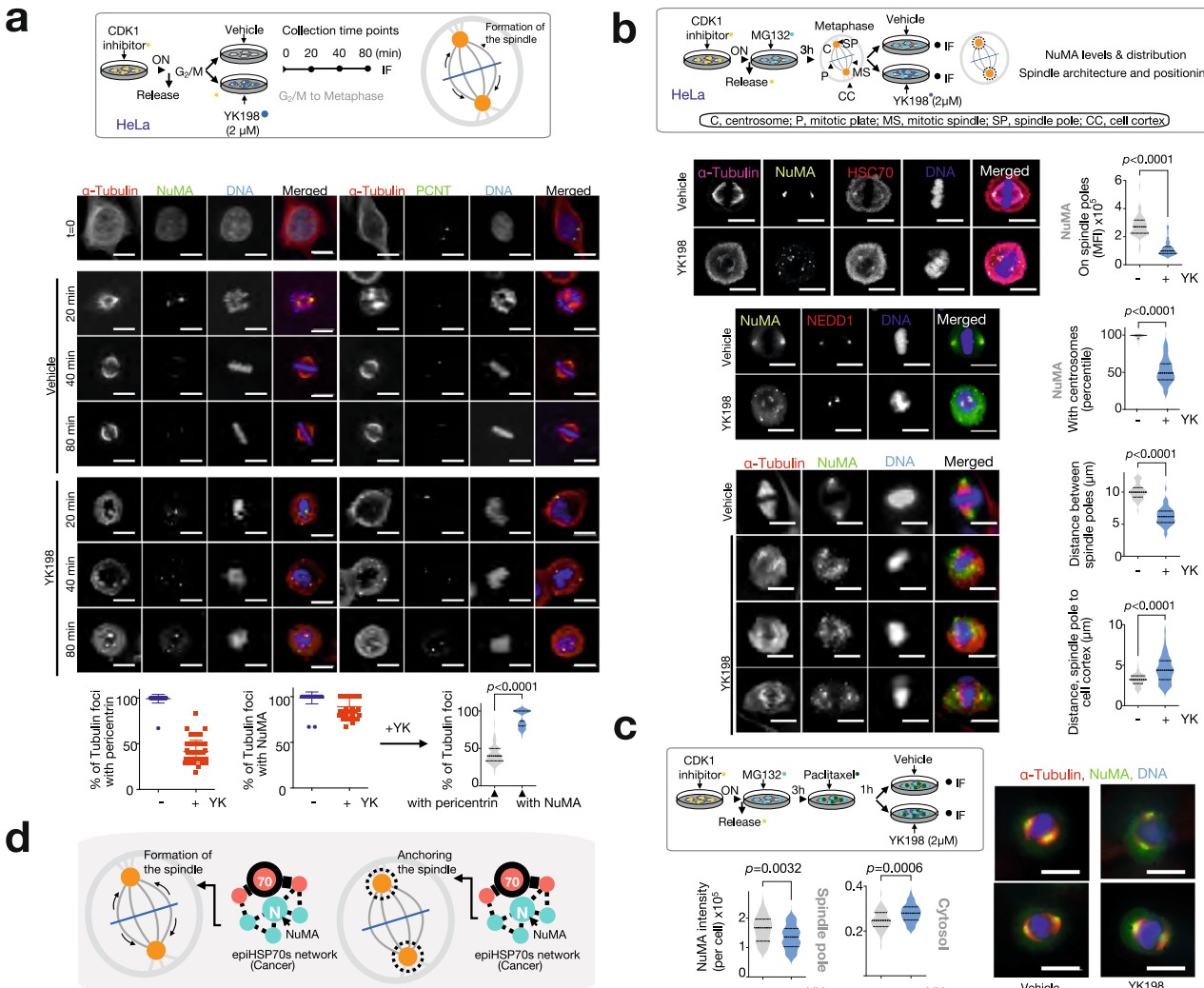

**Fig. 10 | epiHSP70s impact mitotic spindle in part, via regulation of NuMA activities related to spindle assembly, pulling and anchoring. a** Cells synchronized with a CDK1 inhibitor at the G2/M phase border were released into vehicle or YK198, as indicated. Tubulin foci associated with NuMA or pericentrin (centrosomes) were analysed by IF. Graphs, individual cells (*n* = 50 cells each for vehicle and YK198 treated cells, 2 independent experiments, mean±s.e.m.) and violin plots for YK198 treated cells (median, dotted line and quartiles, dashed lines, unpaired two-tailed Mann-Whitney test). **b** Cells synchronized with a CDK1 inhibitor and arrested at metaphase by the addition of the proteasome inhibitor MG132, were released into vehicle or YK198, as indicated. NuMA on the spindle pole or centrosomes, and the distance between spindle poles and from spindle pole to cell membrane were analysed by IF. Graphs, violin plots for Vehicle and YK198 treated cells (median, dotted line and quartiles, dashed lines, unpaired two-tailed Mann–

Whitney test, *n* = 50 individual cells each, 2 independent experiments). **c** Same as in (**b**) with microtubules stabilized by paclitaxel, as indicated. NuMA intensity on the spindle pole and in the cytosol is graphed. Graphs, violin plots for Vehicle and YK198 treated cells (median, dotted line and quartiles, dashed lines, unpaired two-tailed Mann-Whitney test, *n* = 50 individual cells each, 2 independent experiments). Representative IF micrographs are shown for (**a**–**c**). Scale bar, 10 μm for all images. **d** Schematic of the findings showing how NuMA is functionally dependent on epiHSP70s in mitotic cells. NuMA, together with its protein partners and regulatory proteins, is required for the proper assembly and maintenance of the mitotic spindle as well as spindle orientation and elongation. These NuMA-associated functions are impaired when epiHSP70s are inhibited. Source data, along with relevant statistical analyses and analysis data, are provided as a Source Data file.

treated prior to epiHSP70 inhibition with paclitaxel, a microtubule interactor that stabilizes the spindle, retained a bipolar spindle, yet statistically less NuMA was found on the spindle while more NuMA was found in the cytosol (Fig. 10c). NuMA loss from mitotic structures therefore reflects a direct loss of function due to epiHSP70s inhibition.

Collectively, these aggregated biochemical and functional lines of evidence support the hypothesis that epiHSP70s have emerged as a mechanism of adaptation of mitosis in the context of epiHSP70 positive cells. Rewiring of mitotic PPI networks through epiHSP70s is critical for the proper division of epiHSP70 expressing cells. Importantly, NuMA is one key protein identified among hundreds of mitotic proteins whose interactions and function we identify herein are directly or indirectly altered in cancer through epiHSP70s formation. Key processes including localization of NuMA in mitotic cells, forming and

tethering of mitotic spindles onto centrosomes and to the cell cortex by NuMA (alone or as part of mitotic complexes with regulatory proteins), are dependent on epiHSP70 formation (Fig. 10d). We demonstrate unequivocally that these functions cannot effectively occur in the absence of epiHSP70s activity in the specific context of epiHSP70s positive cancer cells, providing a mechanism for stressor adaptation of select cancer cells.

## Discussion

There is high interest in developing reagents enabling hypothesis-driven investigations to address context-dependent disease-relevant problems[75], and this manuscript aims to address this knowledge gap by introducing key reagents and methods while also providing proof-of-principle on how they can be used to derive systems-level context-

specific biology. Towards accomplishing this larger goal, we report on the YK-type chemical probes for which we provide several layers of evidence, from unbiased systems-level to targeted hypothesis-driven biochemical and functional, on their applicability to epichaperomics investigations.

Using the YK-type epiHSP70s probes, we provide an experimental roadmap for large-scale mechanistic investigations of context-specific PPI network dysfunctions through epichaperomics. Our analyses precisely pinpoint the thousands of proteins of whose connectivity in MDA-MB-468 cells is affected through epiHSP70s formation. We show these proteins map onto pathways that become altered in the specific cancer cell context, and are key to maintaining malignant phenotypes, supporting the use of YK5-B epichaperomics datasets to inform on cell-specific PPI networks perturbed in the specific context of epiHSP70s positive cells.

Recent large scale quantitative profiling of thousands of proteins by MS across 375 cancer cell lines proposed functional mapping across cancer cell lines as a more powerful proteome organizer than is mutational analysis, transcript inventory or tissue lineage[12]. Because proteome function is encoded in the architecture of PPIs, it is the analysis of context-dependent PPI networks that provides a most comprehensive functional mapping of disease. PPI changes in disease are sometimes driven by changes in protein expression, but most often tend to be modulated by alterations in the strength of interactions and in cellular mislocalization, both which in turn can be influenced by alterations in post-translational modifications, stabilization of disease-enriched protein conformations, and other protein-modifying mechanisms[24]. Epichaperomics thus is positioned as an important complement to current omics approaches that aim to functionally examine disease proteomes across cells and tissues.

A question arises regarding the applicability of this method in disease identification and therapeutics. A prior study in cancer identified 60-70% of tumors express variable levels of epiHSP90s, independently of tissue origin, tumor subtype, or genetic background[17,33]. While a similar systematic profiling of epiHSP70s remains to be performed, our assessment of 73 cancer cell lines, encompassing 9 tumor types, and of 19 primary breast tumors, for either vulnerability to YKs or epiHSP70s levels, suggest epiHSP70 formation occurs in ~70% of tumors. This approximates with epiHSP70s level measurements in the brains of AD patients[19], collectively supporting a wide applicability of this method for deciphering PPI network dysfunctions in disease. Our data indicate YK5-B and PU-beads provide both overlapping but also complementary information, proposing the use of both probes for PPI network dysfunction studies. Accordingly, we foresee implementation of the YK-probes, along with the reported PU-probes[22], to the analysis of large biospecimen sets−both cultured cells and primary specimens−to derive important context-specific insights into disease biology, in both cancer and neurodegenerative diseases[2,16].

We acknowledge aminopyridine-derived acrylamides (such as in YK5) have the intrinsic ability to label reactive macromolecules. This potential limitation is only relevant if it confounds biological investigation as it pertains to epichaperomics. The current study shows it does not. This is in agreement with other large-scale studies that found electrophile labeling of proteins in cells is in general specific and depends on the ligand and the structured sites in the protein[76]. Furthermore, it is important to highlight the most relevant feature in the context of epichaperomics are the kinetics of how the probe impacts the target. For the present report this comprises the epichaperome-interactor assemblies in live cells. We show the probes initially trap the assemblies, in the timeframe optimal for epichaperomics, prior to inducing their collapse or disassembly. Thus, in the context of epichaperomes, the kinetics of epichaperome-interactor assembly trapping and collapse are the most important factors.

We provide a comprehensive series of biochemical and functional studies to validate YK-epichaperomics datasets and data analytics.

Lending support to our analysis pipeline and data interpretation we recreated the map of pathway dysfunctions reported for MDA-MB-468 cancer cells and were able to discover with high confidence the exact proteins and protein pathways dysregulated by epiHSP70s formation in cancer. Taken together, multiple, independent lines of experimental evidence attest to the suitability of our experimental pipeline for the purpose of epichaperomics analyses for discovery science.

The present work also unveils an unappreciated role of the cancer epichaperomes. While chaperones are often associated with folding and aggregate dissociation[77], this study highlights much greater resourcefulness of chaperones in cancer. We provide evidence when the chaperome restructures into epichaperomes, it becomes a means to increase, not only balance, the fitness of the cancer proteome. This functional trait renders proteins and protein networks required for mitotic cell transition more effective in adapting to unique demands placed on the cancer cell. Aggregated data affirm epiHSP70s have emerged as a context-dependent regulatory mechanism for the control of mitosis in epiHSP70s positive cancer cells, which has important mechanistic and therapeutic implications.

A diversity of mechanisms may control the formation of mitotic structures, such as the mitotic spindle, likely in distinct ways within different tissues[58]. Our study unveils a context-dependent regulatory mechanism for the control of mitotic structures that has emerged in epiHSP70s positive cancer cells. For example, the tethering of spindle poles, as well as formation of the mitotic spindle, processes reliant on NuMA and its protein partners and regulatory proteins, require epiHSP70s in epiHSP70s-positive cancer cells. Context-dependence refers to cancer cells positive for epiHSP70s, keeping in mind that not all cancer cells express HSP70s-containing epichaperomes. Since non-transformed cells lack epichaperomes, we refer to this mechanism as a cancer phenomenon in epiHSP70s positive cancer cells, rather than a general mechanism of mitosis regulation. We cannot conclude all these effects of epiHSP70s manifest directly through NuMA or indirectly through NuMA regulatory proteins. Several mitotic regulators known to regulate and interact with NuMA (e.g. Aurora A and PLK1, among others)[60,63,65] were identified as epiHSP70s interactors via epichaperomics. Whether epiHSP70s bind separately to each protein or to several as part of a complex, remains unknown. Our previous findings that epichaperomes act as molecular scaffolds that change how thousands of proteins interact in cells exposed to pathologic stressors, argue that epiHSP70s bind not to individual mitotic proteins per se but rather alter the complexation of such proteins. Thus, epiHSP70 may exert its effect on mitosis by altering the complexation of mitotic regulator proteins in such way that the fitness of mitotic processes is increased. It will be important to investigate whether posttranslational modifications of specific HSP70s and HSP90s residues previously reported to affect mitotic functions (e.g., T115 in HSP90α and T66 in HSP70)[50,78] are in fact having their effect through epichaperome formation. Mechanisms changing HSP70s into epiHSP70s remain unknown, but aberrant glycosylation was reported for the HSP90 paralog GRP94 to facilitate epichaperome incorporation[20].

Importantly, YKs, are useful tools to study, and potentially revert, context-dependent perturbations in PPI networks. This has important therapeutic implications proposing YKs, akin to the epiHSP90 agents PU-H71 (zelavespib) and PU-AD (icapamespib)[33,79], may represent drug candidates to target PPI network dysfunctions[24]. Epichaperomes are scaffolds that rewire PPI networks under pathologic stressor conditions. Application of epichaperome disruptors to cancer therapy is governed by complex, PPI-network related principles, as reported[17,21,24]. The higher the epichaperome levels, the higher the number of proteins being negatively impacted, translating to a higher number of PPIs being rewired. Thus disrupting epichaperomes is detrimental to cancer cells in a manner directly proportional to epichaperome levels−the higher the epichaperome levels the higher the vulnerability of cancer cells to epichaperome therapy as a higher sector of the PPI network map is

affected upon epichaperome disruption[17,21]. Thus we foresee implementation of YKs to cancer on these principles developed for epichaperome cancer therapy[24,34], which remains to be validated.

In summary, this work demonstrates the importance of YK5-B-based epichaperomics to identify and study PPI network dysfunctions in disease and to provide mechanistically and therapeutically relevant insights into the PPI networks that support the pathologic phenotype in specific stressor contexts. Innovation and utility range from epichaperomics probe discovery to mechanistic insights and highlights the power of the method in deriving biology undetected or appreciated by conventional 'omics or targeted approaches. Importantly, our systems levels analyses combined with relevant functional studies, support the use of YK agents as therapeutic strategies aimed at normalizing and/or rebalancing PPI networks negatively impacted by epiHSP70s.

## Methods

### Human biospecimens research ethical regulation statement
Surgical specimens were obtained in accordance with the guidelines and approval of the Institutional Review Board at Memorial Sloan Kettering Cancer Center, Biospecimen Research Protocol# 09-121, project title: Ex-Vivo Testing of Breast Cancer Tumors for Sensitivity to Inhibitors of Heat Shock Proteins and Signaling Pathway Inhibitors, S. Modi, PI. The source of samples consists of unused portions of surgical specimens that are taken for reasons other than research (i.e. for patients undergoing the procedures for medical reasons unrelated to need for research samples or to the nature of the research). No individuals were excluded on the basis of age, sex or ethnicity. Because breast cancer is a disease which overwhelmingly affects women, and is a disease that is generally not seen in children, the vast majority of patients enrolled were females >18 years of age. Patient tissue samples were obtained with consent provided in written form. Samples were de-identified before receipt for use in the studies.

### Reagents and chemical synthesis
All commercial chemicals and solvents were purchased from Sigma Aldrich or Fisher Scientific and used without further purification. The identity and purity of each product was characterized by MS, HPLC, TLC, and NMR. Purity of target compounds has been determined to be >95% by LC/MS on a Waters Autopurification system with PDA, MicroMass ZQ and ELSD detector and a reversed phase column (Waters X-Bridge C18, 4.6 × 150 mm, 5 μm) eluted with water/acetonitrile gradients, containing 0.1% TFA. Stock solutions of all inhibitors were prepared in molecular biology grade DMSO (Sigma Aldrich) at 1000× concentrations. The epiHSP70 ligands and probes, the epiHSP90 inhibitor PU-H71, the PU-beads and the control baits were generated using published protocols[27–31,80] and described in Supplementary Notes 1. Thymidine, nocodazole, RO-3306, MG132, paclitaxel, VER155008 and MKT077 were purchased from Sigma-Aldrich. Proteins were generated and purified using published protocols described in Supplementary Note 2[81].

### Cell lines and culture conditions
Cell lines selection was not based on gender, sex or ethnicity. Cell lines were cultured according to the providers' recommended culture conditions. Cells were authenticated using short tandem repeat profiling and tested for mycoplasma. Breast cancer cell lines MDA-MB-468 (HTB-132, RRID: CVCL_0419), MCF-7 (HTB-22, RRID: CVCL_0031), SK-Br-3 (HTB-30, RRID: CVCL_0033), AU565 (CRL-2351, RRID: CVCL_1074), T-47D (HTB-133, RRID: CVCL_0553), BT-20 (HTB-19, RRID: CVCL_0178), MDA-MB-415 (HTB-128, RRID: CVCL_0621), MDA-MB-453 (HTB-131, RRID: CVCL_0418), HCC1806 (CRL-2335, RRID: CVCL_1258), MDA-MB-361 (HTB-27, RRID: CVCL_0620), MDA-MB-231 (CRM- HTB-26, RRID: CVCL_0062) were purchased from ATCC. Leukemia cell lines KASUMI-1 (CRL-2724, RRID: CVCL_0589) and K562 (CCL-243, RRID: CVCL_0004)

were obtained from ATCC and MOLM-13 (ACC-554, RRID: CVCL_2119) was from DSMZ. Pancreatic cancer cell lines ASPC-1 (CRL-1682, RRID: CVCL_0152), BxPc-3 (CRL-1687, RRID: CVCL_0186), SU.86.86 (CRL-1837, RRID: CVCL_3881), Capan-1 (HTB-79, RRID: CVCL_0237), Capan-2 (HTB-80, RRID: CVCL_0026), CFPAC (CRL-1918, RRID: CVCL_1119), Panc-1 (CRL-1469, RRID: CVCL_0480) and MiaPaCa2 (CRL-1420, RRID: CVCL_0428) were from ATCC. 931102 and 931019 were patient derived cell lines provided by Dr. Y. Janjigian, MSKCC[21]. Gastric cancer cell line MKN74 (RRID:CVCL_2791) was obtained from G. Schwarz (Columbia University) and obtained from AcceGen Biotechnology and OE19 (ACC-700, RRID: CVCL_1622) from DSMZ. The ovarian cancer cell lines PEO-1 (RRID:CVCL_2686), PEO-4 (RRID:CVCL_2690), OVCAR4 (RRID:CVCL_1627), OV1847 (RRID: CVCL_D703), A2780 (RRID:CVCL_0134), IGROV-1 (RRID:CVCL_1304) and OVCAR5 (RRID:CVCL_1628) were kindly provided by Dr. D. Solit, MSKCC and obtained from Millipore Sigma. Lung cancer cell line NCI-H3122 (RRID:CVCL_5160) was provided by M. Moore, MSKCC and obtained from Creative Biolabs, and NCI-H2228 (CRL-5935, RRID: CVCL_1543), NCI-H1975 (CRL-5908, RRID: CVCL_1511), NCI-H1373 (CRL-5866, RRID: CVCL_1465), A549 (CCL-185, RRID: CVCL_0023), NCI-H647 (CRL-5834, RRID: CVCL_1574), NCI-H526 (CRL-5811, RRID: CVCL_1569) were obtained from ATCC. Epstein-Barr virus positive Burkitt's lymphoma cell lines Akata1 (RRID:CVCL_0148)[82], Mutu1 (RRID:CVCL_7202)[82] and Rael (RRID:CVCL_7208)[82] were provided by W. Tam (WCMC, obtained from Lonza) and BCP-1 (CRL-2294, RRID: CVCL_0107), Daudi (CCL-213, RRID: CVCL_0008), EB1 (HTB-60, RRID: CVCL_2027), NAMALWA (CRL-1432, RRID: CVCL_0067), P3HR-1 (HTB-62, RRID: CVCL_2676), SU-DHL-6 (CRL-2959, RRID: CVCL_2206), Farage (CRL-2630, RRID: CVCL_3302) and Pfeiffer (CRL-2632, RRID: CVCL_3326) were obtained from ATCC; HBL-1 (RRID:CVCL_4213) from Applied Biological Materials (abm), MD901 (RRID:CVCL_D709)[83] and U2932 (RRID:CVCL_1896) from DSMZ were kindly provided by J. Angel Martinez-Climent (Centre for Applied Medical Research, Pamplona, Spain); Karpas422 (ACC-32, RRID: CVCL_1325), RCK8 (ACC-561, RRID: CVCL_1883) and SU-DHL-4 (ACC-495, RRID: CVCL_0539) were obtained from the DSMZ; OCI-LY1 (RRID:CVCL_1879), OCI-LY4 (RRID:CVCL_8801), OCI-LY7 (RRID:CVCL_1881) were obtained from the Ontario Cancer Institute; TMD8 (RRID:CVCL_A442)[84] was kindly provided by L. M. Staudt (NIH); BC-1 (RRID:CVCL_1079) was derived from an AIDS-related primary effusion lymphoma[85]; IBL-1 (RRID:CVCL_9638) was derived from an AIDS-related immunoblastic lymphoma[86] and BC-3 (RRID:CVCL_1080) was derived from a non-HIV primary effusion lymphoma[87]. Neuroblastoma cells SY5Y (CRL-2266, RRID: CVCL_0019) was purchased from ATCC; LAN5 (RRID:CVCL_0389) and SMS-KCNR (RRID:CVCL_7134) were obtained from the Children's Oncology Group (COG). Multiple Myeloma cell lines U266 (RRID:CVCL_0566) and MM.1 R (RRID:CVCL_8794) from ATCC were kindly provided from Dr. Z. Li (OSU), PCNY1 was derived as reported[88], and NCI-H929 (CRL-9068, RRID: CVCL_1600) was purchased from ATCC. Ewing's sarcoma cells TC71 (RRID:CVCL_2213) from DSMZ and A673 (RRID:CVCL_0080) from ATCC were kindly provided by Dr. S. Ambati, MSKCC. Cervical cancer cells HeLa (CCL-2, RRID: CVCL_0030) and colon fibroblasts CCD-18 (CRL-1459, RRID: CVCL_2379) were purchased from ATCC. HeLa cells expressing histone 2B-mCherry and α-tubulin EGFP were kindly provided by Dr. Daniel Gerlich, Institute of Molecular Biotechnology, Austria and were previously reported[89].

### siRNA transfection experiments
Transient transfections were carried out with Lipofectamine RNAiMax reagent (Invitrogen) according to the manufacturer's instructions. All the siRNAs were purchased from QIAGEN. The siRNA for HSP70 was mixture of HSPA1A: 5'-AGAGATGAATTTATACTGCCA-3' (SI00442974) and HSPA1B: 5'-AAGGTTCTGGACAAGTGTCAA-3' (SI04753938). The siRNA for HSC70 was HSPA8: 5'-AAGGACCTAAATTCGTAGCAA-3' (SI02661477). Scramble siRNA was used as a control (AllStars Negative

control siRNA, Qiagen). Briefly, MDA-MB-468 or HeLa cells were plated in 6-well plates at $1 \times 10^6$ cells per plate. The cells were transfected with the aforementioned siRNAs according to the manufacturer's instructions. The samples were collected 48 h post-transfection and subjected to further assays.

## Application of chemical probes
When probing specificity, it is informative to use probes that act on a select target with distinct potencies—from nanomolar to micromolar efficacy. If these probes show similar biological effects when tested at concentrations reflecting target engagement, that is supportive of specific, on-target activity. This is specifically why we included three probes (YK5, YK198, and LSI137) that act on epiHSP70 with a potency that spans micromolar to low nanomolar ranges. Please see Supplementary Figs. 6 and 13 for the activity ranges and choice of concentrations. Most experiments in cells use a concentration that is >IC$_{50}$ or is close to the C$_{100}$ value (i.e. target saturation). For biochemical experiments where the purpose is to show specificity, such as CETSA and affinity purifications, concentrations are much higher than the IC$_{100}$ for epiHSP70. Employing such high concentrations for long-term treatment of cancer cells is not feasible for epiHSP70-high cancer cells as they are sensitive to epiHSP70 inhibition and die following 24 h treatment. There are several chemical controls in this study. The most appropriate chemical controls are (1) YK56, to exclude the contribution of the acrylamide for possible non-specific covalent modification of proteins (negative control); (2) YK198 and LSI137, non-covalent probes with on-target activity (positive controls) and (3) HSP70 ligands with on-HSP70 activity through other binding mechanisms. The propionic acid amide of YK5 (i.e. where the acrylamide is inactivated) was previously made, characterized and tested, and used as starting point for the discovery of non-covalent probes[30].

## Biochemical studies in reconstituted systems
*Peptide release*. Fluorescence anisotropy experiments were performed on the FluoroMax HORIBA spectro-fluorometer and data was collected with the FluorEssence software (version 3). For peptide release, 5 µM HSC70 or HSP70 was pre-incubated with 25 nM FAM-TSLLMVIMG (LifeTein) and 5 µM ADP in buffer G (100 mM KOAc, 20 mM Hepes-KOH, pH 7.5, and 5 mM MgOAc$_2$) for 1 h at room temperature. Release was started with 1 mM ATP, 1000× excess unlabeled TSLLMVIMG with DMSO or 100 µM YK198. Fluorescence anisotropy was measured over time for FAM (excitation λ 492 nm, emission λ 520 nm). Integration time was 1 s, entrance and exit slits were set to 5 nm. The data was fitted and analysed using Prism. *ATP binding and ADP release*. All experiments were done on the FluoroMax HORIBA with an RX2000 stopped-flow attachment (Applied PhotoPhysics), in buffer G and all protein concentrations were at 5 µM. For ATP-binding; solution A: HSC70 or HSP70 were pre-incubated with DMSO or 50 µM YK198 for at least 1 h, solution B: 5 µM MABA-ATP (Jena Bioscience) with DMSO or 50 µM YK198. For ADP-release; solution A: HSC70 or HSP70 was pre-incubated with 5 µM MABA-ADP and DMSO or 50 µM YK198, solution B: Bag1 was prepared with 1 mM ADP, DMSO or YK198 and the buffer was supplemented with 1 mM phosphate. For all experiments, equal volumes of solutions A and B were quickly mixed, and fluorescence was measured over time for MABA (excitation λ 360 nm, emission λ 450 nm). Integration time was 0.1 s, entrance and exit slits were set to 1 nm. The data was fitted into a single exponential association (ATP-binding) or decay (ADP-release) using Prism. Rate-constants were derived from the equations. *Steady-state and single-turnover ATPase*. For steady-state ATPase experiments, reactions of 0.5 µM HSC70 or HSP70, 0.5 µM DJA2 J-domain with 0.5 µM Bag1 in buffer G were prepared on ice. ATPase reactions were started with 2 mM ATP spiked with radiolabeled [α-$^{32}$P] ATP at 30 °C. Samples were taken at different time points and stopped with 50 mM EDTA pH 8.0. Radiolabeled ATP and ADP were separated on polyethylenimine cellulose thin layer chromatography sheets (J.T. Baker), developed in 0.5 M lithium chloride and 0.5 M formic acid, captured on a phosphorimaging screen then detected on a Typhoon scanner and quantified on the Alpha View. Single-turnover experiments were performed similarly. HSC70 or HSP70 reactions with DJA2 J-domain were quickly mixed with 2 mM ATP spiked with radiolabeled ATP.

## Western blotting
Cells were plated in 6-well plates at $4 \times 10^6$ cells per plate. Cells were treated next day with the compounds at corresponding concentrations as indicated in each figure schematics. Protein extracts were prepared in 50 mM Tris-HCl, pH 7.4, 150 mM NaCl, 1% NP40 buffer supplemented with protease inhibitors cocktail (Complete tablets, Roche). The samples were centrifuged at the maximum speed for 10 min at 4 °C and the supernatants were collected. Protein concentrations in the lysates were determined using BCA kit (Pierce) according to the manufacturer's instructions. The proteins were resolved by SDS-PAGE, transferred onto nitrocellulose membranes and probed with a primary antibody, followed by a horseradish peroxidase (HRP)-conjugated secondary antibody. Horseradish peroxidase (HRP)-conjugated secondary antibody were purchased from SouthernBiotech—goat anti-rat IgG-HRP (3030-05, Lot# J1713-M322, 1:5000), goat anti-rabbit Ig, human ads-HRP (4010-05, Lot# A4211-ZH10E, 1:5000) and goat anti-mouse IgG, human ads-HRP (1030-05, Lot# D1922-X922, 1:5000). Blots were visualized by the method of enhanced chemiluminescence (Biorad). β-actin (A1978, Sigma-Aldrich, RRID: AB_476692, 1:3000) or GAPDH (2118, Cell Signaling, RRID: AB_561053, 1:10,000) were used as protein loading controls. Primary antibodies used in this study are listed below. HSP70 (ADI-SPA-810, RRID:AB_10616513, 1:2000), HSC70 (ADI-SPA-815, RRID:AB_10617277, 1:3000), HOP (SRA-1500, RRID:AB_10618972, 1:2000), HSP60 (ADI-SPA-806, RRID:AB_10617232, 1:1000) and HSP40 (SPA-400, RRID:AB_11180881, 1:3000) were purchased from Enzo; HSP90β (SMC-107, RRID:AB_854214, 1:4000) and HSP110 (SPC-195, RRID:AB_2119373, 1:2000) antibodies were from Stressmarq; HSP90α (ab2928, RRID:AB_303423, 1:5000), AHA1 (ab56721, RRID:AB_2273725, 1:1000), caspase 10 (ab177475, RRID: AB_2924729, 1:1000) and Securin (ab3305, RRID:AB_2173413, 1:1000) from Abcam; cleaved PARP (G734A, RRID:AB_430876, 1:1000) from Promega; CDC37 (4793, RRID:AB_10695539, 1:3000), phospho-AKT (S473) (9271, RRID:AB_329825, 1:2000), AKT (4691, RRID:AB_915783, 1:3000), phospho-ERK (T202/Y204) (4377, RRID:AB_331775, 1:3000), ERK (4695, RRID:AB_390779, 1:5000), MCL1 (5453, RRID:AB_10694494, 1:1000), RAF1 (12552, RRID:AB_2728706, 1:1000), BCL6 (14895, RRID:AB_2798638, 1:1000), eIF4E (2067, RRID:AB_2097675, 1:2000), phospho-eIF4E (S209) (9741, RRID:AB_331677, 1:500), Eg5 (14404, RRID:AB_2798473, 1:1000), phospho-STAT3 (Y705) (9145, RRID:AB_2491009, 1:1000), STAT3 (9139, RRID:AB_331757, 1:2000), Aurora A (91590, RRID:AB_2800171, 1:1000), PLK1 (4513, RRID:AB_2167409, 1:2000), TPX2 (12245, RRID:AB_2716832, 1:250), EGFR (4267, RRID:AB_2246311, 1:1000), Cyclin A2 (67955, AB_2909603, 1:1000), NuMA (ABE1361, RRID:AB_2892052, 1:1000) from EMD Millipore; NuMA (NB100-74636, RRID:AB_1049265, 1:1000) from Novus Biologicals; Cyclin B1 (554178, RRID:AB_395289, 1:1000), Nek2 (610593, RRID:AB_397933, 1:1000) and Aurora-A (610938, RRID:AB_398251, 1:1000) were from BD Transduction Laboratories; Cyclin A (sc H-432, RRID:AB_631329, 1:1000) and Cdc20 (sc-8358, RRID:AB_2291311, 1:1000) were from Santa Cruz. For protein quantification, films were scanned and analysed with ImageJ (Version 2.0.0) (US National Institutes of Health).

## Cell cycle synchronization
The cells were blocked in G$_1$/S phase by double thymidine block, as reported[90,91]. Specifically, HeLa cells were incubated with 2 mM of thymidine (Sigma-Aldrich, T1895) for 18 h. Then thymidine was removed by washing twice with PBS and the cells were released into fresh media for 9 h. Then 2 mM of thymidine was added, and cells were

incubated for 15 h. For single thymidine block, cells were treated with 2 mM thymidine for 24 h. In order to synchronize HeLa cells into M phase, 2 mM of thymidine was added to cells for 24 h. Following thymidine wash-off, cells were allowed to grow in fresh media for 2 h. Nocodazole was then added for 12 h to a final concentration of 25 ng mL$^{-1}$. Mitotic cells were collected by centrifugation. For $G_2$/M synchronization, cells were treated for 16 h with the CDK1 inhibitor RO3306 (10 µM). Following release from the RO3306 block, MG132 (10 µM) was added to cells and incubated for 3 h, as reported[90].

## Immunoprecipitation

MDA-MB-468 cells were plated in 10 cm plates at $4 \times 10^6$ cells per plate and treated with 2 uM LSI137 for the indicated times. Cells were collected and lysed in 20 mM Tris pH 7.4, 20 mM KCl, 5 mM MgCl$_2$, 0.01% NP40 buffer supplemented with protease inhibitors cocktail (Complete tablets, Roche). The samples were centrifuged at the maximum speed for 10 min at 4 °C and the supernatants were collected. Protein concentrations in the lysates were determined using BCA kit (Pierce) according to the manufacturer's instructions. For immunoprecipitation, 250 µg of total protein was mixed with HSC70 antibody (ADI-SPA-815, Enzo, RRID:AB_10617277, 1:100) or normal rat IgG (sc-2026, Santa Cruz, RRID:AB_737202, 1:100). After incubation at 4 °C overnight, 30 µL of protein G agarose beads (EMD Millipore) was added to the samples and incubated for 2 h at 4 °C. The beads were washed five times with the lysis buffer, and the protein complexes were eluted with SDS loading buffer. The eluted samples were applied on SDS-PAGE for protein separation, followed by Western blot analysis.

## Epichaperome staining in cells in culture

Cells were seeded and grown to 60-70% confluency in Lab-Tek II Chambered Cover glass W/Cover #1.5 Borosilicate (Nunc, 155382). On the day of the experiment, media was replaced with a fresh medium containing 1 µM epiTCO probe and incubated for 1 h at 37 °C in a cell culture incubator. The cells were then washed three times with fresh medium for 5 min to remove unbound compound. Following washing, cells were fixed with fresh 4% paraformaldehyde (PFA) for 15 min at room temperature. The fixed cells were washed twice with PBS. The cells were then permeabilized with 0.2% Triton X-100 in PBS for 15 min at room temperature followed by washing two times with PBS. Click reaction was performed with 700 nM Cy5-tetrazine (Click Chemistry Tools, Cat# 1189-1) for 12 min at room temperature. Cells were then washed three times with PBS. Finally, cells were incubated for 5 min with 300 nM DAPI in PBS for nuclear staining. Confocal imaging of the stained cells was done using LSM880 with airyscan (Zeiss) for representative images of epiTCO stained cells in different cell cycle phases. Quantification of mitotic cells per area was done by counting the cells in different mitotic phases based on their DAPI staining pattern using confocal images taken with 20x objective. Ten different microscopic fields were chosen randomly to perform the counting with a minimum of 200 cells per group considered for counting. HSP90α and HSC70 staining was performed following the procedure described below under Immunofluorescence microscopy. All imaging analysis, including measurement of mean fluorescent intensity (MFI), was performed using ZenBlue (Version 3.7) (Zeiss) software.

## Immunofluorescence microscopy

Cells were grown in chamber slides (lab-Tek II). After inhibitor treatment, the cells were fixed with −20 °C methanol for 5 min, followed by treatment with ice-cold acetone for 30 s. After washing with PBS containing 0.1% Triton X-100, the cells were blocked with 3% BSA in washing buffer for 1 h. Primary antibodies were prepared in blocking buffer and added into the chambers for 1 h incubation at room temperature. After three washes, the cells were incubated with Alexa Fluor 488 or Alexa Fluor 568 conjugated secondary antibodies (Life technologies, A-11004 (RRID:AB_2534072), A-11008 (RRID:AB_143165),1:2000) for 1 h. After

washing, cells were mounted with ProLong Gold Antifade Mountant with DAPI (Life Technologies) and visualized under a confocal microscope (Leica, SP5-Up). The primary antibodies used in this study were: HSC70 (ADI-SPA-815, RRID:AB_10617277, 1:250) from Enzo; HSP90α (ab2928, RRID:AB_303423, 1:500) from Abcam; NuMA (ABE1361, RRID:AB_2892052, 1:100) from EMD Millipore; α-tubulin (T5168, RRID:AB_477579, 1:250) from Sigma-Aldrich; Pericentrin (ab4448, RRID:AB_304461, 1:100), NEDD1 (ab57336, RRID:AB_944385, 1:100) from Cell Signaling.

## Cell confluence monitoring via microscopy

IncuCyte live-cell microscopy system (Essen BioScience) was used to evaluate cell confluency. HeLa or ASPC1 cells were plated in 6-well plates at $1–2 \times 10^6$ cells per plate and subjected to double thymidine block as described above. Following the completion of the block, the cells were washed twice with PBS and fresh media was added. Alternatively, cells were kept in thymidine. 2 µM YK198 or vehicle was added to the cells and the culture incubated for 24 h. Cell growth was monitored with the IncuCyte system. Frames were captured from nine separate regions per well using 10× or 20× objective. Confluence was measured using the IncuCyte software, where values were pooled, and the mean used to plot each datapoint on the graph. Data were imported and analysed in GraphPad Prism 9.

## Breast cancer explant treatment

Cell culture provides both direct and rapid analysis of therapeutic sensitivity and resistance. However, therapeutic response is not exclusive to the inherent molecular composition of cancer cells but rather is greatly influenced by the tumor cell microenvironment, a feature that cannot be recapitulated by traditional culturing methods. Cell lines in culture also fail to account for the barrier imposed on drug access to the cancer cell by the surrounding structures (i.e. extracellular matrix and stromal cells). To this extent, we have developed an ex vivo (fresh tissue sectioning, explant) technique, as reported[92]. This technique allows for the direct assessment of treatment response for preclinical and clinical therapeutics development. This technique also maintains tissue integrity and cellular architecture within the tumor cell/microenvironment context throughout treatment response providing a more precise means to assess drug efficacy. Briefly, for this approach, the surgical specimen was harvested in a sterile environment and used under 30 min from the surgical procedure. A tumor tissue from the periphery of the index lesion was selected to avoid potential frank central necrosis (cell death). The necrotic tissue may be grossly recognizable by any of the following criteria: loss of color or paleness of the tissue; loss of strength in which necrotic tissue is soft and friable; a distinct demarcation between the necrotic and viable tissue. After sampling was done, the specimen was embedded in 4% agarose for ex vivo fresh tissue sectioning or frozen for epichaperome analysis by native PAGE. For each case multiple serial sections were performed, sample size permitting, and matched with controls (vehicle treated only). For ex vivo testing, the agarose-embedded tissue was mounted on the stage of a Vibratome submersed in a chilled reservoir (for tissue preservation) containing MEM with 10% FBS and 1% penicillin/streptomycin. The tissue was then sliced using metal blades producing serial sections of the lesion that are 200 µm thick. Each section was immediately placed in a pre-warmed 24-well tissue culture plate containing MEM with 10% FBS and 1% penicillin/streptomycin and treated for 24 h with vehicle control, YK5 (5 µM) or YK198 (2.5 µM). Tissues were removed from media, stored in 70% ethanol, followed by paraffin embedding, sectioning to 5 µm thick slices, and staining with hematoxylin and eosin (H&E) or TUNEL. For H&E staining, the tissue samples were fixed by immersing in ice cold freshly made 4% paraformaldehyde overnight at 40 °C. Following the fixation, the slides were washed in PBS at 40 °C x 30 min each. Then the samples were dehydrated in series of alcohol dilutions (70%, 95% and 100% ethanol)

for 30 min at room temperature and paraffin embedded. Then the tissue slides were stained with hematoxylin and eosin using the Leica Biosystems auto stainer. For TUNEL (Terminal deoxynucleotidyl transferase dUTP nick end labeling) staining, the breast cancer explant slides were manually deparaffinized in histoclear-xylene, rehydrated in series of alcohol dilutions (100%, 95% and 70%) and tap water. The slides were then washed three times with PBS buffer, treated with Protease K for 30 min, and washed again trice with PBS. Then the slides were incubated with Endogenous Biotin Blocking Kit (Roche, cat#760-050) for 8 min and then incubated with labeling mix: TdT (Roche, cat#03333566001, 1000U mL$^{-1}$) and biotin-dUTP (Roche, cat#11093070910, 4.5 nmol mL$^{-1}$) for 2 h. Detection was performed with Streptavidin-HRP and DAB-MAP detection kit (Ventana Medical Systems) according to manufacturer instructions. The slides were counterstained with hematoxylin and coverslipped with Permount (Fisher Scientific). Apoptosis and necrosis of the tumor cells (as percentage) was evaluated by a pathologist by reviewing all the H&E slides of the case (controls and treated ones) in toto, blindly, allowing for better estimation of the overall treatment effect to the tumor. In addition, any effect to precursor lesions (if present) and off-target effect to any benign surrounding tissue were evaluated.

## Chemical affinity purification

For Fig. 1b and Supplementary Fig. 14b, MDA-MB-468 protein extract prepared in a buffer containing 20 mM Tris pH 7.4, 20 mM KCl, 5 mM MgCl$_2$, 0.01% NP40 buffer with added protease inhibitors (buffer B) was incubated with increasing concentrations of YK5-B or biotin for 1 h at 4 °C. High-capacity streptavidin beads (40 μL) were added to each sample and the mix was incubated at 4 °C for 1 h. Following the incubation, the samples were spun down and the pull-down supernatants were collected and applied to native gel. For Fig. 1c, Supplementary Fig. 13c and Supplementary Fig. 30d, MDA-MB-468 cells were lysed as described above and incubated with YK5-B, YK56 or biotin beads for 1 h at 4 °C. Figure 1d and Supplementary Fig. 30b, MDA-MB-468 or ASPC1 cells were processed as described above and incubated with 25 μM of YK5-B or YK56 beads for 1 h at 4 °C. YK5-B beads or biotin beads were made by adding the indicated concentrations of YK5-biotin or biotin to 40 μL of high-capacity streptavidin beads (Thermo Scientific). The mix was incubated at 4 °C for 1 h, after which beads were spun down on a mini centrifuge, the supernatant was removed by aspiration, and then beads were washed 3 times with lysis buffer. The beads were added to the protein extracts and incubated overnight at 4 °C. For Fig. 1e, MDA-MB-468 cells were treated for 2 h with either PU-H71 (1 μM), LSI137 (1 μM) or vehicle. Following treatment, cells were lysed in buffer B and incubated at 4 °C with PU-beads (40 μL for 3 h) or YK5-B beads (1 h). For Supplementary Fig. 14c, MDA-MB-468 protein extract, prepared as described above, was incubated with either YK5-B beads (made as above by incubating streptavidin beads with 100 μM YK5-B) or HSC70 Ab (5 μL added to protein G agarose) at 4 °C overnight. Beads were collected, washed and processed for Western blot analysis. The supernatant was collected and re-incubated four more times with either YK- or Ab-beads, as above. For all affinity purifications, following washing with lysis buffer (4x), the cargo was eluted by boiling in loading buffer and applied to SDS-PAGE. HSC70 values in each experimental condition were normalized to account for the difference in input, and the recorded HSC70 values were normalized to the maximal HSC70 bound values.

## CETSA

Modified protocol from ref. 93 was used for CETSA experiments. In brief, MDA-MB-468 or ASPC1 cells were plated in 15 cm plates at 15 × 10$^6$ cells per plate, treated for 1.5 h with indicated compounds, collected and lysed in PBS by freeze-thaw procedure. The samples were centrifuged at 20,000x$g$ for 20 min at 4 °C and the supernatants

were collected. Protein lysates were aliquoted and subjected to heat treatment at indicated temperatures for 3 min, and subsequently incubated at room temperature for 3 min. The samples were centrifuged, and the supernatant was collected and quantified using BCA kit (Pierce) according to the manufacturer's instructions. The samples were applied to SDS-PAGE and subjected to Western blotting. For Supplementary Fig. 30a, the cells were lysed as described above without treatment and processed for CETSA the same way as above. For the cell lysate CETSA experiments, MDA-MB-468 cells were harvested, and protein was extracted as described above. Protein lysates were aliquoted and incubated with 30 μM LSI137 or 100 μM YK5 or vehicle for 30 min at room temperature. After incubation, lysates were divided into smaller aliquots, heated at different temperatures, and processed further as described above.

## Native gel electrophoresis

MDA-MB-468 or OCI-LY1 cells were plated in 10 cm plates at 4-6 ×10$^6$ cells per plate and treated with indicated compounds for indicated times (please, refer to the specific schematics and figure legends). Cells were lysed in 20 mM Tris pH 7.4, 20 mM KCl, 5 mM MgCl$_2$, 0.01% NP40 buffer by freeze-thaw procedure. 65 μg of total protein was loaded onto 4–8% native gradient gel and resolved at 4 °C for 2-3 h. The protein complexes were transferred onto nitrocellulose membranes in Tris-Glycine-Methanol transfer buffer containing 0.025% SDS at 4 °C for 2 h. The membranes were immunoblotted as described above. A panel of anti-chaperone antibodies has been screened to identify the ones recognizing the target protein in its native form. These native-cognate antibodies were used in native-PAGE analysis of epichaperome assemblies. HSP90β (SMC-107, RRID:AB_854214, 1:3000) and HSP110 (SPC-195, RRID:AB_2119373, 1:2000) antibodies were purchased from Stressmarq; HSP70 (SPA-810, RRID:AB_10616513, 1:2000), HSC70 (SPA-815, RRID:AB_10617277, 1:3000), HOP (SRA-1500, RRID:AB_10618972, 1:2000) from Enzo; HSP90α (ab2928, RRID:AB_303423, 1:6000) from Abcam.

## Time lapse microscopy

HeLa cells were plated in the chambered coverglass (Lab-Tek II). The cells were blocked in G$_1$/S phase by double thymidine block. Specifically, 2 mM thymidine was added to the cells and incubated for 18 h. Then thymidine was removed by washing with PBS and the cells were incubated in fresh media for 9 h. Then 2 mM thymidine was added for the second time and incubated for 15 h. Then thymidine was removed, and fresh media was added. Vehicle or YK198 was added to the cells 4 h later, and time lapse imaging was started with the Axio Observer (ZEISS). The cells were placed in a 37 °C, 5% CO$_2$ environment during the time lapse study. Images were taken every 5 min continually up to 48 h with a 40x objective lens. Images were processed in ImageJ (Version 2.0.0).

## Flow cytometric analysis of apoptosis

Briefly, cells were seeded into 6-well plates at 1 × 10$^6$ cells per plate and allowed to adhere overnight in regular growth media before synchronization with thymidine. For thymidine removal, the culture medium was removed by suction, and cells were washed twice with PBS. Fresh culture medium was added, and cells were then treated with vehicle (DMSO), 2 μM YK198, or PU-H71 for 24 h. Both floating and adherent cells were harvested and washed twice with PBS. Cells were then resuspended in 150 μL of the Annexin V Binding Buffer at a concentration of 0.5 × 10$^6$ cells mL$^{-1}$. Following gentle vortexing, 8 μL of annexin-V-FITC and 8 μL of propidium iodide (Biolegend, Cat. No. 640914) were added to each sample. Cells were incubated for 25 min at room temperature in the dark and analysed by flow cytometry on an LSRFortessa instrument (BD Biosciences). The percentage of cells undergoing apoptosis and necrosis was determined using FlowJo software (Version 10.0) (FlowJo LLC).

## Cell cycle analysis by flow cytometry

Cells were plated in the complete DME media at 50% confluency. After thymidine synchronization, cells were then washed twice with PBS and once with the complete media. For inhibitor experiment, and immediately after the removal of thymidine, cells were treated with vehicle (DMSO) or YK198 for the time indicated. Cells were harvested in PBS, washed, and centrifuged at 100x*g* for 10 min at 4 °C to obtain the pellet, which was then fixed overnight with 70% ethanol (200 μL). Cells were washed twice with PBS and resuspended in 50 μg mL$^{-1}$ propidium iodide (BD Pharmingen, Cat. No. 51-66211E) and 0.5 mg mL$^{-1}$ Ribonuclease A (Qiagen, Cat. No.19101) in a total volume of 200 μL. The samples were incubated overnight at 4 °C in the dark before acquiring the data by flow cytometry using LSRFortessa (BD Biosciences). Data were analysed using FlowJo (Version 10.0) (FlowJo LLC).

## Quantification of phospho-MPM-2 positive cells by flow cytometry

Cells were collected by trypsinization, fixed with 70% ethanol overnight at 4 °C and washed three times with PBS. The fixed cells were incubated with 0.25% Triton X-100 in PBS for 15 min and washed with PBS three times. Next, the cells were stained with phospho-MPM2 (Millipore, #05-368, RRID:AB_309698, 1:500) antibody for 1 h at room temperature and washed three times with PBS. Then, the cells were incubated with Cy5-conjugated anti-mouse secondary antibody (Life technologies, A10524, RRID:AB_2534033, 1:2000) for 30 min and washed three times with PBS. Cells were then suspended in Propidium Iodide (PI)/RNase Staining Buffer (BD Pharmingen) and analysed by flow cytometry (LSR-II, BD Biosciences). FlowJo software (Version 10.0) was used for data analysis (Tree Star, Ashland, OR).

## Cell viability ATP assay

Cell viability was measured using CellTiter-Glo luminescent Cell Viability Assay (Promega) according to the manufacturer's instructions. Cells were plated in clear bottom black 96-well plates (Corning, 3603) at 4000–8000 cells per well and treated with serial dilutions of YK198 or PU-H71 starting at 5 μM for 72 h. After the incubation with the compounds, 100 μL of CellTiter-Glo reagent was added to each well. Plates were incubated with the reagent for 10 min at room temperature. The luminescence signal was measured with Analyst GT microplate reader (Molecular Devices). The percentage cell viability was calculated by comparison of the luminescence reading obtained from treated versus control cells, accounting for initial cell population.

## Sample preparation for epichaperomics

For in-cell YK-B bait affinity purification, MDA-MB-468 cells were plated in 10 cm plates at 6 × 10$^6$ cells per plate and treated with 25–75 μM YK5-B for 4 h. We suggest users consider, and test in their experimental setting, an incubation window of 1–4 h. Cells were next collected and lysed in 20 mM Tris pH 7.4, 150 mM NaCl and 1% NP40 buffer. 500 μg of total protein was incubated with streptavidin agarose beads (ThermoFisher Scientific) for 1 h and washed with 20 mM Tris pH 7.4, 100 mM NaCl and 0.1% NP40 buffer. For in-lysate YK5-B bait affinity purification, MDA-MB-468 cells were lysed in the above-mentioned lysis buffer. Streptavidin agarose beads were incubated with 25–100 μM YK5-biotin for 1 h, washed and added to 500 μg of total protein and incubated overnight. The beads were then washed with the lysis buffer. The samples were applied onto SDS-PAGE. The gels were stained with SimplyBlue Coomassie stain (ThermoFisher Scientific) and submitted for MS analysis. Gel lanes were cut into an average of 12 gel bands. Gel bands were completely destained with 50% methanol and 25 mM NH$_4$HCO$_3$ / 50% acetonitrile and diced into small pieces and dehydrated with acetonitrile and dried by vacuum centrifugation. The gel pieces were rehydrated with 12.5 ng mL$^{-1}$ trypsin solution (Trypsin Gold, Mass Spectrometry Grade, Promega) in 50 mM NH$_4$HCO$_3$ and incubated at 37 °C overnight. Peptides were extracted twice with 5% formic acid / 50% acetonitrile followed by final extraction with acetonitrile. The resulting peptides were desalted using a Stage Tip manually packed with Empora C18 High Performance Extraction Disks (3 M, St. Paul, MN, USA)[94]; eluted peptide solutions were concentrated to a very small volume by vacuum centrifugation and finally, reconstituted in 2% acetonitrile / 4% formic acid for mass spectrometry analysis.

## Sample preparation for proteomics

For the HSP/C70 knock-down experiment, MDA-MB-468 cells were plated in 6-well plates at 2 × 10$^6$ cells per plate and transfected with a combination of the HSP70 and HSC70 siRNAs or with scramble siRNA as described above or treated with 0.2 μM LS137 or vehicle for 24 h. The samples were collected and lysed in 50 mM Tris-HCl, pH 7.4, 150 mM NaCl, 1% NP40 buffer with protease inhibitors cocktail (Roche). 30 μg of total protein for each sample was loaded onto SDS-PAGE, run as a single stack, stained with Coomassie (ThermoFisher Scientific) and submitted for isobaric tandem mass tag TMT labeling and MS analysis[95]. The stacked protein bands were excised from the gel, and destained and in-gel digested as described above. Cleaned peptides were then re-suspended in 18 μL acetonitrile and to each, 57 μL of 0.2 M HEPES buffer, pH 8.5 added. TMT10-plex amine reactive reagents (5 mg per vial) (Thermo Fisher Scientific) were re-suspended in 1000 μL of anhydrous acetonitrile and 25 μL of each reagent was added to each sample (TMT label:peptide [w/w] = 12:1) and mixed briefly by vortexing. The mixture was incubated at room temperature for 1 h, quenched by the addition of 10 μL of 5% hydroxylamine for 15 min, and then acidified by the addition of 10 μL 10% formic acid. A small aliquot (5 μL) from each reaction was desalted on a on a Sep-Pak tC18 1 cc Vac Cartridge (Waters, #WAT03820), analysed by LC-MS/MS with a Q Exactive High Field Orbitrap, and resulting spectra searched with MaxQuant using its corresponding TMT label as variable modifications on N-terminus and lysine. The percentage of peptides with either N-terminal or lysine TMT labels was calculated, indicating the labeling efficiency for each channel. To ensure equal amounts of labeled peptides from each channel were mixed together, a two-step mixing strategy was employed; in the first step, a small (~1 μL) and identical volume of peptides from each channel was mixed and analysed, and the value of the median ratio (defined by the median of the ratios of all peptide intensities of one channel over their corresponding peptide average intensities of all channels) for each channel is determined as the correction factor. In the second step, the rest of the peptides were mixed by adjusting their volume using the correction factors. In this way, median ratios around 1 was achieved as previously reported. The final mixture of reaction products from 10 TMT channels were desalted on a Sep-Pak tC18 1 cc Vac Cartridge (Waters, #WAT03820). Eluted peptides were dried by vacuum centrifugation, and stored at −20 °C.

## Liquid Chromatography-Tandem Mass Spectrometry (LC-MS/MS) and data analysis

Label free proteomics analyses were performed using a Q Exactive mass spectrometer coupled to a Thermo Scientific EASY-nLC 1000 (Thermo Fisher Scientific, Waltham, MA) equipped with a self-packed 75 μm x 20-cm reverse phase column (Reprosil C18, 3 μm, Dr. Maisch GmbH, Germany) for peptide separation. The mass spectrometer was operated in data-dependent analysis (DDA) mode with survey scans acquired at a resolution of 70,000 over a scan range of 300-1750 m/z. Up to ten most abundant precursors from the survey scan were selected with an isolation window of 1.6Th and fragmented by higher-energy collisional dissociation with Normalized Collision Energies (NCE) of 27. The maximum ion injection time for the survey and MS/MS scans was 60 ms and the ion target value for both scan modes was set to 1e6. For TMT experiments, analytical columns ( ~ 30 cm long and 75 μm inner diameter) were packed to provide extra-long columns to

achieve optimal separation of the TMT tagged peptides. The TMT peptide mixture was loaded onto the analytical column with buffer A (0.1% formic acid) at a maximum back-pressure of 300 bar; peptides eluted with a 2 step gradient of 3% to 40% buffer B (100% acetonitrile and 0.1% formic acid) in 110 min and 40% to 90% B in 5 min, at a flow rate of 250 nL/min over 120 min using a 1D online LC-MS2 DDA method as follows: MS data were acquired using a data-dependent top-10 method, dynamically choosing the most abundant not-yet-sequenced precursor ions from the survey scans (300–1750 Th). Peptide fragmentation was performed via higher energy collisional dissociation with a target value of $1 \times 10^5$ ions determined with predictive automatic gain control. Isolation of precursors was performed with a window of 1 Th. Survey scans were acquired at a resolution of 120,000 at $m/z$ 200. Resolution for HCD spectra was set to 60,000 at $m/z$ 200 with a maximum ion injection time of 128 ms. The normalized collision energy was 35. The "underfill ratio," specifying the minimum percentage of the target ion value likely to be reached at the maximum fill time, is defined as 0.1%. Precursor ions with single, unassigned, or seven and higher charge states were excluded from fragmentation selection. Dynamic exclusion time was set at 30 s. Each of the TMT 10plex samples was analysed in technical duplicates.

## Protein identification

All mass spectra were first converted to mgf peak list format using Proteome Discoverer 1.4 and the resulting mgf files searched against a human Uniprot protein database using Mascot (Matrix Science, London, UK; version 2.7.0; www.matrixscience.com). Decoy protein sequences with reversed sequence were added to the database to allow for the calculation of false discovery rates (FDR). The search parameters were as follows: (i) up to two missed tryptic cleavage sites were allowed; (ii) precursor ion mass tolerance = 10 ppm; (iii) fragment ion mass tolerance = 0.1 Da; and (iv) variable protein modifications were allowed for methionine oxidation, deamidation of asparagine and glutamines, cysteine acrylamide derivatization and protein N-terminal acetylation. MudPit scoring was typically applied using significance threshold score $p < 0.01$. Decoy database search was always activated and, in general, for merged LS-MS/MS analysis of a gel lane with p < 0.01, false discovery rate averaged around 1%. The Mascot search result was finally imported into Scaffold (Proteome Software, Inc., Portland, OR; version 4.11) to further analyse tandem mass spectrometry (MS/MS) based protein and peptide identifications. X! Tandem (The GPM, thegpm.org; version CYCLONE (2010.12.01.1)) was then performed and its results were merged with those from Mascot. The two search engine results were combined and displayed at 1% FDR. Protein and peptide probability were set at 95% with a minimum peptide requirement of 1. Protein identifications were expressed as Exclusive Spectrum Counts (ESCs) that identified each protein listed. In each of the Scaffold files that validate and import Mascot searched files, peptide matches, scoring information (Mascot, as well as X! Tandem search scores) for peptide and protein identifications, MS/MS spectra, protein views with sequence coverage and more, can be easily accessed. To read the Scaffold files, free viewer software can be found at: http://www.proteomesoftware.com/products/free-viewer. The mass spectra files were also subjected to Label-Free Quantitation (LFQ) using MaxQuant proteomics data analysis workflow (version 1.6.0.1) with the Andromeda search engine[96]. Raw mass spectrometer files were used to extract peak lists which were searched with the Andromeda search engine against human or mouse proteome and a file containing contaminants such as human keratins. Trypsin specificity with 2 missed cleavages with the minimum required peptide length was set to be seven amino acids. N-acetylation of protein N-termini, oxidation of methionines and deamidation of asparagine and glutamines were set as variable modifications. For the initial main search, parent peptide masses were allowed mass deviation of 20ppm. Peptide spectral matches and protein identifications were filtered

using a target-decoy approach at a false discovery rate of 1%. Label-free quantification of proteins was activated to achieve a global proteome wide normalization of the intensities. For TMT experiments, all data were analysed with the MaxQuant proteomics data analysis workflow (version 1.5.5.1) with the Andromeda search engine[96,97]. The type of the group specific analysis was set to "Reporter ion MS2" with "10plex TMT" as isobaric labels for Q Exactive High Field MS2 data. Reporter ion mass tolerance was set to 0.01 Da, with activated Precursor Intensity Fraction (PIF) value set at 0.75. False discovery rate was set to 1% for protein, peptide spectrum match, and site decoy fraction levels. Peptides were required to have a minimum length of eight amino acids and a maximum mass of 4,600 Da. MaxQuant was used to score fragmentation scans for identification based on a search with an allowed mass deviation of the precursor ion of up to 4.5 ppm after time-dependent mass calibration. The allowed fragment mass deviation was 20 ppm. MS2 spectra were used by Andromeda within MaxQuant to search the Uniprot human database (downloaded 12202017; 20,244 entries) combined with 262 common contaminants. Enzyme specificity was set as C-terminal to arginine and lysine, and a maximum of two missed cleavages were allowed. Carbamidomethylation of cysteine was set as a fixed modification and N-terminal protein acetylation, deamidated (N, Q) and oxidation (M) as variable modifications. The reporter ion intensities were defined as intensities multiplied by injection time (to obtain the total signal) for each isobaric labeling channel summed over all MS/MS spectra matching to the protein group as previously validated[96]. Following MaxQuant analysis, the protein and peptide.txt files were imported into Perseus (version1.5.6.0) software which was used for the statistical analysis of all the proteins identified. The basic statistics used for significance analysis was the moderated t-statistics[98]. Benjamini-Hochberg correction was used to calculate the adjusted $p$-values.

## Grading scale for interactors

To maximize protein detection, while maintaining the specificity of the identified proteome, we introduced a grade system to control for the quality of the proteins identified by MS using exclusive spectrum counts (ESC). Proteins with ESC values in control bait higher than in the epiHSP70s bait were excluded. Then $P$-values were calculated by a two-sided Student's $t$-test comparing exclusive spectrum counts from epiHSP70 baits and control baits. Detected proteins were respectively ranked as Grade A ($p$-value < 0.1), Grade B ($0.1 \le p$-value $\le 0.25$), Grade C ($0.25 < p$-value < 0.5) and contaminants ($p \ge 0.5$) or only one entry across epiHSP70s bait replicates. The identity of the interactors and the associated statistical analyses are included in Supplementary Data 1.

## Bioinformatics analyses

The ESC and intensity values were used for protein relative quantitation[99]. All statistics related to proteomics analyses were performed using R (version 3.3.2)[100]. Building the PPI database. To prepare a PPI database for our analyses, we combined all entries from BioGrid (v.3.4.126) and IntAct (version 07.2015) to create a dataset that inventories all documented human-human (*Homo sapiens*), mouse-mouse (*Mus musculus*) or yeast-yeast (*Saccharomyces cerevisiae* S288c) PPI[101,102]. We also included inter-species PPIs from the IntAct database (e.g human-rat). We used the Uniprot database (https://www.uniprot.org) for ortholog conversion. In UniProt, potential orthologs are initially identified using sequence similarity search programs such as BLAST. Orthology relationships are then verified manually using a combination of resources including scientific literature, sequence analysis tools, phylogenetic and comparative genomics databases such as Ensembl Compara, and other specialized databases such as species-specific collections. HSP70 interactors (BioGrid+IntAct) were defined as the proteins having documented PPIs in BioGrid or IntAct with HSP7C_HUMAN, HS71A_HUMAN, HS71B_HUMAN or their

homologs in mouse and yeast. HSP90 interactors (BioGrid+IntAct) were defined as the proteins having documented PPIs in BioGrid or IntAct PPIs with HS90B_HUMAN, HS90A_HUMAN, or their homologs in mouse and yeast (HSC82_YEAST, HSP82_YEAST). Without otherwise being specified, all PPIs were based on the combined database, prepared as described above. Defining the chaperome. For the chaperome we included proteins annotated to function as chaperone, co-chaperones, isomerases (PPIase cyclophilin-type domain, PPIase FKBP-type domain and protein disulfide isomerases containing a thioredoxin domain)[103,104] and those that may act in scaffolding roles. These proteins were chosen either based on function (reported to act in scaffolding, docking, anchoring or targeting a protein to a cellular location) or on the presence in their structure of domains recognized to mediate such scaffolding functions. Building of the PPI network. The PPI network was built by extracting from the aforementioned PPI database, the proteins identified from the epiHSP70s bait and/or the epiHSP90s bait and the HSP70s/HSP90s interactors as defined in "Building PPI database". The node size was reduced to a dot if the protein was not identified from the epiHSP70s or the epiHSP90s bait but was reported as a direct interactor of HSP70s or HSP90s in the BioGrid or IntAct databases. For a better illustration of the topological layers between chaperome and non-chaperome proteins, we applied edge weighted spring-electric layout in Cytoscape (v3.40)[105]. The weight of the PPI was parameterized with three different absolute values based on the following rule: within the chaperome members, 10,000; between chaperome members and their direct interactors, 100, between the direct interactors and the remaining proteins, 1. The layout of the final topology network was generated by Allegro Spring-Electric layout algorithm with edge weighting on the chaperome.degree column. The resulting epiHSP70s/epiHSP90s interactor network was colored to reflect epiHSP70s or epiHSP90s bait specificity and/or chaperome and non-chaperome classification. For figure clarity, PPI edges were not shown. Proteins were represented by nodes of reduced size if they were not detected by the epiHSP70s bait but were recorded in the BioGrid or IntAct databases as direct interactors of HSP70s or HSP90s (with exclusion of chaperone members, which were represented uniformly by full size node). Simulation-based significance test for the observed number of YK-B and PU-beads interactor co-occurrences. Since the observed number of co-occurrences is driven by the dependency between epiHSP70s and epiHSP90s, the null hypothesis is formulated as the statistical independence between epiHSP70s and epiHSP90s across all proteins. The alternative hypothesis is that the occurrences of epiHSP70s and epiHSP90s are correlated. 100,000 Monte Carlo simulations were conducted to construct an empirical distribution of the number of co-occurrences out of the 5704 interactors, assuming epiHSP70s and epiHSP90s follow independent Bernoulli distributions across all interactors. The parameters of the Bernoulli distributions were estimated by the prevalence of epiHSP70s and epiHSP90s binders across interactors from observed data. Then the $p$-value was obtained by two times the minimum of percentile and quantile of the observed number of co-occurrences, with respect to the simulated empirical distribution. The null hypothesis is rejected when the $p$-value is <0.05 under a two-side test setting. Reactome pathway enrichment analyses. The analyses were performed using Reactome Pathway Database. The calculations of false positive rate (Entities FDR) and enrichment ratio (Entities ratio) were based on established method described in ReactomeWiki, section "Gene list Dataset"[106,107]. Proteins analysed were those from the epiHSP70s bait and had quality graded as A, A + B or A + B + C (see above Grading of interactor datasets). PPI networks were constructed using the comprehensive PPI mentioned above[101,102]. The Reactome pathway figure was generated in Cytoscape (v3.40)[105]. Each node represents a pathway including a collection of relevant proteins. Nodes are functionally interconnected in a hierarchical manner (high to low, indicated by arrowheads) and graphed as a tree structure. The

significance of the enrichment (FDR) and the number of proteins found in relevant pathways is reflected by a color gradient and by the size of the nodes. Irrelevant pathways in which no protein was related were represented as borderless nodes. Fisher's exact tests were performed using Bioconductor package 'clusterProfiler' and gProfiler's web platform[108] (Version e106_eg53_p16_65fcd97) (Supplementary Data 1). Study of proteome-wide functional changes induced by epiHSP70s inhibitions using quantitative proteomics. Experiments were carried out in two replicates. Corrected reporter intensity (the 'Report.intensity.corrected' columns in Supplementary Data 3) of the samples was log2 transformed and was normalized against the reference (containing the mixture of all samples combined, Channel 0) within the same replicate. Missing values were imputed by random forest (R package 'missForest')[109]. Differential expression analyses were performed on completed data matrix (R package 'limma'), comparing KO samples or LSI137 treated samples versus respective control samples using empirical Bayesian method to moderate $t$-tests. Nominal p.values were adjusted by BH method. Proteins differentially expressed in each pairwise comparison (adjusted $p$-value < 0.05) were separated into up-regulated (logFC > 0) and down-regulated (logFC < 0) groups and were respectively subjected to Fisher's exact test (using Bioconductor v3.4 package 'clusterProfiler' and 'ReactomePA') using the Reactome Pathway database. $P$-values from the test were adjusted using the BH method and pathways significantly over-represented (adjusted $p$-values < 0.05) in the down-regulated protein group were visualized in Cytoscape (3.40) using the Reactome pathway map (see 'Reactome pathway enrichment analyses' section). For comparison, functional enrichment analyses on proteins downregulated in each treatment condition were also performed using gProfiler's web platform[108] (Version e106_eg53_p16_65fcd97) (Supplementary Data 3).

## Statistics and reproducibility

Unless as specified above under Protein identification and Bioinformatics analyses, statistics were performed, and graphs were generated, using Prism 9 software (GraphPad). Statistical significance was determined using Student's $t$-tests or ANOVA, as indicated. Means and standard errors were reported for all results unless otherwise specified. Effects achieving 95% confidence interval (i.e. $p < 0.05$) were interpreted as statistically significant. No statistical methods were used to pre-determine sample sizes, but these are similar to those generally employed in the field. No samples were excluded from any analysis unless explicitly stated. Statistical analyses and results are also provided in the Data source along with the raw data.

## Reporting summary

Further information on research design is available in the Nature Portfolio Reporting Summary linked to this article.

## Data availability

The source data underlying all main and supplementary figures are provided with this paper as a Source Data file. Datasets and analytics associated with epichaperomics and proteomics analyses are available in the Supplementary Information as Supplementary Data 1 through 3. LC-MS data (i.e. proteomics and epichaperomics raw mass spectrometry data, peak lists, and results) that support the findings of this study are deposited in MassIVE and can be retrieved with the accession code MSV000087540 and the ProteomeXchange Consortium identifier PXD042262. Cytoscape files have been deposited in Zenodo [https://doi.org/10.5281/zenodo.7433980][110]. The PPI data were obtained from Bio-Grid (https://thebiogrid.org/) and IntAct (https://www.ebi.ac.uk/intact/home). Protein sequences (FASTA files) were obtained from UniProt (https://www.uniprot.org/). Protein pathway and functional mapping information was derived from Reactome Pathway (https://reactome.org/) and GeneOntology (http://geneontology.org/) databases. Source data are provided with this paper.

## Code availability

The R script for our analytics pipeline was deposited into Zenodo [https://doi.org/10.5281/zenodo.7416220][111]. It contains the methods and code used to analyse the epichaperomics datasets.

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

## Acknowledgements

This work was supported by the NIH (R01 CA172546, P01 CA186866, R56 AG061869, R01 AG067598, R01 AG074004, R56 AG072599, P30 CA08748), Mr. William H. and Mrs. Alice Goodwin and the Commonwealth Foundation for Cancer Research and the Experimental Therapeutics Center of MSKCC. S.S. would like to acknowledge funding support from BrightFocus Foundation (Award ID: A2022020F). J.C.Y. held a Canada Research Chair in Molecular Chaperones. We would like to thank Jeffrey J. Li for helpful discussions. We thank David Scheinberg and Christina Leslie for a critical reading of the manuscript and/or for suggestions on experimental design and data interpretation, and the Molecular cytology core for assistance in imaging acquisition and analysis.

## Author contributions

A.R. performed the probe validation studies and the epichaperomics experiments. C.X. performed the mechanistic studies on mitotic regulators. C.S.D. performed chemical synthesis, compound identity and purity evaluations as well as mitotic phenotype analyses by IF. S.J. performed analyses by flow cytometry. Y.P. performed biochemical studies in reconstituted systems. A.R.S., S.B., S.Me., P.Y., C.Y., T.R., P.P., J.A., M.L.A. and H.E-B. performed experiments. L.S., Y.K., T.T., Y.P., S.Mo. and S.S. provided reagents. M.L.G., J.C.Y., M-F.B.T., K.M.-T., H.E-B., T.A.N., G.C. and S.D.G. participated in the design and analysis of various experiments. H.E.-B. and T.A.N. performed the mass spectrometry sample preparation and protein identification. A.C. and M.L.A. performed evaluation of breast cancer explants. T.F. performed statistical analyses. T.W. and A.A. performed statistical analyses, the bioinformatics analyses and participated in figure design. J.C.Y. designed, analysed and wrote the section on biochemical studies in reconstituted systems. J.C.Y., M.-F.B.T., H.E.-B., T.A.N., and S.D.G. participated in manuscript editing. G.C. developed the concept and wrote the paper.

## Competing interests

Memorial Sloan Kettering Cancer Center holds the intellectual rights to this portfolio. Samus Therapeutics Inc, of which G.C. has partial ownership, and is a member of its scientific advisory board, has licensed the epichaperome portfolio. G.C., A.R. and T.T. are inventors on the licensed intellectual property. All other authors declare no competing interests.

## Additional information

[1]Chemical Biology Program, Memorial Sloan Kettering Cancer Center, New York, NY 10065, USA. [2]Department of Biochemistry, Groupe de Recherche Axé sur la Structure des Protéines, McGill University, Montreal, QC H3G 0B1, Canada. [3]Department of Pathology, Memorial Sloan Kettering Cancer Center, New York, NY 10065, USA. [4]Department of Medicine, Division of Solid Tumors, Memorial Sloan Kettering Cancer Center, New York, NY 10065, USA. [5]Department of Medicine, Division of Hematology Oncology, Weill Cornell Medicine, New York, NY 10065, USA. [6]Department of Epidemiology and Biostatistics, Memorial Sloan Kettering Cancer Center, New York, NY 10065, USA. [7]Departments of Psychiatry, Neuroscience & Physiology & the NYU Neuroscience Institute, NYU Grossman School of Medicine, New York, NY 10016, USA. [8]Center for Dementia Research, Nathan Kline Institute, Orangeburg, NY 10962, USA. [9]Department of Neuroscience and Physiology and Neuroscience Institute, NYU Grossman School of Medicine, New York, NY 10016, USA. [10]Cell Biology Program, Memorial Sloan Kettering Cancer Center, New York, NY 10065, USA. [11]Present address: Changhai Hospital, Second Military Medical University, Shanghai, China. [12]Present address: Maimonides Medical Center, Brooklyn, NY, USA. [13]Present address: Rowan University, Glassboro, NJ, USA. [14]These authors contributed equally: Anna Rodina, Chao Xu, Chander S. Digwal, Suhasini Joshi, Yogita Patel. [15]These authors jointly supervised this work: Tony Taldone, Thomas A. Neubert, Katia Manova-Todorova, Meng-Fu Bryan Tsou, Jason C. Young, Tai Wang, Gabriela Chiosis. [16]Deceased: Jason C. Young. ✉e-mail: wangtaifr@gmail.com; chiosisg@mskcc.org

