## [Peer review file · Nature Communications]

REVIEWER COMMENTS

Reviewer #1 (Remarks to the Author): Expert in HSP70 chaperones in cancer, genetics and signalling

This manuscript is from the group that discovered the epichaperome, which is loosely defined as long-lived high molecular weight protein-protein complexes that exists in cancer cells and which governs sensitivity to chaperone inhibitors. In this work Chiosis and colleagues uncover the HSP70 (Hsc70 and HSP70) governed epichaperome, using biotinylated versions of an N terminal HSP70 inhibitor called YK5 compared to an inactive compound. The authors use CETSA to confirm that their YK compounds pull down other associated chaperones in certain cancer cells (ie, the epichaperome). Figures 4 and 5 get to the heart of the matter: the authors probe the protein composition, over time, of the epichaperome, using mass spec, and notably find considerable overlap between YK5B beads and PU beads (an HSP90 binder). Interestingly, among the PPI pathways identified, mitotic cell cycle emerges as significantly impacted, and the authors follow up on this finding by chronicling sensitivity of epichaperome-containing cells treated with YK compounds for mitotic block and localization of NuMa.

Overall the manuscript is well written, very interesting, and it establishes a new paradigm that will be of significant interest to the readership of Nature Comms. The experiments are well-controlled and the figures are clear and compelling. The topic matter is novel and extremely important. Addressing the following minor concerns would improve the manuscript:

1. The manuscript focuses on cancer, and is a compelling systems biology tour de force. But the potential for cancer therapy with YK5 is not really addressed. Moreover, the experiments are heavily reliant on MDA-MB-468 cells. The authors could address both issues with some simple synergy analyses in a number of epichaperone-containing lines using YK plus mitotic poisons like AurK or PLK inhibitors. While there is no a priori reason to expect these compounds will synergize in clonogenic survival assays, such an assay remains low hanging and clinically important 'fruit'. The authors could perform this assay combining YK5 plus AurK or PLK inhibitors in 3 or 4 other epichaperone-containing lines and enhance the broad applicability of this potentially important clinical finding. Frankly, even if the authors were to argue that YK5 is not even close to approaching the clinic, investigating this area would still establish an important paradigm.
2. I feel that the authors have somewhat de-emphasized the literature on HSP70s and mitosis, and that they should highlight some of the following manuscripts a bit more in their discussion:

Sampson J, O'Regan L, Dyer MJS, Bayliss R, Fry AM. Hsp72 and Nek6 Cooperate to Cluster Amplified Centrosomes in Cancer Cells. *Cancer Res.* 2017 Sep 15;77(18):4785-4796. doi: 10.1158/0008-5472.CAN-16-3233. Epub 2017 Jul 18. PMID: 28720575.

Mukherjee M, Sabir S, O'Regan L, Sampson J, Richards MW, Huguenin-Dezot N, Ault JR, Chin JW, Zhuravleva A, Fry AM, Bayliss R. Mitotic phosphorylation regulates Hsp72 spindle localization by uncoupling ATP binding from substrate release. *Sci Signal*. 2018 Aug 14;11(543):eaao2464. doi: 10.1126/scisignal.aao2464. PMID: 30108182; PMCID: PMC6166782.

Fang CT, Kuo HH, Hsu SC, Yih LH. HSP70 is required for the proper assembly of pericentriolar material and function of mitotic centrosomes. *Cell Div*. 2019 May 10;14:4. doi: 10.1186/s13008-019-0047-7. PMID: 31110557; PMCID: PMC6511203.

As well as data that an HSP70 inhibitor can inhibit the APC:

Balaburski GM, Leu JI, Beeharry N, Hayik S, Andrade MD, Zhang G, Herlyn M, Villanueva J, Dunbrack RL Jr, Yen T, George DL, Murphy ME. A modified HSP70 inhibitor shows broad activity as an anticancer agent. *Mol Cancer Res*. 2013 Mar;11(3):219-29.

3. Page 5 line 2: sentence is very confusing, perhaps consider deleting the word "of"

Reviewer #2 (Remarks to the Author): Expert in protein networks, signalling networks, and bioinformatics

Rodina et al. employ biochemical and system levels analysis to evaluate changes in protein-protein interaction driven by epichaperome. Also, and importantly, using epiHSP70s epichaperomics, they provide proof of principle that the connectivity of mitotic spindle formation complex is rewired in cancer cells in a context-specific manner. Overall, the methods and results of this paper are well done and straightforward to follow. The experiments are chronologically conducted to address the grey areas of each experimental result in the subsequent experiments. The study is captivating and should be of interest and relevance to the broader cancer research and systems pharmacology community.

Comments

The results on the regulation of mitosis in a context-dependent manner by epiHSP70s very interesting. However, the findings should also be discussed, considering that the PPI interactions are context-

specific. Concretely, the authors need to state in the discussion that their results accurately represent the cell under study and may not be generalised to other cancer cell types with similar or dissimilar epichaperome profiles to MDA-MB-468 cell lines. For example, lines 2 and 3 on page 12 ("... and thus discover a context the specific mechanism of how the fitness of the mitotic protein pathways is augmented in cancer) should be re-written to show that the results are based on the cancer cell line under study and not "in cancer". The authors can only make the "in cancer" claim once they have validated their findings across many cancer cell lines and primary tumours of different tissues of origin.

Page 4 on line 1: "...and thus epiHSP70s are a small amount when compared to the abundant HSP70s (ref. 17 and Fig. 1a)." Could the authors give the actual number or relative percentage in place of "small amount"?

Page 4 on lines 3-6: Could the authors split the sentence in two to clarify the message?

Page 6 on lines 26 to 29: "Importantly, the extent by which proteins were incorporated into PPIs was independent of the overall expression of each protein within the specific cellular context of MDA-MB-468 cells". Could the authors explain and discuss this result in detail? The interaction of cellular proteins is stoichiometric, i.e., if one protein A interacts with two protein Bs to form the AB₂ complex, then if protein A is not present, so will the AB₂ complex. Please also refer to the next comment.

On page 7, on lines 10 to 13, the authors state that "To provide support for this postulate, we compared pathway enrichment analyses performed on proteins of Grade A, p-value < 0.1, vs Grade A+B, p-value < 0.25, vs Grade A+B+C, p-value < 0.5, to find more lenient statistical thresholds used for interactor identification did not skew biological interpretations". Unfortunately, while the results are straightforward, the authors also did not provide a rationale for their conclusion or discuss the results clearly. Their biological interpretation of pathway analysis results is accurate, but not in terms of biochemistry, e.g., a kinase will only phosphorylate a substrate that is present. Could the authors attempt to reconcile the conflict between biochemistry and pathway enrichment in their discussion?

The authors show that EpiHSP70s inhibition blocked cells in mitosis. However, they have not categorically made a statement on the role(s) of EpiHSP70s in mitosis. Do the authors mean that EpiHSP70s control mitosis in some but not all cancer cells? If so, then what would be the implications of applying their technology on cancer therapeutics, which is the ultimate goal?

Page 10 on lines 1 to 4: "These multiple lines of evidence confirm that the identified mitotic proteins are specific epiHSP70s interactors, and thus involved in the rewiring of mitotic PPI networks in the context of epiHSP70s-positive cancer cells". The sentence is an example of the authors overgeneralising their findings. They studied only a few cell lines; thus the "in the context of epiHSP70s-positive cancer cells"

claim can not be made concretely. Could the authors re-write the manuscript while avoiding widespread over-generalisation in the results and discussion sections?

Reviewer #3 (Remarks to the Author): Expert in HSP70 and other chaperones, proteomics, and biochemistry

Rodina and colleagues present an application of chemical probes for HSP70 to the affinity purification of HSP70-containing epichaperome structures (epiHSP70s). They demonstrate the preference of these probes for epiHSP70s via a combination of immunodepletion and thermal shift assays, then go on to apply this method to examine the role of epichaperomes during cell cycle progression in the MDA-MB-468 cancer cell line. They establish an integral role for epiHSP70s in mitosis in these cells. The study presents a novel and relevant application of existing HSP70 probes and is thoroughly supported by a wealth of supplemental data. The ability to profile epichaperome composition via this method is highly applicable to a wide range of disease pathologies. Thus, I believe this manuscript will be of interest to a broad readership, given the following concerns are addressed:

Major concerns:

Validating the specificity of YKs for epiHSP70s relies in large part on the CETSA results presented in Figure 2. However, incomplete quantitative information for panel B/C make this result challenging to interpret. Similarly, a description of the authors' interpretation of the CETSA result presented in Figure 2 for the control proteins HSP60 (panel B) and GAPDH (panel B, D, F) is lacking.

1. All qualitative blots presented in panel B should be quantified for all biological replicates.
2. HSP60 is listed as a control protein, however it appears to experience some differential changes in solubility in the presence of LSI at high temperatures. Was this expected behaviour for a control protein?
3. Conversely, GAPDH appears to have little to no change in solubility over the entire tested temperature range, which is at odds with published CETSA data for this protein (e.g. Delport & Hewer, Sci Rep. 2022). Please describe how the behaviour of this protein relates to the expected outcome for a control protein.

The assertion that YKs 'trap or stabilise' epiHSP70s (pg 5 line 32 - 36) does not appear well supported by the presented data. 'Stabilised' seems to imply pre-existing structures that would otherwise be dynamically dismantled without YKs. The total cellular pool of HSPs remains unchanged, however this does not rule out the induction of de novo construction of epichaperome structures from existing cellular HSP components. There is a 10-fold increase in epiHSP90 (Fig. 3C) after 2 h incubation while the HSP70-associated HSP90 only increases ~5 fold (Fig. 3D) suggesting YKs interaction with epiHSP70 cannot account for the entire increase in HSP90-positive epichaperomes. Evidence of apoptosis (Figure S4A) would also suggest some cellular stress results from YK incubation.

4. Please clarify what is meant by 'trapped or stabilised' and the evidence supporting this in preference to YK-induced epichaperome assembly.

5. Please discuss the rationale for the 4 h incubation when performing epichaperomics given the induction of apoptosis and switch to disassembly of epiHSP70s between 2 and 3 h incubation with YKs.

The number of proteins identified by proteomics using the two complementary probes PU and YK is an integral aspect of this study (currently Supplementary Figure 5B). The proportion of proteins found in common, versus unique to a single epichaperome subtype, is crucial to justifying why the HSP70-targeted method is required in addition to the established HSP90 method. However, the result is minimally discussed beyond the expected significant overlap in the protein pool.

6. Please provide quantitative summary data (e.g. a Venn diagram or equivalent) for the information presented in Supplementary Figure 5B.

7. It appears that the YK-specific interactors are equal to, or outweigh, the common interactors. There appear to be very few unique HSP90-only interactors. Is this expected, and does this suggest that YK5 provides a broader picture of the cellular epichaperome state?

The authors present the manuscript as a blueprint for conducting epichaperomics studies, however, large portions of the computational analysis have not been shared.

8. It is recommended to follow the guidelines for the provision of research data code via an open-access repository to enable easy uptake of the method: <https://www.nature.com/nature-portfolio/editorial-policies/reporting-standards#availability-of-computer-code>

Minor concerns:

1. Overall balance of data and method schematics makes some figures challenging to digest. Consider reducing/removing/relocating the method schema presented in Figure 1D, E, Figure 2F, Figure 3C, D.

2. Descriptive language should be preferred e.g. "YK5-B outperformed the HSP70s antibody" (pg 6 line 15); in what way did the probe outperform the antibody, amount of HSP70 captured? Number of interactors?

3. Page 6 Line 20 should refer to Figure 4D.

4. Page 8 Line 40-42 - the outcome and interpretation of each of the listed experiments should be described in more detail.

5. Page 11 Line 19 - Sentence clarity: 'Both low abundance proteins and weak interactions are detected successfully, increasing the likelihood to correctly assign the context-specific [?] of a given protein.'

6. Page 11 Line 23 - Sentence clarity: "Accordingly, we foresee implementation of the YK-probes, along [WITH?] the reported PU-probes"

7. Figure 7D: Legend describes n=5, but it is not clear what is displayed on the graph. Likely the mean, and if so please include an estimate of the error across replicates (or indicate that it is within the size of the displayed symbols)

8. Some method details are included in figure schematics which are not present in the Methods text. In addition, there is a general lack of detail in the methods text that would hamper attempts to replicate the study. Please ensure the following information is available for all experimental methods:

- Probe concentrations (especially for epichaperomics, pg 21 line 29).
- Length of incubation
- Number of cells assayed

9. The implications of targeting individual members (i.e. epiHSP70s or epiHSP90s) among the large network of potential epichaperome components should be discussed. Are there expected to be

epichaperome structures not captured by either of the existing probe sets i.e. how generalisable is this method?

10. A code availability statement is required for custom scripted analyses. Regardless of whether the codebase is made open access as requested above, a statement detailing the availability (or lack thereof) should be included.

Reviewer #4 (Remarks to the Author): Expert in HSPs and modifications in cancer, molecular biology, and mitotic regulation

Epichaperomes represent a fraction of the total chaperone pools (i.e. 5-35%, depending on the cancer cell line) and thus epiHsp70 are small amount when compared to the abundant Hsp70 levels. Certain diseases including cancers appear to be depending on these epichaperomes for their survival and growth. However even within cancer types there are different levels of epichaperomes. For example, breast cancer MDA-MB-468 and pancreatic cancer ASPC1 cells have comparable Hsp70 levels and of other epiHsp70 component chaperones but are differentiated by their epichaperomes content, with MDA23 MB-468 being epiHSP70s-high and ASPC1 epiHSP70s-low. Here, the authors used their previously reported and also refined chemical probes (YK-type) for targeting and isolating epiHsp70 proteins from cancer cells. Their analyses has identified thousands of proteins that primarily involved in mitotic checkpoint pathway in these cells. They further suggest that epiHsp70 proteins bind not to individual mitotic proteins per se but rather alter the complexation of such proteins. Thus, epiHSP70 may exert its effect on mitosis by altering the complexation of mitotic regulator proteins in such a way that the fitness of mitotic processes is increased.

The amount of data presented here is insane! This statement is meant as a positive note and in support of this study and not a criticism. The experiments and results are very well planned, conducted, analyzed and presented. They include appropriate controls and statistical analysis. It is for these particular reasons that it would be insulting to ask the authors to provide anymore additional data. However, there are some minor issues and statements that the authors need to fix.

Minor comments:

- Abstract: The abstract does not summarize the presented findings and therefore it needs extensive rewriting. I have no idea why the authors felt the need to start their abstract with the statement "There is a penury of regents".

- Introduction (unofficial section): As proof-of-principle, for validation, we apply the epiHSP70s epichaperomics to cancer cells to identify mitotic regulator proteins whose connectivity is rewired to alter mitotic spindle formation and discover a context-specific mechanism of how the fitness of mitotic protein pathways is enhanced in cancer. The role of molecular chaperone in mitotic checkpoint regulation in cancer has been reported previously (Woodford et al 2016 Cell Rep) please cite this work. Also please speculate on the role of epiHSP70 and aneuploidy.

- Figure 3B. Immediately after the addition of YKs to the cells, we observed what appeared as a transient increase in epichaperome signal on Native-PAGE (Fig. 3c) but no change in the total chaperone levels on SDS-PAGE (Fig. 3b). The last of part of this statement is inaccurate since there is a clear increase in HSC70 levels (lane 2) on SDS-PAGE. This increase appears to be transient.

- The HSP70 proteins pool captured by YK5-B beads is the fraction corresponding to epiHsp70 proteins. Supplementary Fig.3c. i.e Epichaperomes represent a fraction of the total chaperone pools (5-35%, depending on the cancer cell line). My question is if the authors immunoprecipitate the Hsp70 pool that does not bind to the YK5-B (i.e fraction 6), do they see this Hsp70 pool form a complex with clients & co-chaperones. Please do not this experiment for this paper.

- Discussion: There is a minimum discussion of the data here. I urge the authors to rewrite this section but more concise.

Just as an example; "Direct measurement of how thousands of proteins change interaction partners in disease, and moreover, understanding the functional impact of such connectivity changes, is technically challenging using current interrogative approaches"

We sincerely thank the referees for their insightful suggestions and detailed review of the manuscript. With this resubmission, and in response to their suggestions, we have made revisions to both text and figures, as detailed below, that we believe have substantially improved our manuscript. We are also grateful all four referees have found numerous strengths in this work and deemed this manuscript worthy of publication in Nature Communications, pending revisions.

REVIEWER COMMENTS AND POINT-BY-POINT RESPONSE

Reviewer #1 (Remarks to the Author): Expert in HSP70 chaperones in cancer, genetics and signalling

This manuscript is from the group that discovered the epichaperome, which is loosely defined as long-lived high molecular weight protein-protein complexes that exists in cancer cells and which governs sensitivity to chaperone inhibitors. In this work Chiosis and colleagues uncover the HSP70 (Hsc70 and HSP70) governed epichaperome, using biotinylated versions of an N terminal HSP70 inhibitor called YK5 compared to an inactive compound. The authors use CETSA to confirm that their YK compounds pull down other associated chaperones in certain cancer cells (ie, the epichaperome). Figures 4 and 5 get to the heart of the matter: the authors probe the protein composition, over time, of the epichaperome, using mass spec, and notably find considerable overlap between YK5B beads and PU beads (an HSP90 binder). Interestingly, among the PPI pathways identified, mitotic cell cycle emerges as significantly impacted, and the authors follow up on this finding by chronicling sensitivity of epichaperome-containing cells treated with YK compounds for mitotic block and localization of NuMa.

Overall the manuscript is well written, very interesting, and it establishes a new paradigm that will be of significant interest to the readership of Nature Comms. The experiments are well-controlled and the figures are clear and compelling. The topic matter is novel and extremely important. Addressing the following minor concerns would improve the manuscript:

R1.1. The manuscript focuses on cancer, and is a compelling systems biology tour de force. But the potential for cancer therapy with YK5 is not really addressed. Moreover, the experiments are heavily reliant on MDA-MB-468 cells. The authors could address both issues with some simple synergy analyses in a number of epichaperone-containing lines using YK plus mitotic poisons like AurK or PLK inhibitors. While there is no a priori reason to expect these compounds will synergize in clonogenic survival assays, such an assay remains low hanging and clinically important 'fruit'. The authors could perform this assay combining YK5 plus AurK or PLK inhibitors in 3 or 4 other epichaperone-containing lines and enhance the broad applicability of this potentially important clinical finding. Frankly, even if the authors were to argue that YK5 is not even close to approaching the clinic, investigating this area would still establish an important paradigm.

Response 1.1: We thank the Reviewer for this suggestion. Addressing the potential for cancer therapy of the epiHSP70s agents is indeed an important topic, but also a topic that deserves full investigation in a standalone study, which is beyond the scope of the present report, and we aim to report in due time. However, we agree that the addition of experiments to address the biological activity of YKs in more cellular models is of relevance here and would strengthen our study. We therefore added experiments addressing the anti-cancer activity of YKs in a panel of 73 cancer cell lines that encompass 9 tumor types as well as in 12 patient-derived breast cancer explants. These data are presented in the **New Fig. 6**.

R1.2. I feel that the authors have somewhat de-emphasized the literature on HSP70s and mitosis, and that they should highlight some of the following manuscripts a bit more in their discussion:

Sampson J, O'Regan L, Dyer MJS, Bayliss R, Fry AM. Hsp72 and Nek6 Cooperate to Cluster Amplified Centrosomes in Cancer Cells. *Cancer Res.* 2017 Sep 15;77(18):4785-4796. doi: 10.1158/0008-5472.CAN-16-3233. Epub 2017 Jul 18. PMID: 28720575.

Mukherjee M, Sabir S, O'Regan L, Sampson J, Richards MW, Huguenin-Dezot N, Ault JR, Chin JW, Zhuravleva A, Fry AM, Bayliss R. Mitotic phosphorylation regulates Hsp72 spindle localization by uncoupling

ATP binding from substrate release. *Sci Signal*. 2018 Aug 14;11(543):eaao2464. doi: 10.1126/scisignal.aao2464. PMID: 30108182; PMCID: PMC6166782.

Fang CT, Kuo HH, Hsu SC, Yih LH. HSP70 is required for the proper assembly of pericentriolar material and function of mitotic centrosomes. *Cell Div*. 2019 May 10;14:4. doi: 10.1186/s13008-019-0047-7. PMID: 31110557; PMCID: PMC6511203.

As well as data that an HSP70 inhibitor can inhibit the APC:

Balaburski GM, Leu JI, Beeharry N, Hayik S, Andrade MD, Zhang G, Herlyn M, Villanueva J, Dunbrack RL Jr, Yen T, George DL, Murphy ME. A modified HSP70 inhibitor shows broad activity as an anticancer agent. *Mol Cancer Res*. 2013 Mar;11(3):219-29.

Response 1.2: Thank you, we added the references as suggested to both the Results, page 9 and Discussion.

R1.3. Page 5 line 2: sentence is very confusing, perhaps consider deleting the word "of"

Response 1.3: The sentence was corrected as suggested.

Reviewer #2 (Remarks to the Author): Expert in protein networks, signalling networks, and bioinformatics

Rodina et al. employ biochemical and system levels analysis to evaluate changes in protein-protein interaction driven by epichaperome. Also, and importantly, using epiHSP70s epichaperomics, they provide proof of principle that the connectivity of mitotic spindle formation complex is rewired in cancer cells in a context-specific manner. Overall, the methods and results of this paper are well done and straightforward to follow. The experiments are chronologically conducted to address the grey areas of each experimental result in the subsequent experiments. The study is captivating and should be of interest and relevance to the broader cancer research and systems pharmacology community.

Comments

R2.1. The results on the regulation of mitosis in a context-dependent manner by epiHSP70s is very interesting. However, the findings should also be discussed, considering that the PPI interactions are context-specific. Concretely, the authors need to state in the discussion that their results accurately represent the cell under study and may not be generalised to other cancer cell types with similar or dissimilar epichaperome profiles to MDA-MB-468 cell lines. For example, lines 2 and 3 on page 12 ("... and thus discover a context the specific mechanism of how the fitness of the mitotic protein pathways is augmented in cancer) should be re-written to show that the results are based on the cancer cell line under study and not "in cancer". The authors can only make the "in cancer" claim once they have validated their findings across many cancer cell lines and primary tumours of different tissues of origin.

Response 2.1. Thank you for this comment. Context-dependence refers to cancer cells positive for epiHSP70s, keeping in mind that not all cancer cells express HSP70s-containing epichaperomes. Since non-transformed cells lack epichaperomes, we refer to this mechanism as a cancer phenomenon in epiHSP70s positive cancer cells, rather than a general mechanism of mitosis regulation. We verified mechanistic results from unbiased large-scale analyses obtained in epiHSP70s positive MDA-MB-468 breast cancer in a panel of epichaperome positive cancer cells (e.g., OCI-LY1, lymphoma; MiaPaCa2, pancreatic cancer; and HeLa, cervical cancer), and showed the mechanism does not apply to epichaperome-low/negative cells (e.g., ASPC1, pancreatic cancer; MCF7, breast cancer and non-transformed but proliferating, CCD-18 fibroblasts). Thus, we believe this mechanism is context specific for epiHSP70 positive cancer cells, rather than solely for MDA-MB-468 cells, as supported by several positive- and negative-control cell lines. We added a paragraph to the text to summarize the above (page 8). In response to this critique, we also clarified our definition of context-dependence to epiHSP70s positive cancer cells (Discussion).

R2.2. Page 4 on line 1: "...and thus epiHSP70s are a small amount when compared to the abundant HSP70s (ref. 17 and Fig. 1a)." Could the authors give the actual number or relative percentage in place of "small amount"?

Response 2.2. We specified the relative percentage, as suggested.

R2.3. Page 4 on lines 3-6: Could the authors split the sentence in two to clarify the message?

Response 2.3. We revised the sentence, as suggested, into two sentences to clarify our findings.

R2.4. Page 6 on lines 26 to 29: "Importantly, the extent by which proteins were incorporated into PPIs was independent of the overall expression of each protein within the specific cellular context of MDA-MB-468 cells". Could the authors explain and discuss this result in detail? The interaction of cellular proteins is stoichiometric, i.e., if one protein A interacts with two protein Bs to form the AB₂ complex, then if protein A is not present, so will the AB₂ complex. Please also refer to the next comment.

On page 7, on lines 10 to 13, the authors state that "To provide support for this postulate, we compared pathway enrichment analyses performed on proteins of Grade A, p-value < 0.1, vs Grade A+B, p-value < 0.25, vs Grade A+B+C, p-value < 0.5, to find more lenient statistical thresholds used for interactor identification did not skew biological interpretations". Unfortunately, while the results are straightforward, the authors also did not provide a rationale for their conclusion or discuss the results clearly. Their biological interpretation of pathway analysis results is accurate, but not in terms of biochemistry, e.g., a kinase will only phosphorylate a substrate that is present. Could the authors attempt to reconcile the conflict between biochemistry and pathway enrichment in their discussion?

Response 2.4. Thank you for this salient comment. Proteins are present in a cell as heterogenous entities shaped by PTMs, cellular localization, conformational states, and other protein-modifying mechanisms. In the scenario of one protein A interacting with two protein Bs to form an AB₂ complex, if protein A is modified by a PTM that changes its conformational state, and its assembly preference for C over B, or if protein A translocates to a cellular compartment where C but not B is prevalent, protein A may interact with protein C instead of B, even if the cellular levels of B remain unchanged.

This analogy is relevant to HSP70s. At least two distinct pools of HSP70s exist. HSP70s are detected as part of epichaperomes, and HSP70s are part of dynamic folding complexes. Both pools are found in the studied cancer cells. Importantly, the presence or absence of these two pools is not dictated by the total expression of HSP70s. Although mechanisms of chaperone incorporation into epichaperomes remain unknown for HSP70s, we reported previously (Yan et al. Cell Reports 2020) that PTMs, including aberrant N-glycosylation, may be one protein modifying mechanism responsible for this event.

R2.5. The authors show that EpiHSP70s inhibition blocked cells in mitosis. However, they have not categorically made a statement on the role(s) of EpiHSP70s in mitosis. Do the authors mean that EpiHSP70s control mitosis in some but not all cancer cells? If so, then what would be the implications of applying their technology on cancer therapeutics, which is the ultimate goal?

Response 2.5. Yes, epiHSP70s have emerged as a mechanism of regulation of mitosis in the context of epichaperome positive cancer cells. This was clarified in the text.

Epichaperomes are scaffolds that rewire PPI networks under pathologic stressor conditions. Application of epichaperome disruptors to cancer therapy is governed by complex, PPI-network related principles, as reported (Rodina et al. Nature 2016; Joshi et al. Comms Biology 2021; Ginsberg et al. Trends Pharm Sci 2022). The higher the epichaperome levels, the higher the number of proteins being negatively impacted, translating to a higher number of PPIs being rewired. Thus disrupting epichaperomes is detrimental to cancer cells in a manner directly proportional to epichaperome levels. The higher the epichaperome levels the higher the vulnerability of cancer cells to epichaperome therapy as a higher sector of the PPI network map is affected upon epichaperome disruption (Rodina et al. Nature 2016; Joshi et al. Comms Biology 2021). Thus we foresee implementation of YKs to cancer based on these principles developed for epichaperome cancer therapy (Pillarsetty et al. Cancer Cell 2019; Ginsberg et al, Trends Pharm Sci 2022). This concept was clarified in the text (Discussion).

R2.6. Page 10 on lines 1 to 4: "These multiple lines of evidence confirm that the identified mitotic proteins are specific epiHSP70s interactors, and thus involved in the rewiring of mitotic PPI networks in the context of epiHSP70s-positive cancer cells". The sentence is an example of the authors overgeneralising their findings. They studied only a few cell lines; thus the "in the context of epiHSP70s-positive cancer cells" claim can not be made concretely. Could the authors re-write the manuscript while avoiding widespread over-generalisation in the results and discussion sections?

Response 2.6. The sentence was revised to state "in the context of tested epiHSP70s-positive cancer cells".

Reviewer #3 (Remarks to the Author): Expert in HSP70 and other chaperones, proteomics, and biochemistry

Rodina and colleagues present an application of chemical probes for HSP70 to the affinity purification of HSP70-containing epichaperome structures (epiHSP70s). They demonstrate the preference of these probes for epiHSP70s via a combination of immunodepletion and thermal shift assays, then go on to apply this method to examine the role of epichaperomes during cell cycle progression in the MDA-MB-468 cancer cell line. They establish an integral role for epiHSP70s in mitosis in these cells. The study presents a novel and relevant application of existing HSP70 probes and is thoroughly supported by a wealth of supplemental data. The ability to profile epichaperome composition via this method is highly applicable to a wide range of disease pathologies. Thus, I believe this manuscript will be of interest to a broad readership, given the following concerns are addressed:

Major concerns:

R3.1. Validating the specificity of YKs for epiHSP70s relies in large part on the CETSA results presented in Figure 2. However, incomplete quantitative information for panel B/C make this result challenging to interpret. Similarly, a description of the authors' interpretation of the CETSA result presented in Figure 2 for the control proteins HSP60 (panel B) and GAPDH (panel B, D, F) is lacking.

1. All qualitative blots presented in panel B should be quantified for all biological replicates.
2. HSP60 is listed as a control protein, however it appears to experience some differential changes in solubility in the presence of LSI at high temperatures. Was this expected behaviour for a control protein?
3. Conversely, GAPDH appears to have little to no change in solubility over the entire tested temperature range, which is at odds with published CETSA data for this protein (e.g. Delport & Hewer, Sci Rep. 2022). Please describe how the behaviour of this protein relates to the expected outcome for a control protein.

Response R3.1.

1. Thank you for this comment. HSC70 and HSC90 are core components of epichaperomes, i.e. they are present, together, tightly bound, in a major subset of epichaperomes in MDA-MB-468. Thus, with regards to validating the specificity of YKs for epiHSP70s over HSP70s, the most important controls in **Fig. 2**, for CETSA are to show that both HSP70s and HSP90s, rather than only HSP70s, are sensitive to the YK probes in epiHSP70s positive cells but not in epiHSP70s negative but HSP70s positive cells. Please refer to:

- i.* Vehicle treated lysates versus epiHSP70s agents treated lysates in epichaperome high cancer cells for core epichaperome components (eg. HSC70 and HSP90, see schematic **Fig. 2a**) (data graphed and presented in **Fig. 2c**).
- ii.* Data comparing and contrasting Vehicle versus epiHSP70s agents in epiHSP70s negative/low lysates versus epiHSP70s high lysates (data graphed and presented in **Fig. 2f**).

Nevertheless we agree that more support towards the preference of YKs for epiHSP70s vs HSP70s is welcome, and thus we quantified additional components and added the data to Supplementary Figure 4a, as suggested.

2. For HSP60, changes occur at high temperatures, and after epichaperome components moved into the insoluble fraction. At these high temperatures, protein misfolding and other complex and indirect mechanisms may be at play. Elucidating such mechanisms is beyond the scope of this manuscript. To avoid confusion, we removed this protein. The key controls for specificity in the context of CETSA are those detailed in Point 1.

3. We find the thermal stability of GAPDH is dependent on the cell line being analyzed. The publication referred to by Reviewer #3 used HEK293 cells, whereas we employed MDA-MB-468 cells. We found a few publications that report GAPDH is stable at the temperature and assay protocol we use for CETSA. For example: PMC6151341, **Fig. 5** in HCT116 cancer cells shows stability up to 59 °C (higher not tested); PMID: 33805945, Figure 3 in SK-HEP-1 cancer cells shows stability up to the 64 °C (higher not tested). Thus the stability of GAPDH in MDA-MB-468 cells is commensurate with other cancer cells in published protocols. We clarified in **Fig. 2b** that GAPDH is a protein loading control in MDA-MB-468 cells, and not a specificity control.

More to the point of “specificity of YKs for epiHSP70s relies in large part on the CETSA results presented in Figure 2”, we would like to reiterate here that CETSA is one of the many biochemical and functional validation experiments used here, with data shown in almost every figure and supplementary figure, to address specificity.

R3.4+5. The assertion that YKs ‘trap or stabilise’ epiHSP70s (pg 5 line 32 - 36) does not appear well supported by the presented data. ‘Stabilised’ seems to imply pre-existing structures that would otherwise be dynamically dismantled without YKs. The total cellular pool of HSPs remains unchanged, however this does not rule out the induction of de novo construction of epichaperome structures from existing cellular HSP components. There is a 10-fold increase in epiHSP90 (Fig. 3C) after 2 h incubation while the HSP70-associated HSP90 only increases ~5 fold (Fig. 3D) suggesting YKs interaction with epiHSP70 cannot account for the entire increase in HSP90-positive epichaperomes. Evidence of apoptosis (Figure S4A) would also suggest some cellular stress results from YK incubation.

4. Please clarify what is meant by ‘trapped or stabilised’ and the evidence supporting this in preference to YK-induced epichaperome assembly.

5. Please discuss the rationale for the 4 h incubation when performing epichaperomics given the induction of apoptosis and switch to disassembly of epiHSP70s between 2 and 3 h incubation with YKs.

Response R3.4+5: We agree with Reviewer #3 that in cells a complex interplay exists between “preexisting structures” and “structures that form” as an impact of the agent. However, the results in **Fig. 3** need to be viewed in the context of the combined evidence from non-denaturing gels and CETSA, which provide support the structures “exist” in epichaperome positive cells prior to YK addition. These protein complexes are “trapped” by the probe as evidenced by affinity purification and epichaperomics. Epichaperomics performed in cells return an interactome comparable to when the capture is performed in cell homogenates where cellular stress or de novo construction of epichaperomes are not a contributor factor (see Supplementary Fig. 7b). For additional support we performed CETSA with compound added after cells were lysed, thus where the agent can see only “preexisting structures”, to show, that similar to CETSA performed in cells, these epiHSP70s structures are (temporarily) stabilized by the agent (i.e., HSP90 is not directly bound by YKs, yet the thermal stability of HSP90 follows the profile of HSC70, supporting it becomes stabilized by YK because the epichaperome structure becomes stabilized). This new CETSA data was added to Supplementary Fig. 4b.

We also agree with Reviewer #3 that 2-3 h may be better than 4 h in terms of being optimal for capture. Nevertheless, 4 h provides an accurate capture profile, as supported by validation studies in **Figs. 7-10** and the associated Supplementary Figures. We added a sentence to the Methods to suggest users consider, and test in their experimental setting, an incubation window of 1-4 h. We also added to Supplementary Fig. 4d data exploring the effect of incubation time on the capture profile of the probe.

R3.6+7. The number of proteins identified by proteomics using the two complementary probes PU and YK is an integral aspect of this study (currently Supplementary Figure 5B). The proportion of proteins found in common, versus unique to a single epichaperome subtype, is crucial to justifying why the HSP70-targeted method is required in addition to the established HSP90 method. However, the result is minimally discussed beyond the expected significant overlap in the protein pool.

6. Please provide quantitative summary data (e.g. a Venn diagram or equivalent) for the information presented in Supplementary Figure 5B.

7. It appears that the YK-specific interactors are equal to, or outweigh, the common interactors. There appear to be very few unique HSP90-only interactors. Is this expected, and does this suggest that YK5 provides a broader picture of the cellular epichaperome state?

Response R3.6-7. Thank you for this very interesting comment. It is not known whether each epichaperome structure contains both HSP90s and HSP70s, or whether epichaperomes containing HSP90s or HSP70s, along other chaperones, but not together, also exist. From what we know so far, epichaperome composition, and in turn the identity of impacted proteins, are context-dependent. Thus the higher content of YK-interactors as compared to PU-interactors may suggest a higher participation of HSP70s when compared to HSP90s in pathologic epichaperomes, which remains to be validated. We agree with Reviewer #3 that it merits commenting upon, and we added a paragraph to page 6 and to Discussions). We also added a Venn diagram to Supplementary Figure 5b, as suggested.

R3.8. The authors present the manuscript as a blueprint for conducting epichaperomics studies, however, large portions of the computational analysis have not been shared.

8. It is recommended to follow the guidelines for the provision of research data code via an open-access repository to enable easy uptake of the method: <https://www.nature.com/nature-portfolio/editorial-policies/reporting-standards#availability-of-computer-code>

Response R3.8. We deposited the R script for our analytics pipeline into Zenodo. It contains the methods and code used to analyze the epichaperomics datasets.

Minor concerns:

1. Overall balance of data and method schematics makes some figures challenging to digest. Consider reducing/removing/relocating the method schema presented in Figure 1D, E, Figure 2F, Figure 3C, D.

Response: We revised and/or relocated the schematics as suggested.

2. Descriptive language should be preferred e.g. “YK5-B outperformed the HSP70s antibody” (pg 6 line 15); in what way did the probe outperform the antibody, amount of HSP70 captured? Number of interactors?

Response: we clarified it is “amount of captured cargo”.

3. Page 6 Line 20 should refer to Figure 4D.

Response: Thank you. We corrected the typo.

4. Page 8 Line 40-42 - the outcome and interpretation of each of the listed experiments should be described in more detail.

Response: Sentences were added to page 8 as suggested.

5. Page 11 Line 19 - Sentence clarity: 'Both low abundance proteins and weak interactions are detected successfully, increasing the likelihood to correctly assign the context-specific [?] of a given protein.'

Response: Thank you for pointing out the typo. "Function" was added.

6. Page 11 Line 23 - Sentence clarity: "Accordingly, we foresee implementation of the YK-probes, along [WITH?] the reported PU-probes"

Response: Corrected as suggested.

7. Figure 7D: Legend describes n=5, but it is not clear what is displayed on the graph. Likely the mean, and if so please include an estimate of the error across replicates (or indicate that it is within the size of the displayed symbols)

Response: We revised the graph as suggested and fixed an error in the figure legend.

8. Some method details are included in figure schematics which are not present in the Methods text. In addition, there is a general lack of detail in the methods text that would hamper attempts to replicate the study. Please ensure the following information is available for all experimental methods:

- Probe concentrations (especially for epichaperomics, pg 21 line 29).
- Length of incubation
- Number of cells assayed

Response: Methods were detailed as suggested.

9. The implications of targetting individual members (i.e. epiHSP70s or epiHSP90s) among the large network of potential epichaperome components should be discussed. Are there expected to be epichaperome structures not captured by either of the existing probe sets i.e. how generalisable is this method?

Response: We added a sentence to Discussions to address this comment.

10. A code availability statement is required for custom scripted analyses. Regardless of whether the codebase is made open access as requested above, a statement detailing the availability (or lack thereof) should be included.

Response: The statement was added as suggested.

Reviewer #4 (Remarks to the Author): Expert in HSPs and modifications in cancer, molecular biology, and mitotic regulation

Epichaperomes represent a fraction of the total chaperone pools (i.e. 5-35%, depending on the cancer cell line) and thus epiHsp70 are small amount when compared to the abundant Hsp70 levels. Certain diseases including cancers appear to be depending on these epichaperomes for their survival and growth. However even within cancer types there are different levels of epichaperomes. For example, breast cancer MDA-MB-468 and pancreatic cancer ASPC1 cells have comparable Hsp70 levels and of other epiHsp70 component chaperones but are differentiated by their epichaperomes content, with MDA-MB-468 being epiHSP70s-high and ASPC1 epiHSP70s-low. Here, the authors used their previously reported and also refined chemical probes (YK-type) for targeting and isolating epiHsp70 proteins from cancer cells. Their analysis has identified thousands of proteins that are primarily involved in mitotic checkpoint pathway in these cells. They further suggest that epiHsp70 proteins bind not to individual mitotic proteins per se but rather alter the complexation of such proteins. Thus, epiHSP70 may exert its effect on mitosis by altering the complexation of mitotic regulator proteins in such a way that the fitness of mitotic processes is increased.

The amount of data presented here is insane! This statement is meant as a positive note and in support of this study and not a criticism. The experiments and results are very well planned, conducted, analyzed and

presented. They include appropriate controls and statistical analysis. It is for these particular reasons that it would be insulting to ask the authors to provide anymore additional data. However, there are some minor issues and statements that the authors need to fix.

Response: Thank you for appreciating the large body of work and the careful experimental design that went into this manuscript.

Minor comments:

R4.1. Abstract: The abstract does not summarize the presented findings and therefore it needs extensive rewriting. I have no idea why the authors felt the need to start their abstract with the statement “There is a penury of reagents”.

Response R4.1: We have revised the Abstract to highlight the impact of the results in accordance with Reviewer #4’s comments.

R4.2. Introduction (unofficial section): As proof-of-principle, for validation, we apply the epiHSP70s epichaperomics to cancer cells to identify mitotic regulator proteins whose connectivity is rewired to alter mitotic spindle formation and discover a context-specific mechanism of how the fitness of mitotic protein pathways is enhanced in cancer. The role of molecular chaperone in mitotic checkpoint regulation in cancer has been reported previously (Woodford et al 2016 Cell Rep) please cite this work. Also please speculate on the role of epiHSP70 and aneuploidy.

Response R4.2: Thank you for this comment. It provides an intriguing mechanistic link between prior observations on the role of PTMs in HSP90 and HSP70 and our current findings. We added a paragraph and related references to address this link (page 12, Discussion). The suggested link between epiHSP70s and aneuploidy is interesting. Both are mechanisms related to stress maladaptation, possibly favoring the development and selection of aggressive malignant clones in cancer by enabling cells to modulate independent pathways simultaneously and to explore a wide phenotypic landscape. Interdependence of these two mechanisms is plausible considering that several proteins associated with and/or mediating aneuploidy (eg. AURKA, PLK1, BUB3, among others) are epiHSP70s interactors.

R4.3. Figure 3B. Immediately after the addition of YKs to the cells, we observed what appeared as a transient increase in epichaperome signal on Native-PAGE (Fig. 3c) but no change in the total chaperone levels on SDS-PAGE (Fig. 3b). The last of part of this statement is inaccurate since there is a clear increase in HSC70 levels (lane 2) on SDS-PAGE. This increase appears to be transient.

Response R4.3: Please note that both lanes 1 and 7 show baseline HSC70 levels. To avoid confusion, we quantified and graphed the data from several experiments - see revised **Fig. 3b**.

R4.4. The HSP70 proteins pool captured by YK5-B beads is the fraction corresponding to epiHsp70 proteins. Supplementary Fig.3c. i.e Epichaperomes represent a fraction of the total chaperone pools (5-35%, depending on the cancer cell line). My question is if the authors immunoprecipitate the Hsp70 pool that does not bind to the YK5-B (i.e fraction 6), do they see this Hsp70 pool form a complex with clients & co-chaperones. Please do not perform this experiment for this paper.

Response R4.4: Thank you for suggesting this experiment for future studies, we appreciate the input.

R4.5.- Discussion: There is a minimum discussion of the data here. I urge the authors to rewrite this section but more concise.

Just as an example; “Direct measurement of how thousands of proteins change interaction partners in disease, and moreover, understanding the functional impact of such connectivity changes, is technically challenging using current interrogative approaches”

Response R4.5: We revised and focused the Discussion to include greater discourse on the findings and impact of the study.

REVIEWER COMMENTS

Reviewer #1 (Remarks to the Author):

The authors have addressed my very minor concerns.

Reviewer #2 (Remarks to the Author):

All of my comments have been addressed by the authors beyond any reasonable doubt. I want to express my gratitude to the authors for their diligence in producing a well-written and thorough manuscript.

Reviewer #3 (Remarks to the Author):

Rodina and colleagues present an application of chemical probes for HSP70 to the affinity purification of HSP70-containing epichaperome structures (epiHSP70s). I appreciate the authors' substantial effort to address my concerns and those of the other reviewers. The methods are now sufficiently detailed, and the conclusions of the manuscript are now well supported. I am happy to support publication of this work at Nature Communications.

Reviewer #4 (Remarks to the Author):

The authors have made a huge effort to address my and other reviewers' concerns and questions.

This manuscript is ready for publication.

Reviewer #5 (Remarks to the Author): Expert in medicinal chemistry, drug discovery and synthesis

The study by Rodina et al describes the alteration of protein-protein interaction (PPI) networks by “epichaperomes” in cancer cells. The study makes use of small molecule chemical probes covalently binding these epichaperomes, which are long-lived Hsp70-containing scaffolding structures (epiHsp70). The chemical probes were used to trap, enrich, and isolate epichaperome/interactor complexes from proteomes. This pulldown-based epichaperome proteomics platform (referred to as “epichaperomics”) provided systems-level insights into PPI dysfunctions in cancer cell lines. The study presents a wealth of rather complex proteomics data which establishes a link between epichaperomes and the context-dependent reprogramming of mitosis in cancer cells. Overall most of the presented data seems sound and the study is surely of relevance to the general readership of the journal.

The chemical synthesis of the compounds included in the manuscript is described in a conclusive manner in the supplementary information. Compounds were fully characterized by ¹H and ¹³C NMR spectroscopy as well as mass spectrometry and the obtained data seems coherent.

Validation of the epichaperomics chemical probes was performed via CETSA and pulldown. Notably, similar ligands had already been described and characterized in previous studies (see doi 10.1016/j.chembiol.2013.10.008 and 10.1021/cb500256u). While the systems biology part of the manuscript is very strong, the probe validation leaves some open questions. The past has shown that the use of insufficiently validated chemical probes frequently led to the dissemination of invalid conclusions (see 10.1038/nchembio.1867). Therefore, criteria for the validation of chemical probes, which are now widely accepted by the community have been defined (see for example 10.1038/nchembio.1867 and <https://www.chemicalprobes.org/criteria>). Due to their distinct properties, additional criteria should be considered for covalent compounds (see 10.1039/9781839160745-00069 and 10.1016/j.chembiol.2019.09.012). Actually, I see only a part of this implemented here and in the mentioned previous studies. Virtually all probe validation has been done by indirect assays (pulldown, cellular readouts,...) but little is known about target protein binding. This becomes even more important when considering the covalent nature of probe YK5 and its biotinylated analog both targeting Hsp70-Cys267, which confers time-dependent biological properties. I understand that probe validation for PPI targets is more challenging than for enzymes where convenient assay formats and screening panels are available. Nevertheless, I strongly recommend further validation of the probes used, especially with respect to their time-dependent binding behavior. The following point should be considered:

- It seems that IC50s of YK5 and related compounds were only determined in cellular systems. Binding affinities to the target proteins remain obscure. Investigations with isolated proteins or in reconstituted

systems would add value and increase compliance with common standards for chemical probe validation.

- YK5 and its biotinylated analog act via a covalent mechanism meaning their binding is time-dependent. Determination and discussion of aspects related binding kinetics of these compounds are completely absent in the manuscript. What is the reversible binding component (K_i) of YK5/YK5-B and at what rate does the covalent modification occur (kinact)? I strongly suggest to evaluate these parameters, e.g. by mass spec on an intact protein level at least for one representative HSP70 isoform. Knowledge about the rate of covalent capture would also be important to put the data obtained in the time-dependent analysis of epichaperomes into context!

- a binding mode of analogous compounds to an allosteric pocket in the N-terminal domain of Hsp70 was suggested in previous studies based on docking and MS. Structural information confirming this model (e.g. Xray or NMR) would make the study much stronger in my opinion. Is there any structural basis to rationalize the observed preferable binding to epiHSP70s over other Hsp70 pools?

- compound selectivity was validated by CETSA, pulldown and a kinase screening panel. Notably, cell lysate CETSA experiments were incubated 30 min while 1.5 h were used for intact cell experiments. Covalent binding is time-dependent so this should make a difference. Moreover, heating should accelerate covalent bonding, also with respect to off targets. These factors require some discussion. Interestingly, 100 μ M of the covalent probe were used for incubation. This is a very high concentration for a covalent compound and it is hard to believe that the compound does not significantly modify (or reversibly bind to) off-targets at these concentrations. Could it be that off-targets have been overlooked, e.g. due to the high abundance of Hsp70 compared to many other proteins? And how does time-dependence impact the fraction of the cellular Hsp70s pool captured?

-although this may be beyond the expectations for a revision, complementary validation of selectivity of the covalent probes by alternative MS-based chemoproteomic methods (e.g. using a stable isotope labeling-based workflow as described in 10.1038/nature18002) would put the results and their interpretation on an even more robust basis.

- it is nice to see that the authors make use of a structurally related negative control, i.e. regioisomer YK56. For covalent compounds, however, removing the reactive group typically strongly reduces binding/biological activity. Thus, it is common to include a non-reactive analog (here the corresponding propionic acid amide) as negative control. Why did the authors not synthesize and include this compound? This requires at least some discussion in the manuscript.

- the authors confirm a purity of >95% by LC/MS for their final compounds, but do not provide an exact description of their chromatographic method nor retention times. These data should be added along with HPLC traces for final compounds to the SI.

We sincerely thank the Reviewers for their insightful suggestions and detailed re-review of the revised manuscript. We are also grateful to all five referees who found numerous strengths in this work and deemed this manuscript worthy of publication in *Nature Communications*. We address below the remaining concerns on the topic of chemical probe characterization.

REVIEWER COMMENTS AND POINT-BY-POINT RESPONSE

Reviewer #1 (Remarks to the Author): Expert in HSP70 chaperones in cancer, genetics and signalling

The authors have addressed my very minor concerns.

Response: We thank Reviewer #1 for her/his review of our manuscript.

Reviewer #2 (Remarks to the Author): Expert in protein networks, signalling networks, and bioinformatics

All of my comments have been addressed by the authors beyond any reasonable doubt. I want to express my gratitude to the authors for their diligence in producing a well-written and thorough manuscript.

Response: We thank Reviewer #2 for her/his review of our manuscript and enthusiasm for the report.

Reviewer #3 (Remarks to the Author): Expert in HSP70 and other chaperones, proteomics, and biochemistry

Rodina and colleagues present an application of chemical probes for HSP70 to the affinity purification of HSP70-containing epichaperome structures (epiHSP70s). I appreciate the authors' substantial effort to address my concerns and those of the other reviewers. The methods are now sufficiently detailed, and the conclusions of the manuscript are now well supported. I am happy to support publication of this work at Nature Communications.

Response: We thank Reviewer #3 for her/his review of our manuscript.

Reviewer #4 (Remarks to the Author): Expert in HSPs and modifications in cancer, molecular biology, and mitotic regulation

The authors have made a huge effort to address my and other reviewers' concerns and questions. This manuscript is ready for publication.

Response: We thank Reviewer #4 for her/his review of our manuscript and appreciation for our efforts.

Reviewer #5 (Remarks to the Author): Expert in medicinal chemistry, drug discovery and synthesis

The study by Rodina et al describes the alteration of protein-protein interaction (PPI) networks by "epichaperomes" in cancer cells. The study makes use of small molecule chemical probes covalently binding these epichaperomes, which are long-lived Hsp70-containing scaffolding structures (epiHsp70). The chemical probes were used to trap, enrich, and isolate epichaperome/interactor complexes from proteomes. This pulldown-based epichaperome proteomics platform (refereed to as "epichaperomics") provided systems-level insights into PPI dysfunctions in cancer cell lines. The study presents a wealth of rather complex proteomics data which establishes a link between epichaperomes and the context-dependent reprogramming of mitosis in cancer cells. Overall most of the presented data seems sound and the study is surely of relevance to the general readership of the journal.

R5.1. The chemical synthesis of the compounds included in the manuscript is described in a conclusive manner in the supplementary information. Compounds were fully characterized by ¹H and ¹³C NMR spectroscopy as well as mass spectrometry and the obtained data seems coherent.

Response: Thank you for this kind assessment.

Validation of the epichaperomics chemical probes was performed via CETSA and pulldown. Notably, similar ligands had already been described and characterized in previous studies (see doi 10.1016/j.chembiol.2013.10.008 and 10.1021/cb500256u). While the systems biology part of the manuscript is

very strong, the probe validation leaves some open questions. The past has shown that the use of insufficiently validated chemical probes frequently led to the dissemination of invalid conclusions (see 10.1038/nchembio.1867). Therefore, criteria for the validation of chemical probes, which are now widely accepted by the community have been defined (see for example 10.1038/nchembio.1867 and <https://www.chemicalprobes.org/criteria>). Due to their distinct properties, additional criteria should be considered for covalent compounds (see 10.1039/9781839160745-00069 and 10.1016/j.chembiol.2019.09.012). Actually, I see only a part of this implemented here and in the mentioned previous studies. Virtually all probe validation has been done by indirect assays (pulldown, cellular readouts,...) but little is known about target protein binding. This becomes even more important when considering the covalent nature of probe YK5 and its biotinylated analog both targeting Hsp70-Cys267, which confers time-dependent biological properties. I understand that probe validation for PPI targets is more challenging than for enzymes where convenient assay formats and screening panels are available. Nevertheless, I strongly recommend further validation of the probes used, especially with respect to their time-dependent binding behavior. The following point should be considered:

R5.2.- It seems that IC50s of YK5 and related compounds were only determined in cellular systems. Binding affinities to the target proteins remain obscure. Investigations with isolated proteins or in reconstituted systems would add value and increase compliance with common standards for chemical probe validation.

Response: Binding studies to recombinant protein are of limited use in the context of epichaperomes. We have demonstrated this repeatedly for all the epichaperome chemical probes that came out of our laboratory. Our strategy for epiHSP probe and drug discovery is to use a battery of in cellulo assays rather than recombinant and reconstituted systems. This platform is what enabled the discovery of epichaperome probes with specificity and on-target activity demonstrated not only cellularly, but also *in vivo*, at the organismal level, in both mice and humans (for example the epiHSP90 probes: Bolaender et al. Nature Communications 2021; Pillarsetty et al. Cancer Cell 2019).

As evidenced in this manuscript, multiple structural forms of HSP70s (i.e. folding HSP70s, epiHSP70s, other?), shaped by yet unknown factors, exist in the context of cells. These HSP70 forms differ in their interaction with other chaperones, co-chaperones and proteins, but also, importantly, in their interaction with small molecule modulators. These complex HSP70s variants, including the epiHSP70s, cannot yet be recreated in a test tube. Thus, on-target activity and potency of YK molecules is best reflected by and measured in specific cellular contexts that contain the native target (i.e. epiHSP70s), which we have done here. These experiments were controlled by the use of both cellular (ex. negative control: ASPC1 cells, HSP70s high but epiHSP70s-low/negative) and chemical (ex. acrylamide-containing and structurally similar, yet epiHSP70s inactive such as YK56) controls.

However, we agree that value could be derived from reconstituted systems, especially with regards to the impact of YKs on HSP70s conformations, which may provide insights into their mechanism of action and also on their specificity for epiHSP70 over HSP70. Thus, in response to this critique, we provide further evidence in reconstituted systems using HSP70s, HSP70 regulatory co-chaperones (i.e., Bag1 and DJA2) and an HSP70s substrate, alone or combined. Please see New Supplementary Fig. 16a-h, Supplementary Note 2 and text, page 6. Our data show the biochemical effect of YKs on HSP70s manifests principally at the allosteric release of substrate, impairing substrate release and delaying ATP insertion. YKs minimally affect the ATPase activity of HSP70s, consistent with their specificity for epiHSP70s over HSP70s observed in cellulo. These studies confirm the findings generated in cellulo, and provide further support for the site of binding and the mechanism of binding reported in prior studies from our laboratory.

R5.3.- YK5 and its biotinylated analog act via a covalent mechanism meaning their binding is time-dependent. Determination and discussion of aspects related binding kinetics of these compounds are completely absent in the manuscript. What is the reversible binding component (K_i) of YK5/YK5-B and at what rate does the covalent modification occur (kinact)? I strongly suggest to evaluate these parameters, e.g. by mass spec on an intact protein level at least for one representative HSP70 isoform. Knowledge about the rate of covalent capture would also be important to put the data obtained in the time-dependent analysis of epichaperomes into context!

Response: An analysis of the covalent and non-covalent component to binding and the rate of capture were analyzed and reported in prior studies (Taldone et al., DOI: 10.1021/jm401552y; Kang et al., DOI: 10.1021/jm401551n; Rodina et al., doi: 10.1016/j.chembiol.2013.10.008) and also tested and presented in Supplementary Fig. 15d.

R5.4.- a binding mode of analogous compounds to an allosteric pocket in the N-terminal domain of Hsp70 was suggested in previous studies based on docking and MS. Structural information confirming this model (e.g. Xray or NMR) would make the study much stronger in my opinion. Is there any structural basis to rationalize the observed preferable binding to epiHSP70s over other Hsp70 pools?

Response: Please refer to our **Response R5.2**. Also, we add that the mode of binding for YKs to HSP70s is not derived from docking and MS, but rather it was computationally predicted, experimentally demonstrated in cellulo, biochemically and functionally, and further through medicinal chemistry and structure-activity studies (DOI: 10.1021/jm401552y; DOI: 10.1021/jm401551n; doi: 10.1016/j.chembiol.2013.10.008).

R5.5.- compound selectivity was validated by CETSA, pulldown and a kinase screening panel. Notably, cell lysate CETSA experiments were incubated 30 min while 1.5 h were used for intact cell experiments. Covalent binding is time-dependent so this should make a difference. Moreover, heating should accelerate covalent bonding, also with respect to off targets. These factors require some discussion. Interestingly, 100µM of the covalent probe were used for incubation. This is a very high concentration for a covalent compound and it is hard to believe that the compound does not significantly modify (or reversibly bind to) off-targets at these concentrations. Could it be that off-targets have been overlooked, e.g. due to the high abundance of Hsp70 compared to many other proteins? And how does time-dependence impact the fraction of the cellular Hsp70s pool captured?

Response: In addition to CETSA and pulldown from cells and cell homogenates (thus where thousands of other proteins are found, many more abundant than HSP70) and a 96-kinase screening panel, selectivity was validated through each of the multiple complementary experiments and 40 figures in this manuscript, including extensive functional validation studies, both targeted and unbiased large scale, and through the use of numerous positive and negative controls, both chemical and cellular.

With regards to covalent and unspecific capture of other proteins due to reactivity of the acrylamide, if that was an externality, in Figure 5c we would see no or little difference between the cargo of the YK5-B prior to high salt wash and after high salt wash. Clearly, we observe HSP70 as the major, if not sole, interactor of the probe. The negative control YK56 probe, contains a similarly reactive acrylamide yet captures significantly less target, providing further support for specific target capture. Also, if binding of the probe to HSP70 would occur because of non-specific capture of HSP70 because it is an abundant protein, then we should expect the probe to bind same amount of HSP70 from ASPC1 cells – these have same abundant HSP70s levels but little epiHSP70s. Clearly, this is not the case (see Fig. 1d, Fig. 2f, and the many functional and biochemical studies comparing MDA-MB-468 - epiHSP70 and HSP70 high - and ASPC1 – epiHSP70 low and HSP70 high). If activity of the covalent YK were due to unspecific modification of proteins, the reversible, non-covalent, probes YK198 and LSI137 would fail to mimic the effect of YK5 in CETSA and in the many functional validation studies. As evidenced by our data, this is not the case, indicating the specificity for the target, the epiHSP70. Regarding time-dependence, this feature was previously evaluated and described (Rodina et al., Cell Chemical Biology 2013, Figure 3), and is also presented here (see Supplementary Fig. 15d).

The tacit goal of the present report is not to recapitulate prior studies but rather to validate the use of these probes for systems level investigations into epiHSP70-mediated PPI network dysfunctions in disease. The conclusions of the manuscript are well supported in this regard through the extensive biochemical and functional studies provided here, as also stated by Reviewers#1-4.

R5.6.- -although this may be beyond the expectations for a revision, complementary validation of selectivity of the covalent probes by alternative MS-based chemoproteomic methods (e.g. using a stable isotope labeling-based workflow as described in 10.1038/nature18002) would put the results and their interpretation on an even more robust basis.

Response: Thank you for the suggestion.

R5.7.- it is nice to see that the authors make use of a structurally related negative control, i.e. regioisomer YK56. For covalent compounds, however, removing the reactive group typically strongly reduces binding/biological activity. Thus, it is common to include a non-reactive analog (here the corresponding propionic acid amide) as negative control. Why did the authors not synthesize and include this compound? This requires at least some discussion in the manuscript.

Response: Thank you for the comment. Indeed, the corresponding propionic acid amide was made, characterized and tested, and used as starting point for the discovery of non-covalent probes, as reported (see above Taldone et al., J Med Chem 2014 Feb 27;57(4):1208-24.). The most appropriate chemical controls herein are: *i*). YK56, to exclude contribution of the acrylamide for possible non-specific covalent modification of proteins, *ii*). YK198 and LSI137, non-covalent probes with on-target activity, and *iii*). HSP70 ligands with on-HSP70 activity through other binding mechanisms. Accordingly, we added a sentence to the Methods section "Application of chemical probes".

R5.8- the authors confirm a purity of >95% by LC/MS for their final compounds, but do not provide an exact description of their chromatographic method nor retention times. These data should be added along with HPLC traces for final compounds to the SI.

Response: Done as suggested. NMR spectra and HPLC traces were added for all probes (Supplementary Figs 1-13). See text in Supplementary Note 1 for purification method.

REVIEWERS' COMMENTS

Reviewer #5 (Remarks to the Author):

In the revised version of their manuscript, the authors have added the requested analytical chemistry data to confirm purity and identity of the compounds. Data obtained in a reconstituted HSP70 system indicating that YK probes act on the release of interactor proteins but hardly affected ATPase activity were also added. While most of my other request have not been addressed by additional experimentation, the authors have convincingly explained why many of these experiments are difficult or even unfeasible with the currently available assays. Some additional explanation has also been added to the manuscript. Nevertheless, I think it will be an important future task to gain a more quantitative insight into reversible and irreversible binding contributions of these compounds as well as experimental co-structures. Moreover, and as highlighted in my initial peer review, I would strongly recommend additional probe validation by complementary chemoproteomic methods and hope the authors will consider this advice in their follow-up studies.

Given the vast amount of interesting data presented, I do not want to further delay the publication of the manuscript, which is surely of interest to the Nature Communications readership.

Still, I would ask the authors to do a few additional minor revisions:

a) Briefly re-discuss the aspect of time-dependence of potency and selectivity of covalent compounds in the main manuscript

b) As mentioned already in their 2014 SAR publication (<https://doi.org/10.1021/jm401551n>), the higher reactivity of the aminopyrimine-derived acrylamide in YK5 vs its aniline analog seems to be a driver of potency of YK5. Aminopyridine-derived acrylamides are known to be very reactive and it is nowadays standard in covalent chemical probe validation to quantify intrinsic reactivity, e.g. vs model nucleophiles like GSH. I have not seen such data in any of the authors previous publications. These data can be obtained quickly with a simple LC-MS assay (see e.g. <https://doi.org/10.1021/jm501412a>). The authors should at least assess intrinsic reactivity of YK5 and discuss it in the context of their findings (or cite the corresponding publication if this has been done previously).

We sincerely thank the Reviewers for their insightful suggestions and detailed re-review of the revised manuscript. We are also grateful to all five referees who found numerous strengths in this work and deemed this revised manuscript worthy of publication in *Nature Communications*. We address below the remaining concern on the topic of chemical probe characterization by Reviewer #5.

REVIEWER COMMENTS AND POINT-BY-POINT RESPONSE

Reviewer #5 (Remarks to the Author):

In the revised version of their manuscript, the authors have added the requested analytical chemistry data to confirm purity and identity of the compounds. Data obtained in a reconstituted HSP70 system indicating that YK probes act on the release of interactor proteins but hardly affected ATPase activity were also added. While most of my other request have not been addressed by additional experimentation, the authors have convincingly explained why many of these experiments are difficult or even unfeasible with the currently available assays. Some additional explanation has also been added to the manuscript. Nevertheless, I think it will be an important future task to gain a more quantitative insight into reversible and irreversible binding contributions of these compounds as well as experimental co-structures. Moreover, and as highlighted in my initial peer review, I would strongly recommend additional probe validation by complementary chemoproteomic methods and hope the authors will consider this advice in their follow-up studies. Given the vast amount of interesting data presented, I do not want to further delay the publication of the manuscript, which is surely of interest to the Nature Communications readership.

Response: We thank Reviewer #5 for finding our manuscript of interest and for recommending publication in *Nature Communications*. We appreciate the suggestions for additional probe characterization via complementary interrogation and will implement these requested assessments as part of future studies, as appropriate.

Still, I would ask the authors to do a few additional minor revisions:

a) Briefly re-discuss the aspect of time-dependence of potency and selectivity of covalent compounds in the main manuscript

b) As mentioned already in their 2014 SAR publication (<https://doi.org/10.1021/jm401551n>), the higher reactivity of the aminopyrimine-derived acrylamide in YK5 vs its aniline analog seems to be a driver of potency of YK5. Aminopyridine-derived acrylamides are known to be very reactive and it is nowadays standard in covalent chemical probe validation to quantify intrinsic reactivity, e.g. vs model nucleophiles like GSH. I have not seen such data in any of the authors previous publications. These data can be obtained quickly with a simple LC-MS assay (see e.g.

<https://doi.org/10.1021/jm501412a>). The authors should at least assess intrinsic reactivity of YK5 and discuss it in the context of their findings (or cite the corresponding publication if this has been done previously).

Response: We agree that aminopyridine-derived acrylamide has the intrinsic ability to label reactive macromolecules (i.e., reactive cysteines on proteins), and will react with nucleophiles such as glutathione. This potential limitation is only relevant to the present report if it confounds biological investigation as it pertains to epichaperomics. The current study shows it does not. We use rigorous control probes such as YK56, which has a similarly reactive aminopyridine-derived acrylamide. We employ YK-agents with no aminopyridine-derived acrylamide in their structure, where target binding has no covalent component. We also perform extensive QC evaluations of the YK-interactome through bioinformatics and systems-level and targeted biochemical and functional validation studies to arrive to this conclusion. While we appreciate the formal possibility of nonspecific labeling of proteins containing acrylamide-reactive moieties may occur in the context of epichaperomics assays, these background signals do not obstruct true biological determination or their implications. This principle is tested, and the rationale explained, in the section 'YK5-B epichaperomics'.

In our opinion, a chemical probe, intended to address a specific biological problem in a specific biological context, should be characterized accordingly, and not necessarily in other systems. We made this opinion public on prior occasions and reported it in a piece written by the editors of *Nature Chemical Biology*, stating "Much of our mechanistic knowledge behind cellular processes has been gained by recreating a system *in vitro* or by engineering a cell or a model organism such as yeast, nematodes or mice. One may argue that probing a hypothesis in a model system may, in numerous instances, lead to an understanding that serves the model but has little to do with the real human disease. Having the chemical toolset to probe and validate such models in their real settings is, in my opinion, a historical contribution of chemical biology to biological and biomedical

sciences.” Voices of chemical biology. *Nat Chem Biol* **11**, 446–447 (2015). <https://doi.org/10.1038/nchembio.1845>. Therefore, we have refrained from using other assay systems as suggested by Reviewer #5, as we do not deem them pertinent to the specific questions we have asked (and answered) in this report.

In terms of using glutathione as a reactivity model and unspecific labeling of HSC70, prior large-scale studies evaluated libraries of electrophiles for their ability to label proteins in cells and found “proteomic reactivity of fragment electrophiles was only marginally correlated with their glutathione adduction potential, which is a commonly used surrogate assay for measurements of proteinaceous cysteine reactivity” (*Nature* **534**, 570–574 (2016) <https://doi.org/10.1038/nature18002>). This study noted that despite these electrophile fragments being screened at 500 μ M, a concentration deemed likely of low specificity towards cellular proteins, only a few proteins were labeled. Therefore, “ligand–cysteine interactions are, in general, specific, in that they depend on both the binding groups of ligands and structured sites in proteins” (*Nature* **534**, 570–574 (2016)). Examining Supplementary Table 1 and Figure 3 from this report, where a comprehensive ~6,150 cysteines from ~2,900 proteins were quantified in aggregate across all data sets, we could not find HSC70 (nor any HSP90s for that matter) among the list of labeled proteins. In fact, both HSC70 and HSP90 were listed as proteins with cysteines not labeled by electrophiles.

Moreover, the 2014 SAR publication (<https://doi.org/10.1021/jm401551n>) does not show the reactivity of the aminopyrimine-derived acrylamide in YK5 versus its aniline analog is a driver of potency of YK5. In fact, the J Med Chem and accompanying back-to-back J Med Chem DOI: 10.1021/jm401552y manuscripts demonstrate the contribution of enthalpy to binding, stating “However, in spite of the presence of an acrylamide, derivatives described here are not excessively reactive and, in addition, have an appropriate fit in the active site of the target, anticipating a favorable enthalpic effect.”

With regards to the topic of time dependence on potency and selectivity, we agree with Reviewer #5 this is a salient consideration for covalent probes. One needs however to appreciate, and query, this feature in the context of the specific target in the context of the biological systems under evaluation (as stated above), which we did here and in prior studies (doi: 10.1016/j.chembiol.2013.10.008). It is important to highlight the most relevant feature in the context of binding kinetics is not the aspect of the chemical reaction between the probe and the protein (or proteins) but rather the kinetics of how the probe impacts the target. For the present report this comprises the epichaperome-interactor assemblies in live cells. We show the probes initially trap the assemblies, in the timeframe optimal for epichaperomics, prior to inducing their collapse or disassembly. Thus, in the context of epichaperomes, the kinetics of epichaperome-interactor assembly trapping and collapse are the most important factors. These kinetic properties were determined in our study through IP and native PAGE, and discussed in the Methods section. We reiterate that binding kinetics should be considered and evaluated when designing and implementing the probe for epichaperomics studies.

As suggested by Reviewer #5, we add a summary of the above paragraphs to the Discussion section.